

# Seasonal variability and source apportionment of volatile organic compounds (VOCs) in the Paris megacity (France)

A. Baudic[1], V. Gros[1], S. Sauvage[2], N. Locoge[2], O. Sanchez[3], R. Sarda-Estève[1], C. Kalogridis[1,*],
J.-E. Petit[1,4,**], N. Bonnaire[1], D. Baisnée[1], O. Favez[4], A. Albinet[4], J. Sciare[1,***], and B. Bonsang[1]

[1]LSCE, Laboratoire des Sciences du Climat et de l'Environnement, Unité Mixte CEA-CNRS-UVSQ, CEA/Orme des Merisiers, 91191 Gif-sur-Yvette, France
[2]Mines Douai, Département Sciences de l'Atmosphère et Génie de l'Environnement (SAGE), 59508 Douai, France
[3]AIRPARIF, Association de Surveillance de la Qualité de l'Air en Île-de-France, 75004 Paris, France
[4]INERIS, Institut National de l'EnviRonnement Industriel et des risqueS, DRC/CARA/CIME, Parc Technologique Alata, BP2, 60550 Verneuil-en-Halatte, France
*Now at: Institute of Nuclear Technology and Radiation Protection, Environmental Radioactivity Laboratory, National Center of Scientific Research 'Demokritos', 15310 Ag. Paraskevi, Attiki, Greece
**Now at: Air Lorraine, 20 rue Pierre Simon de Laplace, 57070 Metz, France
***Now at: Energy Environment Water Research Center (EEWRC), The Cyprus Institute, Nicosia, Cyprus

Correspondence to: V. Gros (valerie.gros@lsce.ipsl.fr)

**Abstract.** Within the framework of air quality studies at the megacity scale, highly time-resolved volatile organic compounds ($C_2$ - $C_8$) measurements were performed in downtown Paris (urban background sites) from January to November 2010. This unique dataset included non-methane hydrocarbons (NMHCs) and aromatic/oxygenated species (OVOCs) measured by a GC-FID (Gas Chromatograph with a Flame Ionization Detector) and a PTR-MS (Proton Transfer Reaction - Mass Spectrometer), respectively. The current study presents the seasonal variability of atmospheric VOCs being monitored in the French megacity and their various associated emission sources. Clear seasonal and diurnal patterns differed from one VOC to another as the result of their different origins and the influence of environmental parameters (solar radiation, temperature). Source Apportionment (SA) was comprehensively conducted using a multivariate mathematical receptor modeling. The United States Environmental Protection Agency's Positive Matrix Factorization tool (US EPA, PMF) was used to apportion and quantify ambient VOC concentrations into six different sources. The modeled source profiles were identified from near-field observations (measurements from three distinct emission sources: inside a highway tunnel, at a fireplace and from a domestic gas flue, with hence a specific focus on road-traffic, wood burning activities and natural gas emissions) and hydrocarbon profiles reported in the literature. The reconstructed VOC sources were cross-validated using independent tracers such as inorganic gases (NO, $NO_2$, CO), black carbon (BC) and meteorological data (temperature). The largest contributors to the predicted VOC concentrations were traffic-related activities (including motor vehicle exhaust, 15 % of the total mass on the annual average, and gasoline evaporation, 10 %), with the remaining emissions from natural gas and background (23 %), solvents use (20 %), wood burning (18 %) and a biogenic source (15 %). An important finding of this work is the significant contribution from wood burning, especially in winter, where it could represent up to ~ 50 % of the total mass of VOCs. Biogenic emissions also surprisingly contributed up to ~ 30 % in summer (due to the dominating weight of OVOCs in this source). Finally, the mixed natural gas and background source exhibited a high contribution in spring (35 %, when continental air influences were observed) and in





autumn (23 %, for home heating consumption).

*Keywords :* Urban atmospheric pollution, Volatile organic compounds (VOCs), Ozone precursors, Vehicular emissions, Positive Matrix Factorization (PMF), Source apportionment.

## 1   Introduction

More than half of the world's population is now living in urban areas and about 70 % will be city-dwellers by 2050 (United Nations, 2014). Many of these urban centers are ever-expanding, leading to the gradual growth of megacities. Strong demographic and economic pressures are exerting increasing stress on the natural environment, with impacts at local, regional and global scales. Megacities are hotspots of atmospheric gaseous and particulate pollutants, which are subjects of concern for sanitary, scientific, economic, societal and political reasons. The adverse health effects of outdoor air pollutants are recognized today. Indeed, ambient air pollution has been classified as *carcinogenic to humans* by the International Agency for Research on Cancer since October 2013 (IARC, 2013). In recent decades, air pollution has become one of the most widespread problems in many megacities and should be more investigated.

The understanding of the pollutants in urban areas remains complex given the diversity of their emission sources (unequally distributed in space and time) as well as their formation and transformation processes. Volatile organic compounds (VOCs) are of a great scientific interest because they play an important role in atmospheric chemistry. In the troposphere, primary VOCs take part in chemical and/or photochemical reactions, thus contributing to the formation of ground-level ozone ($O_3$) (Logan et al., 1981; Liu et al., 1987; Chameides et al., 1992; Carter et al., 1994) and secondary organic aerosols (SOA) (Tsigaridis and Kanakidou, 2003 and references therein; Ng et al., 2007). While some megacities face very poor air quality (such as Beijing, Gurjar (2014)) with pollutant concentrations way above recommended thresholds, European megacities experience stagnant pollution levels at the annual scale. However, pollution episodes related to high $O_3$ and PM concentrations still regularly occur, leading to detrimental health consequences.

Epidemiological studies revealed that outdoor air pollution, mostly from $PM_{2.5}$ and $O_3$, could lead to 17 800 premature deaths in France (with 3 100 for the Paris megacity) in 2010 and projections for the future are even worse (3 800 in 2025 and 4 600 in 2050 for Paris) (Lelieved et al., 2015). Paris and its surroundings (also called the *Île-de-France* region) constitute the second largest European megacity with about 12 million of inhabitants, representing 20 % of the French national population distributed over only 2 % of its territory (Eurostat, 2014). Although this region is surrounded by a rural belt, it is considered as a large central urban area where a strong pollution signal can be detected. Deguillaume et al. (2008) have shown that the urban area of Paris was frequently associated with a "VOC-sensitive" chemical regime (also called "$NO_x$-saturated" regime according to Sillman, 1999), for which VOC anthropogenic emission reductions are more effective to decrease ozone levels than $NO_x$ anthropogenic emission reductions. Obtaining accurate knowledge on VOC emissions and sources is consequently essential for $O_3$ and SOA abatement measures.



Qualitative and quantitative assessments of VOC sources and variability have already been conducted within the Paris area during May-June 2007 (Gros et al., 2011; Gaimoz et al., 2011). This study concluded that road-traffic activities (traffic exhaust and fuel evaporation) influenced the total VOC fingerprint, with a contribution of ~ 39 %. This finding was considered as being in disagreement with the local emission inventory provided by the air quality monitoring network AIRPARIF (http://www.
airparif.asso.fr), for which the main contribution was related to solvents usage (from industries and from residential sectors). However, this work was performed over a short period of time (only few weeks). Although it provided valuable information about ambient VOC emissions and sources during a specific period (spring), it did not show their seasonal variations over longer time scales. More resolved observations are therefore required to check the representativity of these first conclusions.

In this context, the EU-F7 MEGAPOLI (Megacities: Emissions, urban, regional and Global Atmospheric POLlution and climate effects, and Integrated tools for assessment and mitigation) (Butler, 2008) and the French PRIMEQUAL-FRANCIPOL research programs involving several (inter)-national partners in the atmospheric chemistry community have been implemented. These MEGAPOLI - FRANCIPOL projects partly consisted in documenting a large number of gaseous and particulate compounds and determining their concentration levels, variabilities, emission sources and geographical origins (local or imported) within the Paris urban area. These experiments go therefore beyond the scope of this paper and a full description of scientific studies conducted under the programme can be found in the special issue "MEGAPOLI - Paris 2009/2010 campaign" available in the Atmospheric Chemistry and Physics (ACP) journal.

Here, this work presents near real-time measurements of VOCs performed at urban background sites in downtown Paris from 15 January to 22 November 2010. Its objectives are to (1) assess ambient levels of a VOC selection, (2) describe their temporal (seasonal and diurnal) variabilities, (3) identify their main emission sources from statistical modeling and (4) quantify and discuss their source contributions on yearly and seasonal bases.

In order to identify and apportion ambient VOC levels to their emission sources, the advanced multivariate receptor modeling technique Positive Matrix Factorization (PMF) was applied. As no prior knowledge of the number or the chemical nature of source profiles is explicitly required (Paatero and Tapper, 1994), the identification of PMF source profile outputs must be made a posteriori. It usually relies on speciation profiles available in the literature. Within this study, near-field additional measurements (at source points: inside a highway tunnel, at a fireplace and from a domestic gas flue) were performed to help strengthen this identification of VOC profiles derived from PMF simulations. This experimental approach is dedicated to provide a specific fingerprint of VOC sources related to road-traffic, residential wood burning activities and domestic natural gas consumption, respectively. The originality of this work stands in using these near-field speciation profiles to refine the identification of apportioned sources.

First, Section 2 will describe (i) sampling sites, (ii) analytical techniques conducted and (iii) two combined approaches for identifying and characterizing the main VOC emission sources. Then, Section 3 will investigate VOC levels and their seasonal and diurnal patterns from ambient air measurements. An accurate identification of PMF factors to real physical sources will be proposed in the Sub-section 3.4. Finally, yearly and seasonal contributions of each modeled source will be discussed in the Sub-section 3.5 and compared to previous studies performed in Paris and widely in the world in Section 4.



## 2 Material & Methods

### 2.1 Sampling sites description

Ambient air measurements of VOCs and ancillary gaseous and particulate pollutants were sequentially performed within the framework of two different research projects. As part of the European EU-F7 MEGAPOLI (Megacities: Emissions, urban,

regional and Global Atmospheric POLlution and climate effects, and Integrated tools for assessment and mitigation (http://www.megapoli.info, 2007-2011)) program, a winter campaign involved measurements of a large amount of atmospheric compounds – with techniques including GC-FID and PTR-MS for VOCs – from 15 January to 16 February 2010 at an urban background site in downtown Paris (Baklanov et al., 2010 ; Beekmann et al., 2015). Following this first project, a second measurement campaign involving less instrumentation (only PTR-MS for VOCs) was conducted at the same location from 24

March to 22 November 2010 (as part of the French PRIMEQUAL-FRANCIPOL program: Impact of long-range transport on particles and their gaseous precursors in Paris and its region (http://www.primequal.fr, 2010-2013)). Nonetheless, no data was available between 16 February and 24 March 2010.

    The main sampling site was the Laboratoire d'Hygiène de la Ville de Paris (LHVP). This site is located in the Southern part of Paris centre (the 13th district – 48°82′ N, 02°35′ E – 15 m above ground level, a.g.l.) and therefore regarded as being

representative of urban background conditions (Gros et al., 2011). From March to November 2010 (e.g. FRANCIPOL sampling period), hydrocarbon measurements by GC-FID were carried out by the regional air quality network AIRPARIF at the "Les Halles" subway station (48°51′ N, 02°20′ E – 2.7 m a.g.l.), also considered as an urban background site located 2 km away from LHVP. The location of these two sampling sites is presented in Fig. 1.

### 2.2 Experimental set-up

#### 2.2.1 VOC measurements using a Proton Transfer Reaction-Mass Spectrometer (PTR-MS)

Within the MEGAPOLI and FRANCIPOL projects, online high-sensitivity Proton Transfer Reaction-Mass Spectrometers PTR-MS (Ionicon Analytik GmbH, Innsbruck, Austria) were used for real-time (O)VOC measurements. As this instrument has widely been described in recent reviews (Blake et al., 2009 and references therein), only a description of analytical conditions relating to ambient air observations is given here.

During these two intensive field experiments, a PTR-MS was installed in a small room located on the roof of the LHVP site (15 m a.g.l.). For the MEGAPOLI winter campaign, PTR-MS measurements performed by the Laboratoire de Chimie Physique (LCP, Marseille, France) have already been described in Dolgorouky et al. (2012). As those performed by the Laboratoire des Sciences du Climat et de l'Environnement (LSCE) during FRANCIPOL have not yet been described elsewhere, more technical details are presented below.

Air samples were drawn up through a Teflon line (0.125-cm inner diameter) fitting into a "DEKABON" tube in order to protect it from light. A Teflon particles filter (0.45-μm pore diameter) was settled at the inlet to avoid aerosols and other fragments from entering the system. The PTR-MS was operating at standard conditions: a drift tube held at 2.2 mbar pressure,



60°C temperature with a drift field of 600 V voltage to maintain an *E/N* ratio of ~ 130 Townsend (Td) [*E*: electrical field strength [V cm$^{-1}$]; *N*: buffer gas number density [molecule cm$^{-3}$]; 1 Td = $10^{-17}$ V/cm$^2$]. First water cluster ions $H_3O^+H_2O$ (at *m/z* 37.0) and $H_3O^+H_2O\,H_2O$ (*m/z* 55.0) were also measured as well as $NO^+$ and $O_2^+$ masses to indicate any leak into the system and assess the PTR-MS performances.

(O)VOC measurements performed in a full-scan mode were enabled to browse a large range of masses (*m/z* 30.0 – *m/z* 150.0). Eight protonated target masses were considered here: methanol (*m/z* = 33.0), acetonitrile (*m/z* = 42.0), acetaldehyde (*m/z* = 45.0), acetone (*m/z* = 59.0), methylvinylketone (MVK) + methacrolein (MACR) + isoprene hydroxy hydroperoxides (ISOPOOHs) (*m/z* = 71.0), benzene (*m/z* = 79.0), toluene (*m/z* = 93.0) and xylenes (*m+p*-, *o*-) + $C_8$ aromatics (*m/z* = 107.0). With a dwell time of five seconds per mass, a mass spectrum was obtained every two to ten minutes for MEGAPOLI and FRAN-

CIPOL campaigns, respectively. Around 80 % of PTR-MS data were validated. Missing data were partly due to background measurement periods and calibrations. The PTR-MS background for each mass was monitored by sampling zero air through a catalytic converter heated to 250°C to remove chemical species. Daily background values were averaged and subtracted from ambient air measurements.

    In order to regularly ensure the analytical stability of the instrument, injections from a standard containing benzene (5.7 ppbv

[parts per billion by volume] ± 10 %) and toluene (4.1 ppbv ± 10 %) were performed approximately once a month from March to November 2010. These measurements have shown that the analyzer stability remained stable during the year with variations within ± 10 %. In addition, two full calibrations were performed before and during the intensive field campaign with a Gas Calibration Unit (GCU, Ionicon Analytik GmbH, Innsbruck, Austria). The standard gas mixture provided by Ionicon contained 17 VOCs at 1 ppmv [parts per million by volume]. These calibration procedures consisted of injecting defined concentrations

(in the range from 0 to 10 ppbv) of different chemicals (previously diluted with synthetic air) with a relative humidity at 50 %. Gas calibrations allowed to determine the repeatability of measurements, expressed here as a mean coefficient of variation. This coefficient was less than 5 % for most of the masses. Slightly higher coefficients were observed for *m/z* 69.0 (isoprene) and *m/z* 71.0 (crotonaldehyde) with 5.6 % and 5.2 %, respectively. Observed differences between both full calibration procedures were from 1.1 % for methanol to 9.8 % for toluene, thus illustrating good analyzer stability over time.

Detection limits (LoD) were calculated as three times the standard deviation of the normalized background counts when measuring from the catalytically converted zero air. For the MEGAPOLI winter campaign, LoD ranged from 0.020 to 0.317 µg m$^{-3}$ whereas they were estimated between 0.018 to 0.330 µg m$^{-3}$ during the FRANCIPOL intensive campaign. The analytical uncertainty on all data was evaluated at ± 20 %.

### 2.2.2 NMHC on-line measurements by Gas Chromatography (GC)

Two different automated Gas Chromatographs equipped with a Flame Ionization Detector (GC-FID) were used in order to continuously measure light ($C_2$ - $C_6$) VOCs in ambient air. The AirmoVOC $C_2$ - $C_6$ analyzer (Chromatotec, Saint-Antoine, France), provided by LSCE, was installed near the PTR-MS at the LHVP site from January to February 2010 (e.g. MEGAPOLI period). An in-depth description of the analyzer, sampling set up and technical information (sampling flows, preconcentration, desorption-heating times, types of traps and columns . . . ) can be found in Gros et al. (2011). For each half-hour analysis, more





than 20 VOCs were monitored. A certified standard gas mixture (NPL, National Physical Laboratory, Teddington, Middlesex, UK) containing in average $4.00$ ppbv of major $C_2$ - $C_9$ NMHCs was used for calibration procedures. The injections of this standard allowed checking compound retention times, testing the repeatability of atmospheric measurements and calculating average response factors to calibrate all the measured ambient hydrocarbons. Detection limits were in the range of $0.013$

(*n*-hexane) – $0.060$ µg m$^{-3}$ (*iso-/n*-pentanes).

From March to November 2010 (e.g. FRANCIPOL period), a GC-FID coupled to a thermo-desorption unit was in operation at the "Les Halles" subway station monitored by the regional air quality network AIRPARIF. Air samples were drawn up at $2.7$ m a.g.l.. 29 hydrocarbons from $C_2$ - $C_9$ were measured during this experiment. A full calibration was performed once a month with a standard gas mixture containing only propane. As the FID response is proportional to the Effective Carbon

Number (ECN) in the molecule, calibration coefficients were calculated for each compound and regularly checked so that they drifted no more than $\pm 5\%$. LoD were assessed at $0.024$ µg m$^{-3}$ for all the selected compounds, excepted for *n*-hexane ($0.013$ µg m$^{-3}$).

### 2.2.3 Additional data available

Some ancillary pollutants and parameters were also measured and used as independent tracers with the aim of strengthening

the identification of VOC emission sources derived from the receptor modeling.

Black carbon (BC) was measured using a 7-wavelength (370, 470, 520, 590, 660, 880 and 950 nm) AE31 Aethalometer (Magee Scientific Corporation, Berkeley, CA, USA) with a time resolution of five minutes. BC data were acquired by this instrument from 15 January to 10 September 2010. Raw data were compensated using the correction algorithm described in Weingartner et al. (2003) and Sciare et al. (2011). BC concentrations issued from fossil fuel and wood burning (BC*ff* and BC*wb*,

respectively) were assessed in accordance with their own absorption coefficients using the "Aethalometer" model described by Sandradewi et al. (2008).

Carbon monoxide (CO) measurements were performed using an analyzer based on InfraRed absorption (42i-TL instrument, Thermo Fisher Scientific, Franklin, MA, USA) with a time resolution of five minutes. Nitrogen monoxide and dioxide (NO, NO$_2$) were measured by chemiluminescence using an AC31M analyzer (Environment SA, Poissy, France) and ozone

(O$_3$) was monitored with an automatic UltraViolet absorption's analyzer (41M, Environment SA, Poissy, France). NO, NO$_2$ and O$_3$ measurements were provided with a 1-min time resolution by the local air quality network AIRPARIF. In addition, Gas Chromatography-Mass Spectrometry (GC-MS) measurements were performed to measure $C_3$ - $C_7$ OVOCs, including aldehydes, ketones, alcohols, ethers and esters during the MEGAPOLI winter campaign. This instrument has been described in details by Roukos et al. (2009).

Meteorological parameters (such as temperature, relative humidity as well as wind speed and direction) were provided by the French national meteorological service "Météo-France" from continuous measurements at the Paris-Montsouris monitoring station, located at about $2$ km away from the LHVP site.

As the measurement frequency was different for each analyzer, a common average time was defined to get all data sets on a similar time step of one hour.



## 2.3 Two combined approaches for characterizing VOC emission sources

### 2.3.1 Bilinear receptor modeling: Positive Matrix Factorization (PMF) tool

Developed about 20 years ago, Positive Matrix Factorization (PMF) is an advanced multivariate factor analysis tool widely used to identify and quantify the main sources of atmospheric pollutants. Concerning VOCs, PMF studies have been conducted in
urban (Brown et al., 2007 – LA, USA; Lanz et al., 2008 – Zürich, Switzerland; Morino et al., 2011 – Tokyo, Japan; Yurdakul et al., 2013 – Ankara, Turkey) and rural areas (Sauvage et al, 2009 – France). For this current study, the PMF 5.0 software developed by the EPA (Environmental Protection Agency) was used in the robust mode from ambient air VOC measurements from January to November 2010. A more detailed description of this PMF analysis is given in Appendix A.

PMF mathematical theory was extensively described in Paatero and Tapper (1994). Briefly, this statistical method consists in decomposing an initial chemically speciated dataset into factor profiles and contributions. Equation (1) summarizes this principle in its matrix form:

$$X = G\,F\,+\,E \tag{1}$$

Where $X$ is the input chemical dataset matrix; $G$ is the source contribution matrix; $F$ is the source profiles matrix and $E$ the
so-called residual matrix.

The initial chemical database used for this statistical study contains a selection of 19 hydrocarbon species and masses divided into ten compound families: alkanes (ethane, propane, *iso*-butane, *n*-butane, *iso*-pentane, *n*-pentane and *n*-hexane), alkenes (ethylene and propene), alkyne (acetylene), diene (isoprene) aromatics (benzene, toluene, xylenes + $C_8$ species), al-
cohol (methanol), nitrile (aetonitrile), aldehyde (acetaldehyde), ketone (acetone) and enones (methylvinylketone, methacrolein and isoprene hydroxy hydroperoxides), which have been measured from 15 January to 22 November 2010 ($n = 6\,445$ with a 1h-time resolution). This combination of hydrocarbon species and masses is similar to that from Gaimoz et al. (2011), excepted for *iso*-butene. Each missing data point was substituted with the median concentration of the corresponding species over all the measurements and associated with an uncertainty of four times the species-specific median, as suggested in Norris et al.
(2014). The proportion of missing values was estimated to be between 19 % (especially for compounds measured by PTR-MS) and 41 % (only for isoprene). This high percentage for isoprene can be mainly explained by analytical problems on GC-FID in July. Despite these constrains, it was decided to take into account this compound as isoprene is a key tracer related to biogenic emissions.

The uncertainty matrix was built upon the procedure described by Norris et al. (2014), adapted from Polissar et al. (1998).
This matrix requires the Method Detection Limit (MDL, here in µg m$^{-3}$) and the analytical uncertainty ($u$, here in %) for each selected species. MDLs were calculated as $3\sigma$ baseline noise and in some cases were homogenized to keep consistency in uncertainty calculations. Species MDLs were ranged from 0.013 to 0.060 µg m$^{-3}$ for NMHCs measured by GC-FID and from



0.020 to 0.330 μg m$^{-3}$ for VOCs measured by PTR-MS. Their analytical uncertainties were respectively estimated at 15 % and 20 % and kept constant over the experiments.

Single species additional uncertainties were also calculated using an equation-based on the Signal-to-Noise (S/N) ratio. As a first approach, Paatero and Hopke (2003) suggested categorizing a species as *Bad* if the Signal to Noise Ratio (SNR) was less than 0.2, *Weak* if it was between 0.2 and 2, and *Strong* if it was greater than 2. *Bad* variables are excluded from the dataset, *Weak* variables get their uncertainties tripled while uncertainties of *Strong* variables stay unchanged. Here, all species exhibited a SNR greater than 3, except for isoprene which had a ratio of 1.7 due to its 41 % missing values recorded. Several empirical tests (for instance, calculating an averaged diurnal pattern to substitute missing values or increasing the analytical uncertainty) were performed in order to better categorize and display this compound. Finally, these different processings resulted in categorizing it as *strong* because no improvement of modeling parameters was obtained. In addition to isoprene, acetonitrile exhibited a SNR less than 3 (S/N ratio of 2.7). It was the only VOC to be defined as *weak* because it may be eventually contaminated with local emissions from laboratory exhausts (although visible spikes of acetonitrile were excluded from the initial dataset). Keeping in mind these limitations for isoprene and acetonitrile, it was decided to include these two compounds into the PMF model because they are considered as relevant tracers for biogenic and wood burning activities, respectively. ΣVOC was defined as *Total Variable* and automatically categorized as *weak* to lower its influence in the final PMF results. No optional Extra Modeling Uncertainty was applied here.

All the *Q* values (*Qtrue*, *Qrobust*, *Qexpected*), scaled residuals, predicted *versus* observed concentrations interpretation and the physical meaning of factor profiles were investigated to determine the optimum number of factors. Although some mathematical indicators pointed towards a 5-factor solution, a source mixing solvents use, natural gas and background emissions was detected. In order to split individually each emission source, the 6-factor solution was then investigated and chosen in terms of interpretability and fitting scores. More technical details are reported in Appendix A, Sub-section A.2.

PMF output uncertainties were estimated using two Error Estimation methods starting with DISP (d*Q*-controlled displacement of factor elements) and finally processing to BS (Classical bootstrap). The DISP analysis results were considered validated: no error could be detected and no drop of *Q* was observed. As no swap occurred, the 6-factor PMF solution was considered sufficiently robust to be used. Bootstrapping was then carried out, executing 100 iterations, using a random seed, a block size of 874 samples and a minimum Pearson correlation coefficient (R-Value) of 0.6. All the modeled factors were well reproduced through this bootstrap technique over at least 88 ± 2 % of runs, hence highlighting their robustness. A low rotational ambiguity of the reconstructed factors was found by testing different degrees of rotations of the solution using the Fpeak parameter (Fpeak = 0.1).

### 2.3.2 Determination of source profiles from near-field observations

As emphasized in Paatero and Tapper (1994), a Positive Matrix Factorization analysis does not require a priori knowledge on the chemical nature of factor profiles. To help strengthen the identification of VOC emission sources derived from this



statistical tool, near-field additional measurements (at limited source points) were worthwhile. These in-situ observations were performed close to specific local emission sources in real conditions as far as possible. They aim at providing chemical finger-prints (considered as reference speciation profiles) of three significant VOC sources representative within the Paris area: road traffic (i), residential wood burning (ii) and domestic natural gas consumption (iii). The speciated profiles of these different

anthropogenic sources and their representativity are specially given here. All the technical details of these experiments are reported in Section S1 in the Supplement.

### Highway tunnel experiment

Road-traffic was considered to be one of the most significant sources of primary hydrocarbons in many megacities (Seoul –

Na, 2006; Los Angeles – Brown et al., 2007; Zürich – Lanz et al, 2008), including Paris (Gros et al., 2007, 2011; Gaimoz et al., 2011). The accurate characterization of the vehicle fleet footprint is therefore important. Consequently, near-field VOC measurements (within the "PRIMEQUAL – PREQUALIF" program) were conducted inside a highway tunnel located at about 20 km southeast of inner Paris centre in autumn 2012.

This first experiment has the advantage of supplying a realistic assessment of the average chemical composition of vehicular

emissions, as these in-situ measurements were performed under on-road real driving conditions. Most of VOCs emitted from road-traffic are representative of local primary emissions (due to their relatively short lifetimes). Photochemical reactions leading to changes in the initial composition of the air and to the formation of secondary products can be considered of minor importance.

VOC levels during traffic jam periods (07-09 am – 5:30-07 pm, Local Time) were considered as the most representative

values of vehicular emissions. In order to get out from any local background, nighttime values (as suggested in Ammoura et al., 2014) were subtracted from the peak VOC concentrations. The mass contribution of 19 selected compounds was calculated and reported in Fig. 2. Each compound is expressed in terms of w/w of the Total VOC (TVOC) mass. The two predominant species measured inside the highway tunnel were toluene (20.1 %, 27.1 µg m$^{-3}$ on average) and *iso*-pentane (19.3 %; 26.0 µg m$^{-3}$). The next most abundant VOCs were aromatics (benzene, $C_8$) and oxygenated compounds (acetaldehyde and methanol

– ~ 6.5 %; ~ 8.7 µg m$^{-3}$). In addition, significant contributions of light alkenes (ethylene, propene, acetylene; 3.4 % – 4.6 %; 4.6 – 6.2 µg m$^{-3}$) and alkanes (such as butanes, *n*-pentane, *n*-hexane, methyl alkanes) were also noticed. These observations were found to be consistent with the literature (Na, 2006; Araizaga et al., 2013 concerning NMHCs and Legreid et al., 2007 for OVOCs measurements) and more importantly with the study from Touaty and Bonsang (2000), for which *iso*-pentane, ethene, acetylene, propene and *n*-butane were considered as the major aliphatic compounds observed in the same highway tunnel in

August 1996 (Aromatics and OVOCs were not measured during this study).

### Fireplace experiment

Residential wood burning activities have been shown to be a significant source of (O)VOCs to local indoor and outdoor air pollution during winter months in urban areas (Evtyugina et al., 2014). Currently, only a few studies about the characterization

of VOCs from wood burning have been conducted in Europe (Gustafson et al., 2007; Gaeggeler et al., 2008). After being subject





to lively debates within the Île-de-France region, an in-depth investigation of this source would therefore appear necessary to better understand its emission specificities and its potential impacts on atmospheric chemistry.

In order to complete the information on wood burning activities, VOC measurements (within the "CORTEA-CHAMPROBOIS" program) were performed at a fireplace facility located ~ 70 km northeast of the inner Paris centre in March 2013. On-line (PTR-MS) and off-line (sampling flasks analyzed later on at the laboratory with a GC-FID) measurements were performed. These in-situ observations represent a more qualitative (predominant species identification) than quantitative approach as the resulting speciation profile is based on a limited number of data. As illustrated in Fig. 3, 19 VOC species could be detected. $C_2$ hydrocarbons (ethylene, acetylene), $C_6$ - $C_8$ aromatics (benzene, xylenes) and oxygenated species (methanol, acetaldehyde and acetone) can be considered as predominant compounds from domestic wood burning. This finding is still consistent with intensive field studies of wood burning performed in Europe (Barrefors and Petersson, 1995; Gaeggeler et al., 2008; Evtyugina et al., 2014).

#### Natural gas experiment

Natural gas is predominately composed of methane ($CH_4$) accounting for at least 80 % of the total chemical composition. It is also a mixture of other pollutants including lightweight VOCs and lower paraffins (approximately 10 % by volume).

In a first approach to determine the speciation profile from natural gas used in Paris, near-field samplings were performed from a domestic gas flue using three stainless-steel flasks, which have been analyzed by GC-FID at the laboratory. Main results (Fig. 4) show a large dominance of alkanes, such as ethane (~80 %), propane (~11 %) and heavier hydrocarbons (like butanes, pentanes) ranging from 4.5 to 0.4 %. Ethane and propane therefore appear as a significant profile signature of natural gas leakages (Passant, 2002).

## 3   Results

### 3.1   Meteorological conditions during the 11-months observation period

Meteorological parameters are key factors governing seasonal and diurnal variations of air pollutant levels. Weather indicators (temperature, relative humidity, rainfall, sun exposure and boundary-layer height) representative of the Paris region were analyzed from January to November 2010.

Monthly average weather conditions observed during the studied period were fairly comparable to standard values determined by the French national meteorological service "Météo-France" (available at: http://meteofrance.com), with however an uncommon cold and snowy wintertime. Mean monthly temperatures recorded in January and February 2010 were respectively between -2°C and -3.5°C below monthly normal values. Several unusual cold outbreaks and a few flurries affected the Paris region, thus explaining higher temperature anomalies during that period. Levels of rainfall and hours of sunshine were globally consistent with standard values. In addition, atmospheric boundary layers showed seasonal changes with mean heights up to ~ 800 m in winter and up to 1 600 m in spring and summer occurring during the afternoon (see Section S2 in the Supplement).




These average seasonal heights are expected to play a key role in pollutant dispersion and consequently impact ambient VOC concentrations.

In 2010, Paris was mainly influenced by winds coming from the northwest to south (Fig. 5 - panels a, c, g-k). These wind sectors are characterized by clean air masses originating from the Atlantic Ocean (with wind speeds up to 10 m.s$^{-1}$) and usually associated with local and regional pollution conditions. The average wind speed recorded was mostly in the range from 2.7 to 4.3 m.s$^{-1}$, thus illustrating the local origin of emissions from Paris and its surroundings. To a lesser extent, north and northeast winds especially blew during springtime (Fig. 5 - panels d, e and f), due to the stagnation of an anticyclone surrounding the British Isles (Monthly weather report for Paris and its surroundings during April 2010, Météo-France) and leading to continental air masses influences from the eastern Europe, the Benelux area and the north of Germany.

## 3.2 VOC concentration levels in ambient air

The main results of descriptive statistics for all the measured VOCs (from both GC-FID and PTR-MS instruments) on the whole sample set were summarized in Table 1. From this table, it was observed that the average composition in VOCs was mainly characterized by oxygenated species (0.7 – 5.9 µg m$^{-3}$; 36.5 % of the TVOC mass), alkanes (0.5 – 4.6 µg m$^{-3}$; 39.1 %) followed by aromatics (1.1 – 3.3 µg m$^{-3}$; 16.9 % ) and to a lesser extent by alkenes, alkynes and dienes (0.3 – 1.6 µg m$^{-3}$; 7.5 %). Both alkanes and OVOCs significantly contribute to the tune of 75 % of the TVOC concentrations. With ethane (10.9 %, 4.6 µg m$^{-3}$ on average) being the main alkane, methanol (14.0 %, 5.9 µg m$^{-3}$) and acetone (11.6 %, 4.9 µg m$^{-3}$) are considered to be the two major oxygenated compounds measured in this study. This conclusion is in agreement with previous VOC measurements performed in downtown Paris in 2007 (Gros et al., 2011).

The comparison between these average ambient levels and VOC measurements reported in the literature for different urban areas is restricted here to PTR-MS data as they constitute the most original dataset of this study. Most atmospheric studies were indeed conducted in urban metropolitan areas by investigating only NMHC measurements.

Table 2 summarizes PTR-MS data collected during the intensive experiment together with average VOC levels reported from other cities around the world. For the Paris megacity, a significant decrease in VOC concentrations was observed between spring 2007 and spring 2010 (from - 52.7 % for xylenes and C$_8$ aromatics to - 30 % for benzene and MVK+MACR+ISOPOOHs), excepted for methanol and acetaldehyde (+ 10 % , + 36 %, respectively). Among selected species, benzene (as a carcinogenic compound) is one of the few regulated VOCs. According to the Directive 2000/69/EC, the annual mean benzene concentration in ambient air should not exceed 5 µg m$^{-3}$. Background levels of benzene were relatively stable in recent years, with an annual average concentration of 1.1 µg m$^{-3}$ (Airparif, 2015).

Average VOC concentrations were also calculated in line with sampling periods of the other European and Global studies over different years (see Table 2). VOC levels measured in this study were in the range of those found within some European cities (Barcelona, London – from -0.2 to -3.7 µg m$^{-3}$). However, average VOC levels observed in Paris were significantly lower than mean concentrations measured in Houston (USA – from -0.1 to -9.3 µg m$^{-3}$) and more particularly in Beijing (China – from -1.8 to -13.3 µg m$^{-3}$), in Mohali (India – from -1.5 to -43.1 µg m$^{-3}$) and Mexico City (Mexico – from -0.1 to -86.3 µg m$^{-3}$). Observed differences can be partially explained by different enforcements of governmental rules and standards



to control pollutant emissions as well as the location of sampling points (or distances from main sources) and meteorological conditions, thus strongly affecting VOC concentrations in the considered urban environments.

## 3.3   Seasonal and diurnal time variations

Variability in VOC concentration levels is controlled by a combination of factors including source strengths (e.g. emissions),
dispersion and dilution processes as well as photochemical reaction rates with OH radicals and other oxidants (Filella and Peñuelas, 2006). Variations of selected trace gases (nitrogen/carbon monoxide, NO/CO – Fig. 6) and VOCs illustrating contrasting emission sources and atmospheric lifetimes were analyzed at different time scales. As (O)VOCs measured by PTR-MS constitute the most original data of this study (and representing ~ 37 % of the TVOC mass), a discussion on their variations (Fig. 7) and their respective sources is given here. An overview of seasonal and diurnal profiles of lighter hydrocarbons ($C_2$ -
$C_6$) measured by GC-FID is reported in Section S3 in the Supplement.

Known as combustion tracers (traffic, wood burning), nitrogen monoxide (NO) and carbon monoxide (CO) exhibit higher median concentrations during winter and in late autumn, while lower concentrations appear in summer (Fig. 6a, 6b). These low levels can be explained by greater photochemical reaction rates (linked to higher solar radiation) combined with a stronger vertical atmospheric mixing compared to the other seasons. Another explanation is the increase in NO and CO emissions due to home heating fuels consumed in winter. NO concentrations are significantly enhanced between 06 am and 12 pm LT (with a
maximum peak around 09 am). Contrary to NO, the diurnal pattern of CO is characterized by a "double wave" profile with an initial increase at 07-10 am (maximum peak at 09 am) and a second increase at the end of the afternoon between 04 and 08 pm. These increases typically correspond to morning and evening rush-hour traffic periods, as previously observed in Ammoura et al. (2014). The evening peak is smaller in magnitude than the morning one partly due to a higher Planetary Boundary-Layer
(PBL) height in the afternoon leading to dispersion and dilution processes and to more disperse traffic periods. This evening event is not observed for NO as during this time ozone ($O_3$) presents its highest concentrations, leading to the titration of NO.

Good correlations between CO and some alkanes (*iso-/n*-pentane, *n*-hexane), alkenes (ethylene, propene), acetylene and aromatics (benzene, toluene and xylenes + $C_8$) were found when considering a Pearson's correlation coefficient $r$ greater than 0.6. All these compounds follow a similar seasonal and diurnal pattern, indicating that they share some or almost all
common sources related to anthropogenic combustion processes (e.g. road traffic and/or wood burning). These observations are in agreement with the conclusions from Gros et al., 2011 and Gaimoz et al., 2011.

With atmospheric lifetimes from a few hours to several days, oxygenated species (OVOCs) are emitted from primary sources mainly of biogenic origins and significant secondary sources related to the oxidation of hydrocarbons. High concentration levels of OVOCs (for instance, methanol) and CO were observed during winter months (the season with coldest temperatures
and where wood burning-related activities can play an important role). The low height of the PBL is also a relevant factor to consider as it can lead to the accumulation and the stagnation of VOC species into the troposphere during that season. In addition, significant OVOC levels were observed from April to September. In springtime, elevated baseline levels were measured when the Paris region was mostly influenced by continental air-masses (see Fig. 5d, 5e and 5f). This finding suggests





that these high compound levels partly depended on continental imported and already processed air-masses. Biogenic emissions indeed contributed to high OVOC concentrations during this season and in summer.

With an atmospheric residence time of 12 days, methanol is usually released into the atmosphere by vegetation and man-made activities contributing to a relatively high background level during most of the year. This compound displays a specific
diurnal pattern depending on the season and atmospheric dynamics (see Section S4 in the Supplement). In winter and autumn, methanol shows a "double wave" profile with two increases at 10-11 am and 07-08 pm, suggesting the influence of anthropogenic activities (e.g. road-traffic, wood burning sources). A slight delay (1-2 hours) is observed for methanol in comparison with other primary species (for instance, aromatics). In spring and summer, methanol is characterized by high concentrations during night hours (12 - 06 am), followed by a significant decrease until the early afternoon and another increase from 06 pm
to midnight (Fig. 7c). This night-time maximum of methanol has already been observed in urban environments however with no clear explanation (Solomon et al., 2005). This diurnal cycle can possibly be interpreted as the accumulation of species concentrations during the night from a local source under a shallow inversion layer, which is decreasing when the Boundary-Layer Height (BLH) is increasing (as dilution and dispersion processes occurring). However, the corresponding nighttime source has not been yet identified.

With a relatively short lifetime (~ 9 hours), acetaldehyde shows a diurnal cycle fairly comparable to acetone (Fig. 7d, 7e). Lower concentrations were observed during the night and from 06 pm. Average levels increase from sunrise to a maximum at noon and slightly decrease in the afternoon. For these two OVOC species, the reduction of concentrations does not occur in the same way: acetaldehyde concentrations decrease linearly whereas acetone levels decrease over a long time period. This finding depends on their different photochemical reaction rates ($1.5 \times 10^{-11}$ cm$^3$ molecule$^{-1}$ s$^{-1}$ for acetaldehyde and $1.8$
$\times 10^{-13}$ cm$^3$ molecule$^{-1}$ s$^{-1}$ for acetone) [Atkinson et al., 2006] and their respective emission sources and strengths. As acetone has a relatively long atmospheric lifetime (~ 68 hours), concentration levels are often more homogeneous.

Finally, methylvinylketone, methacrolein and isoprene hydroxy hydroperoxides (MVK+MACR+ISOPOOHs), three photo-oxidation products of isoprene (as a good indicator of biogenic activities), exhibit high levels in the late afternoon due to the oxidation of daytime isoprene. High emissions of these compounds mostly occur in summer, but also in winter (Fig. 7f). This
fact could eventually be related to anthropogenic activities such as wood burning (see Sub-section 2.3.2, Fig. 3).

## 3.4 Source apportionment

The chemical composition of the modeled factors issued from PMF simulations presented in the Sub-section 2.3.1 is reported in Fig. 8. The source profiles were identified from a comparison with near-field observations (see Sub-section 2.3.2) and with reference profiles from literature. The temporal source variations were assessed with independent parameters used as tracers
of specific sources like inorganic gases (NO, NO$_2$, CO), black carbon (BC) and meteorological data (temperature). Seasonal and daily patterns of these VOC sources were also investigated and are presented in Fig. 9.



### 3.4.1    Motor Vehicle Exhaust factor

The speciation profile of Factor 1 (see Fig. 8a) exhibits high contributions of alkanes, such as pentanes (*iso-*; *n-*) and *n*-hexane with on average ~ 50 % of their variabilities explained by this factor. Aromatic compounds (toluene, xylenes + $C_8$, benzene; ~ 35 %) and light alkenes (ethylene, propene), which are considered as typical combustion products, are also the

predominant species in this factor. Such compounds like *iso*-pentane and toluene have already been identified in the highway tunnel experiment (see Sub-section 2.3.2, Fig. 2), thus allowing to label this profile as "Motor vehicle exhaust". This factor 1 displays good correlations with nitrogen monoxide/dioxide ($NO/NO_2$), carbon monoxide (CO) and black carbon (from its fossil fuel fraction), which are known to be relevant vehicle exhaust markers ($0.53 < r < 0.64$).

The average contribution of this factor is rather stable throughout the year ($5.8$ µg m$^{-3}$). A smaller contribution is found

during winter ($3.2$ µg m$^{-3}$) whereas the highest emissions from motor vehicle exhaust occur in autumn ($8.6$ µg m$^{-3}$ on average) with a contribution of up to $10.1$ µg m$^{-3}$ in September. This seasonal cycle has already been observed and described in Bressi et al. (2014) for the road-traffic source of fine aerosols in Paris. The diurnal variation of this source is characterized by a "double wave" profile with an initial increase at 07-10 am (LT, Local Time) and a second increase at the end of the afternoon between 04 and 07 pm (LT). These increases correspond to morning and evening rush-hour traffic periods. Lower

contributions are generally displayed on late mornings/early afternoons and at night. This reduction in factor contributions could be mainly explained by dilution and OH oxidation processes of more reactive species, which are not being balanced by additional vehicular emissions. This pronounced cycle has already been reported in previous studies (Gaimoz et al., 2011 and references therein). The temporal source strength variation is usually much more pronounced during weekdays than the weekend.

### 3.4.2    Gasoline Evaporation factor

The profile of Factor 2 (see Fig. 8b) exhibits a high contribution from propane and *iso-/n*-butanes, with more than 47 % of their variabilities explained by this factor. It was already identified by Gaimoz et al. (2011) and is used here as reference profile from "gasoline evaporation" emissions (including storage, extraction and distribution of gasoline or Liquid Petroleum Gas). Known to be a key tracer for evaporation, *iso*-pentane does not appear in this speciated profile but is significantly observed in

the motor vehicle exhaust source (Factor 1). Factor 2 also includes a significant proportion of isoprene (20 %). This finding is still consistent with the conclusions of Borbon et al. (2001), which have shown that traffic activities emit a small amount of isoprene. In the same way, oxygenated compounds (acetaldehyde (4 %), acetone (6.6 %)) were found in fugitive evaporative emissions in agreement with what was observed during the highway tunnel experiment (see Fig. 2).

Among independent tracers used, only NO displays a fair agreement with this factor ($r = 0.35$). A correlation between F2

and F1 can also be noted ($r = 0.36$), thus indicating that these two factors are related to a common source (e.g. road-traffic). This gasoline evaporation source is in the range of ~ $3.9$ µg m$^{-3}$ over the whole studied period. The annual trend of F2 seems to be consistent with the motor vehicle exhaust factor (F1), even though its monthly change remains ambiguous. Lower evaporative contributions are recorded both in winter and in early summer with minimum average contributions in June and





July (1.7 µg m$^{-3}$). This finding was already identified by Frachon H., 2009 (personal communication). This value in June is somewhat puzzling as road-traffic emissions are usually significant (4.9 µg m$^{-3}$). In July, propane and butanes (*iso-/n-*) values were missing due to analytical problems on the operating GC-FID. Consequently, these compounds were simulated by the PMF model (e.g. missing values were virtually substituted by median values) which may underestimate the contribution

of this factor during this specific period of time. However, high contributions of this source occur in August (6.4 µg m$^{-3}$). Although exhaust emissions are not particularly important, this observation could be eventually explained by gasoline storage and distribution sources, which may have increased with higher temperatures during that month. Maxima temperatures have generally been in the range of 16 to 32°C. The gasoline evaporation source contribution is in average higher in autumn (6.1 µg m$^{-3}$) with a contribution of up to 6.3 µg m$^{-3}$ in October.

The diurnal variation of this factor contribution is characterized by a nighttime minimum, an increase from 07 to 10 am (consistent with the motor vehicle exhaust factor, F1) and a much slower decrease in emissions during the afternoon than those observed for the vehicle combustion profile. This second factor therefore represents the emissions of less reactive species (OVOCs, propane, butanes), for which concentrations cannot be expected to be consumed photochemically in short transport times. The temporal source strength variation is less pronounced on weekends than weekdays, which is typical of mobile source

activity patterns.

According to the Copert IV (European Environment Agency, EEA) program for the calculation of air pollutant emissions from road transport, gasoline evaporation emissions can be explained by the evaporation of VOCs due to temperature, vehicle refueling, running losses, diurnal and "hot soak" reactions (when a hot engine is switched off). It was speculated that hot engines would emit more in the morning than in the evening, considering typical conditions of active inhabitants going to

and from their workplace. Fugitive gasoline emissions from the loading of tank trucks, transportation and unloading from tank trucks at service stations and distributions depots can also be likely sources of this factor. In summary, this gasoline evaporation source depends on several parameters (related to road-traffic conditions, the vehicle fleet composition, economic activities and meteorological observations), which can make the interpretation of its seasonal variability difficult.

### 3.4.3 Wood Burning factor

In Paris, domestic wood burning represents a non-negligible part (about 5 %) of the energy consumption by fuel used for home heating (Airparif, 2011). The chemical profile of this source (Factor 3), shown in Fig. 8c, is mainly dominated by acetylene with approximately 80 % of its variability explained by this factor. It also includes ethylene (57.4 %), benzene (22.7 %) and oxygenated compounds, such as acetonitrile, acetaldehyde and methanol (with 18.3 %, 12.6 % and 8.2 %, respectively). These chemical species typically reflect an anthropogenic source related to wood combustion processes (Lanz et al., 2008; Leuchner

et al., 2015), in agreement with the fireplace emission profile (see Sub-section 2.3.2, Fig. 3). Furthermore, acetonitrile is a hydrocarbon commonly used as a marker of biomass burning (Holzinger et al., 1999). This profile F3 is therefore labelled "wood burning" factor.

Biomass burning emissions are well correlated with black carbon originating from residential wood burning (BC*wb*) and carbon monoxide, a long-lived compound especially emitted from combustion reactions ($0.6 < r < 0.7$). In addition, they well



co-vary with napthalene (*m/z* 129.0 measured by PTR-MS) - a known polyaromatic hydrocarbon emitted from combustion processes (industry, tailpipe emissions) including wood burning (Purvis and McCrillis, 2000). As expected, wood burning contributions display a distinct cycle with a winter maximum (20.5 µg m$^{-3}$ on average) and a summer minimum (3.3 µg m$^{-3}$). Average contributions of this factor are rather stable in both spring and fall (6.9 and 5.9 µg m$^{-3}$, respectively).

Wood burning emissions linked to home/building heating are obviously highly dependent on meteorological conditions and particularly with cold temperatures. A clear negative relationship between the wood burning factor and temperature is found (*r*= - 0.56). The diurnal variation of this source exhibits a "double wave" profile. Average contributions increase from sunrise to a maximum in midmorning and decrease until 04 - 05 pm. At the end of the day, a second increase is observed with another maximum contribution at 07 - 09 pm. This diel cycle can be explained by domestic behaviors. An important finding is that the diurnal pattern of this source is fairly comparable to that of the motor vehicle exhaust factor. However, the wood burning factor does not display any distinct weekly variation. High contributions are observed all week (without any distinction between weekdays and weekends) compared to motor exhausts, for which vehicular emissions are less pronounced on weekends than weekdays. In addition, it exhibits poor correlations with NO, NO$_2$ and BC*ff* (*r* = 0.30, 0.29 and 0.19, respectively), thus indicating that the wood burning factor is completely independant of the motor vehicle exhaust source.

### 3.4.4 Biogenic factor

The profile of Factor 4 (see Fig. 8d) exhibits a high contribution from isoprene, a known chemical marker of biogenic emissions, with more than 79 % of its variability explained by this factor. In addition, this factor profile includes the isoprene's oxidation products (methylvinylketone (MVK), methacrolein (MACR) and isoprene hydroxy hydroperoxides (ISOPOOHs)) with more than 48 %, methanol and acetone - a selection of compounds having a large contribution from biogenic emissions (Kesselmeier and Staudt, 1998; Guenther, 2002). It also accounts a significant contribution of some light alkenes (e.g. ethylene and propene), which can be evenly emitted by plants (Goldstein et al., 1996). Consequently, this factor F4 is termed "biogenic factor". Amounts of light alkanes (butanes, *iso*-pentane, *n*-hexane) were also found in this profile and could be attributed to a mixing with other temperature-related sources or artefacts from the PMF model (Leuchner et al., 2015).

Biogenic emissions are directly related to Photosynthetically Active Radiation (PAR) and ambient temperature (*r* > 0.7). For that reason, they display a diurnal variation with higher contributions during the day and lower values at night-time. The average contribution of this source is approximately estimated at 5.9 µg m$^{-3}$ during the studied period. The highest biogenic factor contributions occur in summer (10.5 µg m$^{-3}$ on average) with a contribution of up to 14.3 µg m$^{-3}$ in July.

### 3.4.5 Solvents use factor

The profile of Factor 5, shown in Fig. 8e, is associated with a large contribution of selected OVOCs (acetaldehyde, methanol and acetone) with on average ∼ 33 % of their variabilities explained by this factor. Significant contributions from aromatic compounds (toluene, xylenes + C$_8$ and benzene) and some alkanes (pentanes, butanes, propane and *n*-hexane) are also observed. Toluene, in addition to road-traffic, is a good marker for solvents originating from an industrial source (Buzcu and Fraser, 2006). Benzene, due to its toxic and carcinogen nature, was regulated in recent years and is strongly limited in solvent





formulations. Current standards establish limits in benzene concentrations at 0.1 % in cleaning products. However, PMF results point out the presence of benzene in this factor, suggesting that this compound might potentially still be in use by some manufacturers. Finally, the presence of these aforementioned species illustrates that this profile could be linked to industrial emissions, although a mixing of different sources cannot be excluded.

This factor co-varies well with ethanol, butan-2-one (also called methylethylketone – MEK), isopropyl alcohol or even ethyl acetate ($0.68 > r > 0.52$, respectively) – four organic compounds that were measured by GC-MS during the MEGAPOLI campaign (January-February 2010). These species are often used as solvents, diluents or cleaning fluids in industrial processes (Zheng et al., 2013). Some manufactories can consume fossil fuels for their activities, which may explain the fairly good correlation between this factor and black carbon originating from fossil fuels (BC$ff$, $r = 0.50$). Indeed, these fossil fuels could

be used by industries as diverse as paints, paintings inks and lacquers (Tsai et al, 2001; Cornelissen and Gustafsson, 2004).

The highest contribution of this factor is observed during winter (14.2 µg m$^{-3}$) with a contribution of up to 20.9 µg m$^{-3}$ in January. Chemical lifetimes of concerned species are enhanced due to lower OH concentrations and weaker UV radiation compared to summer. In addition, the height of the PBL is often lower in winter than in summer, thus leading to the accumulation of these species into the atmosphere and explaining the diurnal pattern of this source during that season (Fig. 9, panel 5). It dis-

plays enhanced contributions from 6 am to a maximum in the afternoon in agreement with the diel cycle of independent tracers (ethanol, butan-2-one). Reconstructed mass concentrations associated with this factor are also significant in summer (12.6 µg m$^{-3}$ in July), which could be mainly explained by the evaporation of solvent inks, paints and other applications during that month due to higher temperatures. Its diurnal variability is changing according to the season. In summer, it is characterized by an initial increase at 08 – 10 am (with a maximum at 09 am) and a decrease from 10 am to 03 pm, explained by more intense

photochemical reactions and temperatures. A slight enhancement at 03 – 04 pm (maybe corresponding to business resumption) and a small decrease in concentrations at the end of the day are observed. The temporal source strength variation is much more pronounced during weekdays than the weekend, excepted on Saturday morning. These diel and weekly patterns seem to be consistent with industrial source activities.

### 3.4.6 Natural Gas and Background factor

The profile of Factor 6, shown in Fig. 8f, is mainly dominated by ethane with around 45 % of its variability explained by this factor. It also contains propane (14.7 %) and light alkanes (butanes), which are key long-lived compounds known to be associated with natural gas leakages. Such species have already been identified in the natural gas experiment (see Sub-section 2.3.2, Fig. 4), thus allowing to confirm the identification of this profile. The diel pattern of this factor is mainly based on the diurnal variation of ethane, which is characterized by a nighttime maximum and a mid-afternoon minimum. Mainly due to

its low reactivity, the behavior of ethane can be interpreted as homogeneous species levels during the night under a shallow inversion layer, then followed by concentration reductions caused by the increase of the BLH and vertical mixing - leading to dispersion and dilution processes. Average contributions of this factor were significantly higher when the BLH was low (~11.0 – 14.0 µg m$^{-3}$) and lower when the BLH was high (~ 6.0 µg m$^{-3}$).



This F6 profile is also characterized by the presence of oxidized pollutants (OVOCs including acetone and methanol) and aromatic compounds (like benzene), which have relatively long atmospheric residence times of respectively 53, 12 and 9 days (assuming OH = 2.0 x $10^6$ molecules cm$^{-3}$) [Atkinson et al., 2000]. Because of their low reactivity, all the species of this factor tend to accumulate in the atmosphere and show significant background levels, especially in the northern hemisphere.

The resulting emissions can be considered as a partly aged background air, implying a possible regional background and/or a long-range (intercontinental) transport.

The average contribution of this mixed source (combining both natural gas and background emissions) is in the range of 9.2 μg m$^{-3}$ during the whole studied period. The highest contributions occur in spring (13.3 μg m$^{-3}$) when the Paris region is mostly influenced by prevailing winds originating from the north and the northeast parts of Europe passing over Germany and

the Benelux area (see Fig. 5d, 5e and 5f). These continental imports constitute background events, which significantly impact baseline levels of ethane and oxygenated species. Slightly lower reconstructed mass contributions of this factor F6 were also observed in autumn. This fact can be explained by the consumption of natural gas (for home heating) during this season as average temperatures are progressively going down. No significant continental influences occur during the fall period as main air masses were coming from the west, south and southeast sectors, thus illustrating the importance of local pollution emissions

during this season.

### 3.5 VOC source contributions

PMF simulations revealed the significant contribution of six VOC emission sources (e.g. five specific factor profiles and a mixed one, for which the natural gas source could not be isolated from background emissions). This Source Apportionment (SA) analysis concluded that the predominant sources at the receptor site were road-traffic-related activities (including both

motor vehicle exhaust, 15 % of the Total VOC (TVOC) mass on the annual average, and gasoline evaporation, 10 %), with the remaining emissions from natural gas and background (23 %), solvents use (20 %), wood-burning (17 %) and biogenic sources (15 %). Each modeled factor exhibits distinct patterns due to the variations of the different source emissions and meteorological conditions. Monthly averaged contributions (expressed in %) of these factors to TVOC mass and all sources are reported in Fig. 10. Seasonal variations of the individual sources have already been commented in the previous sections. Therefore, only

the most important features are reported here.

Road-traffic emissions were identified by PMF simulations to be the main source of VOCs in Paris. The sum of motor vehicle exhaust and gasoline evaporation sources accounted for a quarter of the TVOC mass. It showed higher contributions at the end of the year (21 % and 15 %, respectively), that is still consistent with the study from Bressi et al., 2014 and with long-term black carbon measurements (Petit et al., 2015) linked to enhanced traffic during autumn in Paris. Most importantly,

it was observed that the wood burning source exhibited a significant contribution in winter months (almost 50 % in January and February), which is still in agreement with wood-burning related-particle emissions (Favez et al., 2009). The biogenic source also displayed a significant contribution (~30 %) in summer (mainly due to the weight of oxygenated species in the factor profile). The solvent use source displayed high contributions during winter months (~33 %, due to a lower PBL height and slower photochemical reactions during that period) and in July (due to the evaporation of solvents controlled by temperature).





The source mixing natural gas and background showed a higher proportion in springtime (~34 %) and lower proportions during autumn (~25 %). This conclusion can be explained by pollution events that are both related to air masses imported from continental Europe (see Fig. 5d, 5e and 5f) and/or specific meteorological conditions (low temperatures involving the use of home heating), respectively.

## 4 Discussions

### 4.1 Comparison with previous Source Apportionment (SA) studies performed in Paris

Based on 1-year daily $PM_{2.5}$ measurements (September 2009 – September 2010), Bressi et al. (2014) also conducted a Source Apportionment (SA) analysis using the PMF method (EPA PMF 3.0) with the aim of identifying and characterizing major fine aerosols emission sources within the Paris area. Seven factors, namely Ammonium Sulfate (A.S.)-rich factor, Ammo-
nium Nitrate (A.N.)-rich factor, heavy oil combustion, road-traffic, biomass burning, marine aerosols and metal industry were identified. Special attention is paid here to common modeled factor categories.

Primarily of local origin, the road-traffic source (resulting from exhaust and non-exhaust processes) constitutes approximately 14 % of $PM_{2.5}$ mass (~2.1 μg m$^{-3}$, on average) over the whole sampling period. Its annual contribution was considered as significant but surprisingly low given the high traffic density in Paris and its surroundings. It exhibits stable averaged
contributions throughout the year, with a smaller proportion in winter (6 %, 1.3 μg m$^{-3}$) and higher in autumn (19 %, 2.5 μg m$^{-3}$). This temporal source variation is still in agreement with the seasonal cycle of the road-traffic source (combining motor vehicle exhaust and evaporative running losses) issued from our VOC PMF analysis (see Sub-sections 3.4.1 and 3.4.2). The second common wood burning source is estimated for the first time over long periods and contributes to around 12 % (1.8 μg m$^{-3}$) of the total $PM_{2.5}$ mass. As expected, higher contributions were significantly observed during winter (22 %, 4.7 μg
m$^{-3}$) and in autumn (18 %, 2.4 μg m$^{-3}$ ). This finding is still consistent with the seasonal pattern of the wood burning VOC source. Because of the daily time resolution of filter sampling, no diurnal variation of modeled sources was reported in Bressi et al.(2014), thus limiting any additional comparison with this study.

Based on 1-month VOC measurements (25 May – 14 June 2007) performed at the LHVP site, Gaimoz et al. (2011) also
conducted a PMF analysis with the aim of identifying and apportioning major VOC sources in Paris. Seven factors, namely vehicle exhaust, fuel evaporation, remote industrial sources, natural gas + background, local sources, biogenic and fuel evaporation and wood burning, were found. For an appropriate comparison between this study and our work, special attention is paid here to the modeled speciation profiles and source contributions.

Chemical profiles from Gaimoz et al. (2011) revealed consistent findings with this study. The fuel evaporation factor is
mainly composed of butanes, propane and ethane whereas the vehicle exhaust factor includes *iso*-pentane, benzene, toluene, $C_8$ and $C_9$-aromatics and in lower proportions ethylene, propene and acetylene. These observations are consistent with modeled gasoline evaporation and motor vehicle exhaust profiles obtained in this work. A biogenic and fuel evaporation source is also identified and essentially made of isoprene, methanol, acetone and a high proportion of *iso*-pentane, suggesting that this factor



is mixing up biotic emissions and road-traffic activities. Highly dependent on (continental) air-mass origins, a remote industrial factor (related to industrial activities and long-range transport of secondary VOCs) is found to exhibit high contributions of OVOCs (methanol, acetone), aromatic species (toluene, $C_8$-$C_9$ aromatics) and some light alkanes. Our PMF study emphasized a solvents use source, for which these aforementioned compounds were observed, in addition to benzene. The wood burning

source includes only a high contribution of acetonitrile although ethylene, acetylene and benzene are significantly emitted - in accordance with findings from the fireplace experiment (see Sub-section 2.3.2). The mixed natural gas + background source is especially driven by ethane, methanol and acetone. No aromatic species appear in this factor profile. Finally, the local source (LPG – Liquefied Petroleum Gas) including propane and pentanes seems to be associated with fuel evaporation sources. This kind of factor was not identified in our study.

During May – June 2007, Gaimoz et al. (2011) concluded that the major VOC sources were related to road-traffic emissions (~39 % of the TVOC mass), with the remaining emissions from wood burning (2 %), biogenic and fuel evaporation (5 %), remote industrial sources (35 %), natural gas and background (13 %) and local sources (7 %) during the whole studied period (Fig. 11, left pie chart). To accurately compare VOC sources proportions between 2007 and 2010 (for a similar combinaison of hydrocarbons and masses), the contribution of each main factor was recalculated for the specific time period May – June 2010

(Fig. 11, right pie chart).

Significant differences between biogenic and wood burning sources contributions could eventually be both explained by the weight of major OVOCs into speciation profiles (Relative proportions of methanol, acetaldehyde, acetone in these factors are higher than those of the comparative study) and the differences in temperatures affecting the Paris region. The mean temperature recorded during May – June 2007 was in the range of ± 20°C whereas it was estimated at ± 16°C in 2010. This would explain a

higher home heating consumption and consequently, a higher contribution of the wood burning factor in 2010 (9 % *versus* 2 % for the previous work). Regarding to solvents use source contributions, differences can also be explained by temperatures (as they constitute a relevant indicator in solvent emissions) and by the amount of solvents used in manufactories due to recent regulatory frameworks in place (20 % in 2010 *versus* 35 % in 2007). The difference of natural gas and background source contributions can be due to the importance of air masses coming from the north and northeast parts of Europe between 25

May and 14 June 2010. These air-masse origins were also observed in 2007 and could have affected remote industrial-related emissions and not the mixed source. Slight differences of the motor vehicle exhaust source between 2007 and 2010 (22 % *versus* 14 %) could be explained by densification strategies and technological innovations for reducing car use and emissions. Finally, observed differences for the gasoline evaporation source (5 % for 2010 and 17 % in 2007) are related to emissions and high temperatures observed in 2007.

## 4.2   Comparison with some global SA studies

Yearly average contributions of the modeled VOC sources (see Section 3.5) were also compared with other Source Apportionment (SA) studies performed within urban areas in Europe and in the World. From the different European SA studies available, only one is based on a long VOC time series, which is strengthening the novelty and the originality of the current study.



Based on 2-years hourly measurements of $C_2$ - $C_7$ NMHCs, Lanz et al. (2008) permitted the identification and characterization of between eight and six emission sources at an urban background site in Zürich (Switzerland) in the years 1993-1994 and 2005-2006. Only measurements from 2005 to 2006 are compared here as they are the most recent observations we have available. Six factors, namely gasoline evaporation, solvents, propane, ethane, wood burning and fuel combustion were determined

using the PMF method. This SA study highlighted the importance of vehicular, solvents use, wood burning and gas leakage emissions. The road-traffic-related source included both gasoline evaporation and fuel combustion (motor exhaust) factors. While the first factor is mainly dominated by butanes (*iso-/n-*) and *iso*-pentane, the second one is essentially driven by ethane, ethene, propene, benzene and toluene. These two speciation profiles are still consistent with those obtained from this PMF analysis, excepted for *iso*-pentane. Considered as a key species of evaporative processes, *iso*-pentane mostly contributed to the

motor vehicle exhaust source (Fig. 8a). It was also identified as one of the main compounds emitted in the highway tunnel experiment (Fig. 2), where measured hydrocarbons were representative of fresh emissions (e.g. fuel combustion). This modeled road-traffic source contributed to 26 % of the TVOC mass (13 % for gasoline evaporation and 13 % for fuel combustion factors), which is in the same order of magnitude than that of our vehicle-related source (25 %). The solvents use factor is characterized by pentanes, S-isohexanes (including 2-methylpentane, 3-methylpentane, 2,2-dimethylbutane, 2,3-dimethylbutane)

and toluene, in agreement with our solvents use VOC profile which also included oxygenated species (not measured in Lanz et al. (2008)). This industrial factor accounted for 20 % of the TVOC mass. This source contribution is comparable to what we obtained from January to November 2010 (20 %). The wood burning factor is mainly dominated by ethylene, acetylene, ethane, benzene and contributed to 16 % of the TVOC mass for the 2005-2006 sampling period. This finding is fairly in agreement with our annual wood burning contribution. Finally, a natural gas source was also identified and consisted of the combination

of two separated factors ("ethane" with "propane"). Its annual contribution is evaluated at 35 % of the TVOC mass whereas our mixed natural gas and background source accounted for 23 %. No biogenic source was detected for this comparative study. To sum up, average contributions of the road-traffic, solvents use and wood burning sources well matched between this SA study and our modeled results although the input chemical matrix and sampling dates are different.

The importance of these three anthropogenic sources was often reported in other existing urban SA studies from short-term

measurements performed in Europe. For instance, Niedojadlo et al. (2007) (Wuppertal, Germany) paid particular attention to solvents use and road-traffic sources contributions using the Chemical Mass Balance (CMB) modeling technique. Main results showed that the road-traffic source dominated total VOC emissions (more than 50 % of the total mass) compared to solvents use. In addition, it was considered that the proportion of solvents emissions to TVOC concentrations fell in the range of ~20 % in German cities, which is significantly consistent with Lanz et al. (2008) and with this SA study in Paris.

The consistency in VOC source contributions in European urban areas raises the question of their representativity at a larger scale. Special attention is paid to two SA examples performed in Asia and in Central America. Based on VOC measurements from 2007 to 2010, Cai et al. (2010) conducted source apportionment simulations (using the PMF model) in order to identify the major VOC sources in downtown Shanghai. Key findings revealed seven emission factors, including vehicle-related source,

solvent based source, fuel evaporation, paint solvent usage, steel related industrial production, biomass/biofuel burning and



coal burning. Among these modeled sources, six of them can be compared to those obtained from our PMF analysis. The traffic source (e.g. vehicle and fuel evaporation factors) constituted approximately 40 % of the TVOC mass over the whole sampling period. Another VOC source, commonly named solvents, combined "solvent", "paint solvent usage" and "steel-related industrial production" factors. It contributed to ~47 % of TVOC concentrations and thus became the major VOC source

in Shanghai. Finally, the biomass/biofuel burning source only accounted for 9 % of the total mass. Significant differences in source contributions can be partly explained by the input chemical matrix, sampling dates or the number of PMF factors considered. No seasonal and diurnal variations of modeled sources were reported in Cai et al. (2010), thus limiting any additional comparison with this study. Based on 1-month measurements of NMHCs and OVOCs (May-June 2010), Zheng et al. (2013) conducted a SA analysis in order to apportion ambient air concentrations into main emission sources in Mexico. PMF modeled

factors included industrial solvent usage, gasoline vehicle exhaust, diesel vehicle exhaust and aged plume (related to background and/or long-range transport). Main modeled results revealed the importance of the industrial source (58 % of the TVOC mass), followed by vehicular emissions (33 %) and background influences (9 %).

Finally, this comparison highlighted that road-traffic conditions and industrial activities had significant impacts on the TVOC

concentrations in urban areas, whether in Europe or in the rest of the world. Among all studied SA analyses, the road-traffic source plays a dominant role (between 30 % and 50 %) in TVOC emissions. Similarities in VOC source contributions (especially for traffic, wood burning and solvent sources) were observed for European SA studies, thus illustrating the consistency of this PMF analysis for the Paris megacity. However, slight differences in VOC source contributions were found between our SA study and global SA examples (Shanghai, Mexico) for which the industrial source exhibited largest contributions ranging

from 50 % to 60 %. They can be partially explained by different governmental rules and standards between countries to control pollutants emissions. The location of sampling points (or distances from main sources) and meteorological conditions can strongly affect VOC concentrations and their respective emission sources in the considered urban environments.

## 5   Conclusions

Within the framework of the EU-F7 MEGAPOLI and PRIMEQUAL-FRANCIPOL research programs, a selection of volatile

organic compounds (VOCs) were continuously measured in real-time at two background urban sites located in downtown Paris (France) from 15 January to 22 November 2010. Assessed hydrocarbons included alkanes, alkenes-alkynes, isoprene, aromatics and oxygenated compounds (OVOCs). The current study allowed evaluating VOC concentration levels in ambient air and describing their temporal (seasonal and diurnal) time courses over a long period of time in the French megacity. It also showed an innovative methodology to identify, quantify and understand the main VOC emission sources in Paris by combining

field experiments (near-field and ambient air measurements) with *source-receptor* statistical modeling. The modeled factor profiles were interpreted with respect to those obtained from literature and from three near-field experiments (inside a highway tunnel, at a fireplace and from a domestic gas flue) performed within the Paris area. These additional measurements helped better characterizing and/or confirming traffic, wood burning and natural gas-related sources among the existing different



source profiles, which can be directly derived from a PMF modeling analysis. These source profile studies therefore allowed to check the representativity and the robustness of our conclusions. This PMF analysis successfully reconstructed at least $88 \pm 2\%$ of the measured total VOC mass.

Among the six identified PMF factors, road-traffic activities appeared to be the main VOC source in Paris with an average
contribution of 25 % of the TVOC mass at the annual scale. This source both included motor vehicle exhaust (15 %) and gasoline evaporation (10 %). For the first time, it was also shown that the residential wood burning source exhibited an important contribution in winter (almost 50 %) due to cold temperatures during that season (home heating consumption). The biogenic source also displayed a significant contribution (~30 %) in summer mainly due to the weight of oxygenated species in the factor profile. A solvents source was identified and annually contributed to 20 % of the total VOC mass. Finally, it was also revealed
that a source mixing natural gas and background (23 %) could be highly dependent on air-mass origins (especially during continental-influenced periods) and meteorological conditions (temperatures). It exhibited a higher proportion in springtime (34 %, explained by intercontinental imports) and in autumn (25 %, partly for home heating consumption reasons).

From this initial source apportionment study, natural gas could not be isolated from background emissions by the PMF method, thus leading to a limitation of this analysis. A further work will aim at constraining the reference speciation profile
(obtained from domestic gas flue measurements) in order to evaluate the relative contribution of natural gas emissions. Lastly, the quantitative assessment of the contributions from different modeled sources presented in this study will provide an independent evaluation of the quality and the relevance of the corresponding emission inventories. In particular, the comparison will be very valuable with the updated local emission inventory (provided by the regional air quality network AIRPARIF) as some discrepancies had been pointed out with its previous version.

*Acknowledgements.* Authors would like to thank B. Temime-Roussel and N. Marchand from the Laboratoire Chimie Provence (LCP, University of Provence, Marseille, France) for PTR-MS measurements performed at the LHVP site during the MEGAPOLI winter campaign. We would like to thank also M. Beekmann for the coordination of the EU-F7 MEGAPOLI project. We gratefully acknowledge MM. Squinazzi and Y. Le Moullec for having hosted MEGAPOLI and FRANCIPOL intensive campaigns as well as all colleagues involved in the monitoring process of ambient air measurements, especially Hanitriniala Ravelomanantsoa, Thomas Chaigneau (LHVP) and Laurent Martinon
(Laboratoire d'Etude des Particules Inhalées, LEPI). T. Le Priol and J.-F. Petit from the Centre d'Etudes et d'expertise sur les Risques, l'Environnement, la Mobilité et l'Aménagement (CEREMA) are also acknowledged for the logistical assistance in the road-tunnel experiment. This work was supported by the CEA, CNRS, IPSL, ADEME, Île-de-France region funds, the EU-PF7 ANR MEGAPOLI and the French PRIMEQUAL-FRANCIPOL, PREQUALIF and CORTEA-CHAMPROBOIS projects. We are thankful to Sabina Assan for helping with the English version of the manuscript.



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



**Table 1.** Statistics ($\mu g\ m^{-3}$) of selected VOC concentrations measured at urban background sites, Paris (15 Jan. - 22 Nov. 2010).

| | Species | Minimum | 25th Percentile | Median | Mean | 75th Percentile | Maximum | $\sigma$ |
|---|---|---|---|---|---|---|---|---|
| | Ethane | 0.83 | 3.07 | 4.14 | 4.56 | 5.42 | 26.31 | 2.26 |
| | Propane | 0.23 | 1.63 | 2.44 | 2.78 | 3.45 | 25.64 | 1.80 |
| | *Iso*-butane | 0.23 | 1.12 | 1.58 | 1.96 | 2.30 | 23.52 | 1.51 |
| ALKANES | *N*-butane | 0.40 | 1.88 | 2.69 | 3.35 | 3.93 | 56.10 | 2.88 |
| | *Iso*-pentane | 0.25 | 1.25 | 1.82 | 2.24 | 2.68 | 25.81 | 1.65 |
| | *N*-pentane | 0.10 | 0.58 | 0.85 | 1.04 | 1.28 | 12.04 | 0.76 |
| | *N*-hexane | 0.06 | 0.27 | 0.40 | 0.49 | 0.59 | 4.25 | 0.34 |
| ALKENES | Ethylene | 0.04 | 0.81 | 1.25 | 1.55 | 1.92 | 14.04 | 1.14 |
| | Propene | 0.09 | 0.37 | 0.53 | 0.64 | 0.78 | 5.93 | 0.44 |
| ALKYNE | Acetylene | 0.04 | 0.29 | 0.48 | 0.68 | 0.81 | 7.39 | 0.64 |
| DIENE | Isoprene | 0.08 | 0.13 | 0.19 | 0.26 | 0.31 | 1.74 | 0.22 |
| | Benzene | 0.04 | 0.62 | 0.89 | 1.05 | 1.26 | 7.60 | 0.66 |
| AROMATICS | Toluene | 0.12 | 1.79 | 2.46 | 3.29 | 3.68 | 34.56 | 2.86 |
| | Xylenes + $C_8$ | 0.26 | 1.58 | 2.19 | 2.76 | 3.25 | 21.84 | 1.97 |
| ALCOHOL | Methanol | 0.86 | 3.66 | 4.83 | 5.89 | 6.83 | 39.29 | 3.89 |
| NITRILE | Acetonitrile | 0.10 | 0.30 | 0.40 | 0.71 | 0.67 | 31.87 | 1.09 |
| ALDEHYDE | Acetaldehyde | 0.54 | 2.02 | 2.71 | 3.17 | 3.71 | 15.04 | 1.83 |
| KETONE | Acetone | 0.73 | 3.05 | 4.33 | 4.87 | 5.79 | 22.24 | 2.64 |
| ENONE | MVK + MACR + ISOPOOHs | 0.05 | 0.30 | 0.48 | 0.65 | 0.77 | 6.27 | 0.59 |



**Table 2.** Comparison of mean concentrations of selected VOCs (measured by PTR-MS) with ambient levels observed in the literature from different urban atmospheres. All average values are reported in μg m$^{-3}$.

| VOCs measured by PTR-MS (m/z) | Paris[a] (P.) Jan.-Nov. (Spring) 2010 | Paris[b] Spring 2007 | Barcelona[c,1] Winter 2009 | London[d] (P.) October 2006 (2010) | Mohali[e] (P.) May 2012 (2010) | Mexico City[f] (P.) March 2006 (2010) | Beijing[g] (P.) August 2005 (2010) | Houston[h] (P.) Aug.-Sept. 2000 (2010) |
|---|---|---|---|---|---|---|---|---|
| Methanol (33.0) | 5.9 (8.6) | 7.8 | NA | NA (4.4) | 50 (6.9) | NA (2.1) | 15.6 (3.7) | 14.4 (5.1) |
| Acetonitrile (42.0) | 0.7 (1.2) | 0.7 | 0.4–0.8 | 0.6 (0.4) | 2.4 (0.9) | 0.4–2.4 (0.3) | NA (0.5) | 0.9 (0.9) |
| Acetaldehyde (45.0) | 3.2 (3.4) | 2.5 | 1.5–3.1 | 6.5 (2.8) | 12.3 (3.0) | 5.5–22.0 (2.0) | 6.6 (2.0) | 6.2 (2.7) |
| Acetone (59.0) | 4.9 (5.9) | 7.1 | 2.7–4.0 | 3.8 (5.1) | 14.3 (5.0) | NA (2.4) | 10.7 (3.8) | 9.7 (4.9) |
| MVK + MACR + ISOPOOHs (71.0) | 0.7 (0.7) | 1.0 | NA | NA (0.5) | NA (0.4) | NA (0.2) | 2.5 (0.6) | 2.3 (0.7) |
| Benzene (79.0) | 1.1 (0.9) | 1.2 | 0.7–1.9 | 0.5 (1.2) | 5.5 (0.9) | NA (0.9) | NA (0.6) | 2.0 (0.9) |
| Toluene (93.0) | 3.3 (3.4) | 5.2 | 3.1–10.4 | 7.1 (3.6) | 10.4 (2.9) | 11.5–107.3 (2.1) | 3.8–15.3 (2.0) | 3.1 (3.3) |
| Xylenes + C$_8$ (107.0) | 2.8 (2.6) | 5.5 | 3.8–15 | 0.7 (3.1) | 8.8 (2.5) | NA (1.8) | NA (1.8) | 2.7 (2.6) |

[a] This study (*Values in brackets from VOC measurements performed during the same sampling period of the other urban studies are given for comparison*).

[b] Gros et al. (2011)

[c] Seco et al. (2013)

[d] Langford et al. (2010)

[e] Sinha et al. (2014)

[f] Fortner et al. (2009)

[g] Shao et al. (2009)

[h] Karl et al. (2003)

[1] A full comparison was not possible because no data was available between 16 February and 24 March 2010.

NA – Non Available data.



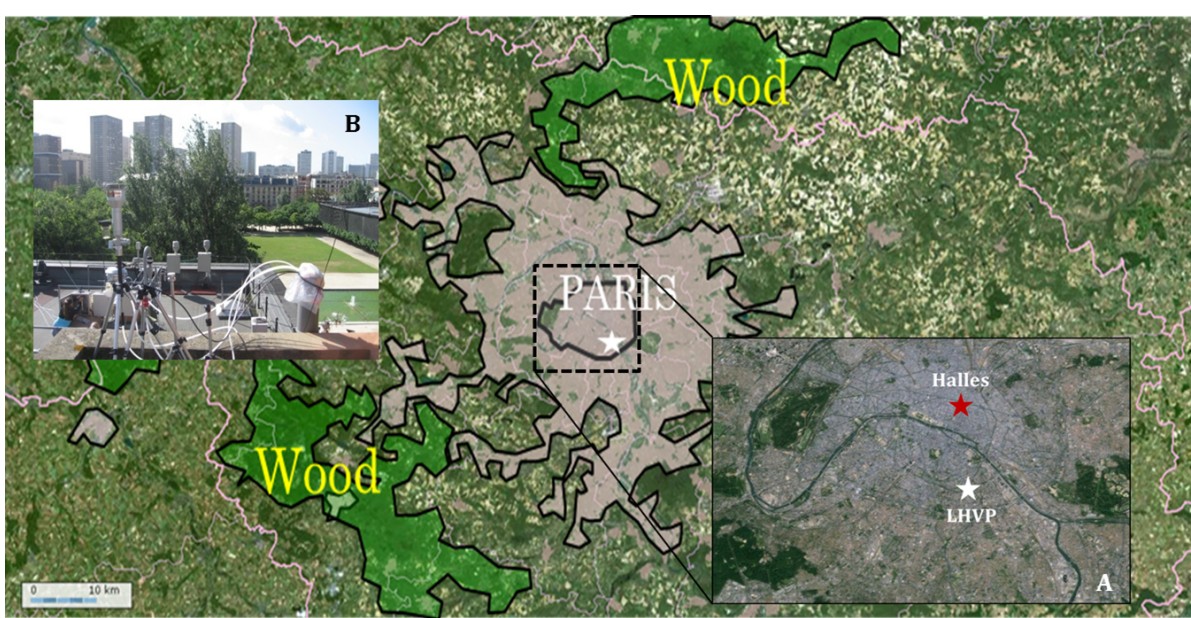

**Figure 1.** Maps of Paris and Île-de-France region. The picture A shows the location of the two main sampling sites in downtown Paris. The white and red stars locate the position of the LHVP laboratory and the AIRPARIF site, respectively. The picture B shows the terrace roof of LHVP.




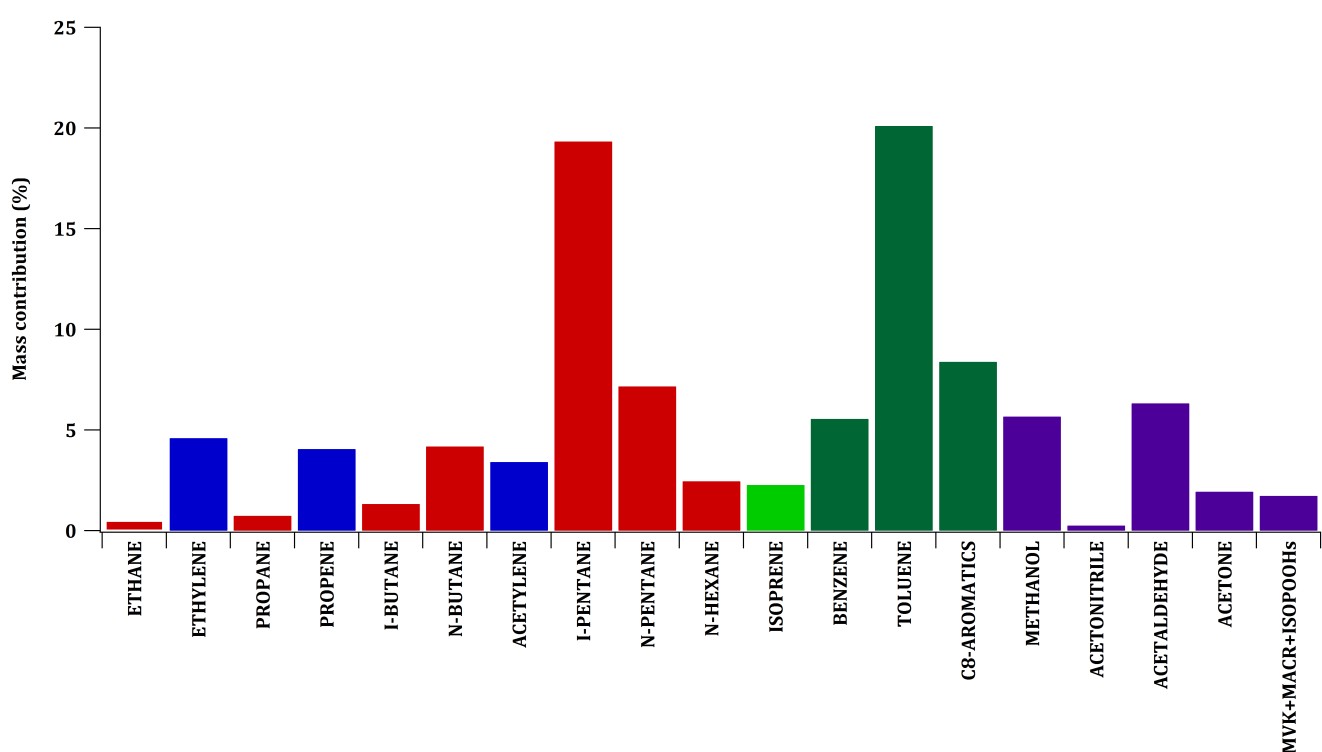

**Figure 2.** Average highway tunnel profile (in mass contribution, %) assessed from traffic peaks concentrations and subtracted from nighttime values. Red, blue, light/dark green and purple bars correspond to alkanes, alkenes-alkynes, isoprene/aromatics and oxygenated species, respectively.



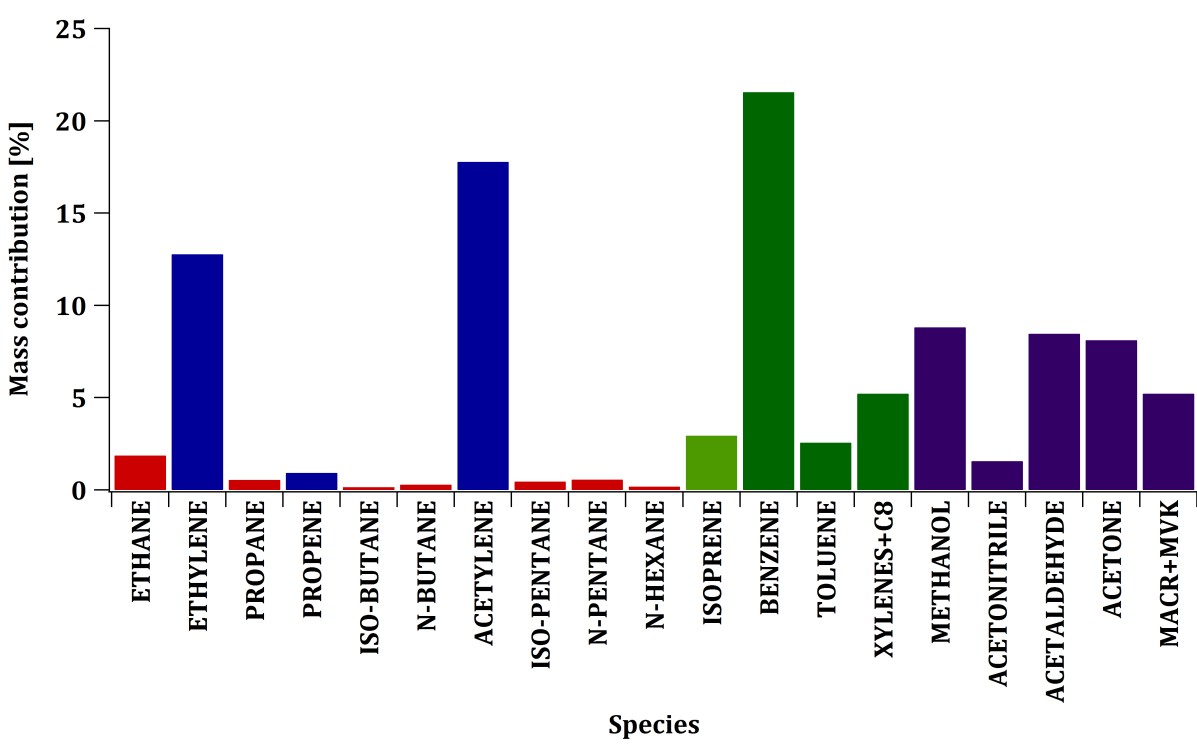

**Figure 3.** Average VOC fingerprint (in mass contribution, %) from domestic biomass burning obtained during the fireplace experiment. Red, blue, light/dark green and purple bars correspond to alkanes, alkenes-alkynes, isoprene/aromatics and oxygenated species, respectively.





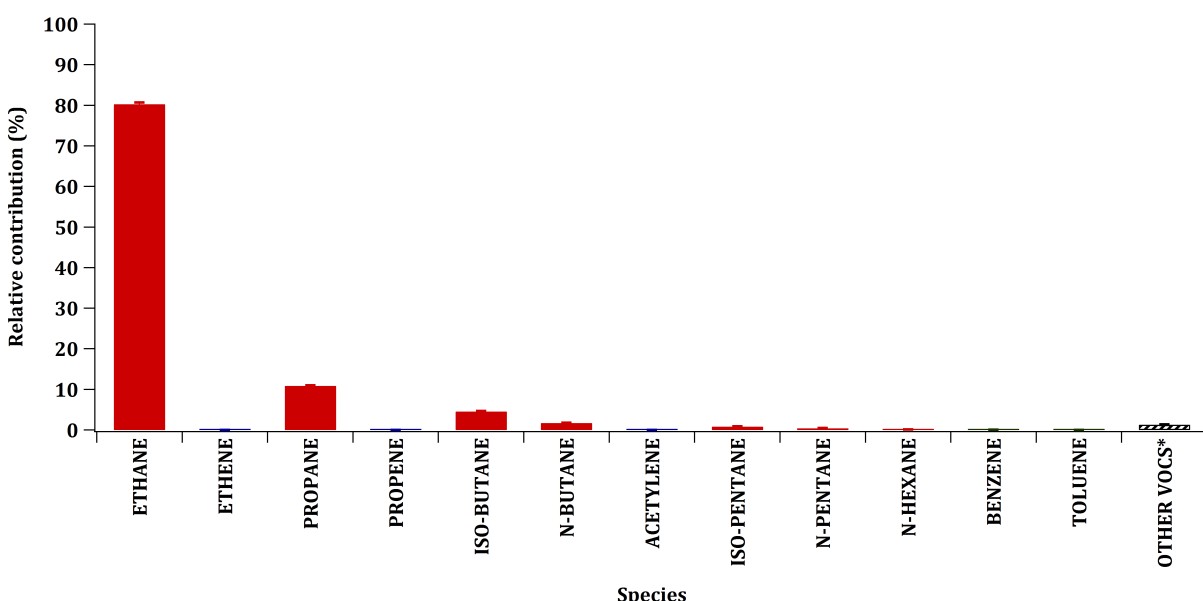

**Figure 4.** Average chemical composition of natural gas used in Paris. * Other VOCs include heavier alkanes (e.g. cyclopentane/hexane, dimethyl butanes) and butenes in lower proportions. Whiskers correspond to error bars ($1\sigma$).









**Figure 5.** Average wind roses from January (a) to November (k) 2010. Headings 0 (or 360), 45, 90, 135, 180, 225, 270 and 315 expressed as degrees (°) correspond to main cardinal directions and their intermediate points (N, NE, E, SE, S, SW, W, NW, respectively). The radial axis represents the wind occurrence (in %).



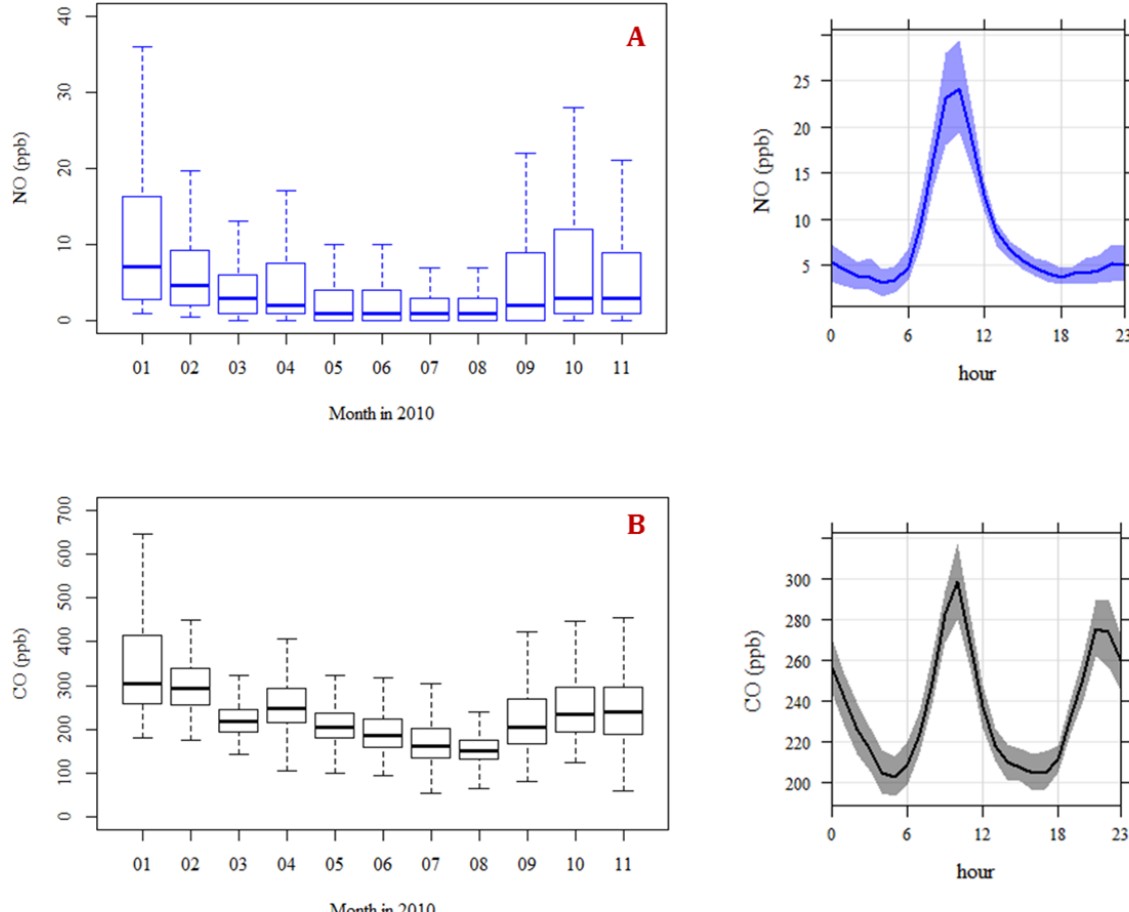

**Figure 6.** (Left) Monthly box and whisker plots of NO (A) and CO (B), expressed as ppb. Solid lines represent the median concentration and the box shows the InterQuartile Range (IQR). The bottom and top of the box depict the 25th (the first quartile) and the 75th (the third quartile) percentile. The ends of the whiskers correspond to the lowest and highest data still within 1.5 times the IQR of Q1 and Q3, respectively. (Right) Diurnal variations of NO and CO averaged over the whole sampling period. Time is given as Local Time. Lines correspond to hourly means and shaded areas indicate the 95 % confidence intervals of the mean.









**Figure 7.** (Left) Monthly box and whisker plots of benzene (A), toluene (B), methanol (C), acetaldehyde (D), acetone (E) and MVK + MACR + ISOPOOHs (F) expressed as $\mu g\ m^{-3}$. Solid lines represent the median concentration and the box shows the InterQuartile Range (IQR). The bottom and top of the box depict the 25th (the first quartile) and the 75th (the third quartile) percentile. The ends of the whiskers correspond to the lowest and highest data still within 1.5 times the IQR of Q1 and Q3, respectively. (Right) Diurnal variations of (O)VOCs averaged over the whole sampling period. Time is given as Local Time. Lines correspond to hourly means and shaded areas indicate the 95 % confidence intervals of the mean.





**Figure 8.** Source composition profiles of the 6-factor PMF solution. The concentrations (μg m$^{-3}$, logarithm scale) and the percent of each species apportioned to the factor are displayed as a pale blue bar and a color box, respectively. **(a)** F1 – Motor Vehicle Exhaust ; **(b)** F2 – Gasoline Evaporation ; **(c)** F3 – Wood Burning; **(d)** F4 – Biogenic ; **(e)** F5 – Solvents use ; **(f)** F6 – Natural Gas and Background.







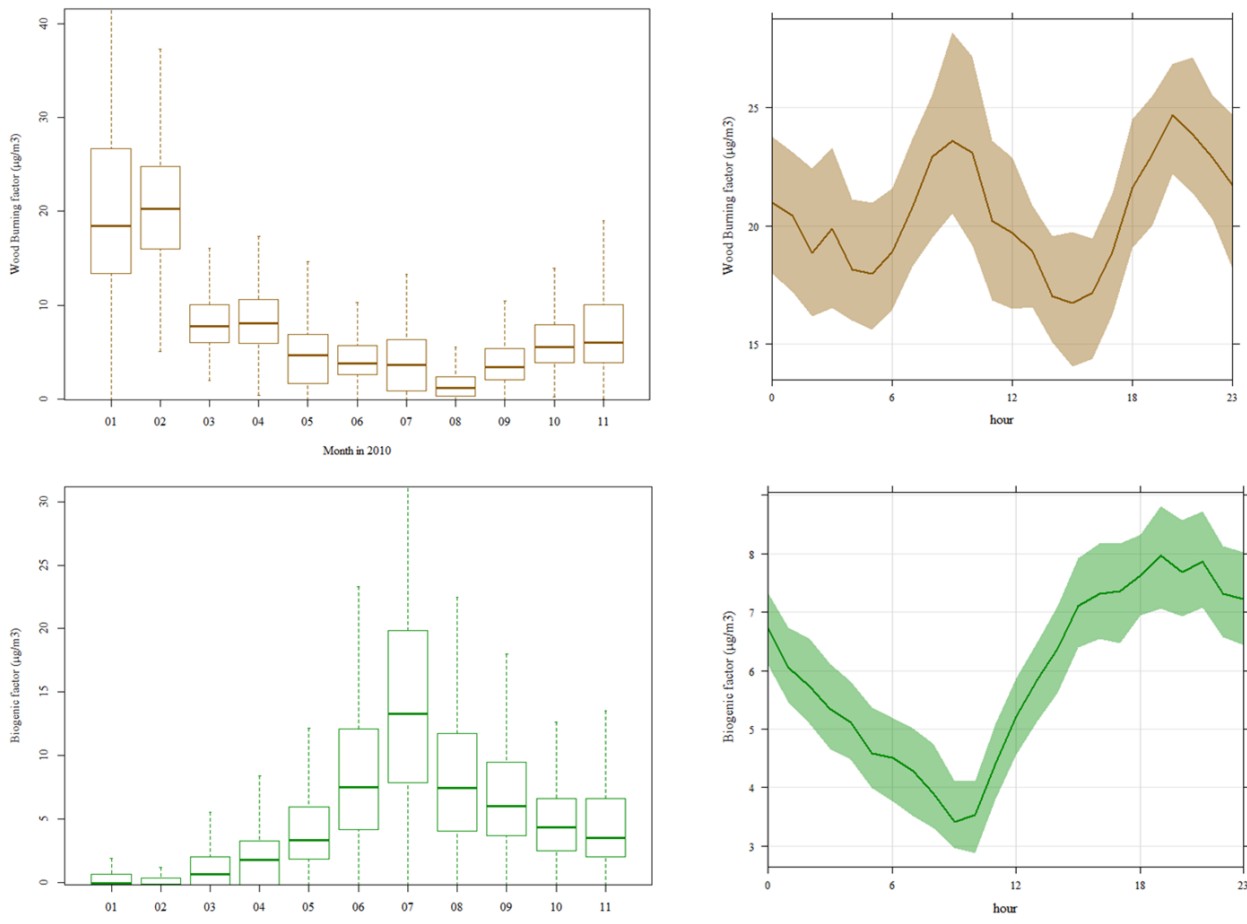





**Figure 9.** (Left) Monthly box and whisker plots of factors from the 6-factor solution. Concentration levels are expressed in µg m$^{-3}$. Solid lines represent the median concentrations and the box shows the InterQuartile Range (IQR). The bottom and top of the box depict the 25th (the first quartile) and the 75th (the third quartile) percentile. The ends of the whiskers correspond to the lowest and highest data still within 1.5 times the IQR of Q1 and Q3, respectively. (Right) Diurnal variations of the resolved PMF factors. Time is given in Local Time. Lines correspond to hourly means and shaded areas indicate the 95 % confidence intervals of the mean.





**Figure 10.** Variations of monthly averaged contributions of the six modeled VOC sources (expressed in %); (Top) - Average predicted VOC concentration levels per month ($\mu$g m$^{-3}$); (Bottom) - Completeness of the data per month ( %).





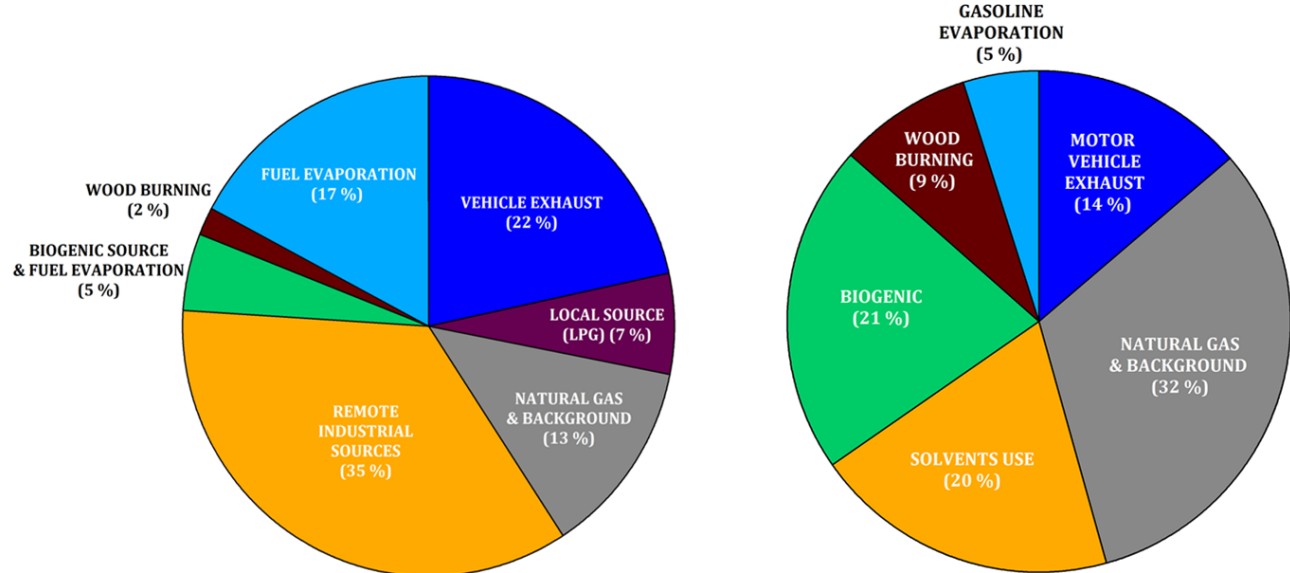

**Figure 11.** Relative contributions to TVOC mass of 7 and 6 PMF sources identified from 25 May to 14 June 2007 (Gaimoz et al., 2011) and 2010 (this study), respectively.





## Appendix A: Application of the Positive Matrix Factorization (PMF) approach in source apportionment of VOCs in Paris.

### A1    Data preparation

Initially, the EPA PMF 5.0 model requires two input datasets: one with the chemical species atmospheric concentrations for
each observation point and another with either uncertainties values or parameters for calculating the associated uncertainty.

The initial chemical dataset contains a selection of 19 hydrocarbon species and masses (for a detailed overview, see list of compounds in the Sub-section 2.3.1) measured from 15 January to 22 November 2010. Non methane hydrocarbons (NMHCs) and aromatic/oxygenated species (OVOCs) were respectively monitored with GC-FID and PTR-MS instruments belonging to
different partners involved during the MEGAPOLI and FRANCIPOL intensive field campaigns. Unfortunately, no intercomparison between these instruments was possible because there was approximately a one-month delay between both experiments. However, preliminary PMF modeling simulations were performed using only the FRANCIPOL dataset (24 March – 22 November). The results have shown similar source profiles, as those already described in this paper. Consequently, two datasets (corresponding to MEGAPOLI and FRANCIPOL ones, respectively) were considered to form a single one and use it as an
input unified database for the final PMF analysis.

The uncertainty dataset was built upon the equation-based method described by Norris et al. (2014). It requires both Method Detection Limit (MDL, here in µg m$^{-3}$) and the analytical uncertainty ($u$, here in %) for each considered species. Two sets of MDLs were used, one for each measurement campaign. Slight differences among species MDLs were found for $n$-
hexane, aromatics, acetaldehyde and MVK+MACR+ISOPOOHs between both experiments. Of these VOCs, MDLs from the FRANCIPOL campaign were chosen for representativity reasons (as the corresponding dataset represents ~ 88 % of the total data matrix) to keep consistency in uncertainty calculations. The analytical uncertainties were respectively estimated at 15 % and 20 % and kept constant over the experiments.

The PMF uncertainty ($\sigma$) is therefore calculated as follows:

$$If\ X_{ij} \leq MDL,\ \forall j;\ X_{ij} = \frac{MDL}{2}\ and\ \sigma_{ij} = \frac{5}{6} \times MDL \tag{A1}$$

$$If\ X_{ij} \geq MDL,\ \forall j;\ X_{ij}\ does\ not\ change\ and\ \sigma_{ij} = \sqrt{(ErrorFraction\ u\ \times\ X_{ij})^2 + (0.5 \times MDL)^2} \tag{A2}$$

MDLs and analytical uncertainties ($u$) for each VOC are reported in Table A1.



## A2 Estimation of the number of PMF factors ($p$)

The accurate number of PMF factors ($p$ values) in models must be ultimately estimated by the user using several exploratory means. Specific parameters were used to determine the appropriate $p$ value such as the assessment of $Q$ values, scaled residuals, predicted *versus* observed concentrations interpretation and the physical meaning of factor profiles.

    Eight different modeling conditions were examined with $p$ values ranging from 3 to 10, each simulation being randomly conducted 20 times. The reviewing of the IS (the maximum individual standard deviation) parameter highlighted a slope failure for $p = 5$ whereas the IM (the maximum individual column mean) indicator reported another slope failure for $p = 6$. Choosing less factors, $p < 6$, concatenated three source profiles (attributed to solvents use, natural gas and background emissions, respectively) into a factor, whereas choosing $p = 6$ allowed splitting one of them. Opting for $p > 6$ did not provide any supplemental physical meaningfulness to existing profiles. The investigation of $r^2$ from ten modeled solutions also reported a slope failure for $p = 6$. In addition, only *Qtrue* / *Qexpected* value for $p = 6$ was closer than 1.0 (e.g. 0.94 in comparison with 1.12 for $p = 5$ and 0.8 for $p = 7$), thus suggesting that the 6-factor configuration is supposed to be the most optimum solution for the current PMF analysis. Finally, this configuration was investigated over all the details. Usually, PMF identifies the best solution by the lowest *Qrobust* value (e.g. the mimimum Q). Within this analysis, its corresponding PMF solution was not considered due to a lack of physical significance for one factor profile (e.g. solvents). Therefore, another PMF solution closest to the selected *Qrobust* value was subsequently examined and chosen in terms of interpretability and fitting scores.

## A3 Robustness of PMF results

Further technical and mathematical indicators regarding to the 6-factor configuration are reported here to assess the robustness and the quality of the final PMF solution. Firstly, the ratio between *Qrobust* and *Qtrue* reached around 1.0, thus indicating that the modeled results were not biased by peak events. Almost 100 % of the scaled residuals were within $\pm 3\sigma$ and were normally distributed for all species. In addition, the Kolmogorov-Smirnoff (KS) test granted a KS $p$-value very close to zero, thus illustrating a statistically significant test with a $\alpha$ risk of 5 %. The correlation between total VOC reconstructed concentrations from all the factors with total VOC observed concentrations is depicted in Fig. A1. With $R^2$ very close to 0.9, almost all variance in the total concentration of the 19 VOCs can be explained by the PMF model.

    Almost all the chemical species also displayed good determination coefficients ($r^2$ higher than 0.6 for 15 compounds) between predicted and observed concentrations, with the exception of propane and $n$-hexane showing a fairly reasonable coefficient between 0.5 and 0.6 (due to their 30.5 % and 35.7 % missing values, respectively). Isoprene and acetonitrile exhibited bad $r^2$ values (0.29 and 0.06, respectively) due to either a relatively high number of missing values or a "weak" additional error, for which sample uncertainties were tripled. Slopes were close to 1.0 for most species (higher than 0.6 for 17 VOCs), excepted for isoprene (0.5) and acetonitrile (0.02). The limitations of the PMF model to simulate isoprene and acetonitrile should therefore be kept in mind when discussing reconstructed results.



## A4 Estimation of model prediction uncertainties

PMF output uncertainties can be estimated using the Error Estimation options starting with DISP (d$Q$-controlled displacement of factor elements) and processing to BS (Classical bootstrap). These two uncertainty methods are designed to provide key information on the stability and the precision of the chosen PMF solution (Paatero et al., 2014).

The DISP, Base Model Displacement Error Estimation, assesses the rotational ambiguity of the PMF solution by exploring intervals (minimum and maximum) of source profile values. During the DISP, a minimum $Q$ value is newly calculated (based on the adjustment up and down in factor profile values) and compared with the unadjusted solution $Q$ value. The difference between the initial $Q$ value and the modified $Q$ value (so-called d$Q$) should be lower than d$Q$max value, for which four levels

(values: 4, 8, 15 and 25) were taken into account. For each d$Q$max value, 120 intervals were estimated. The DISP analysis results were considered validated: no error could be detected and no drop of $Q$ was observed. As no swap occurred, the PMF solution was considered sufficiently robust to be used.

The BS, Base Model Bootstrap Error Estimation, is also used to evaluate the reproducibility of the PMF solution, with a

specific focus on the original sub-matrix F. A further description on the bootstrapping technique is presented in Norris et al. (2014) and in Paatero et al. (2014). A base model bootstrap method was then carried out, executing one hundred iterations, using a random seed, a block size of 874 samples (calculated according to the methodology of Politis and White (2004)) and a minimum Pearson correlation coefficient (R-Value) of 0.6. All factors were well reproduced through this technique over at least 88 % of runs, thus indicating that BS uncertainties can be interpreted and the number of factors may be appropriate.

Consequently, 12 % of runs were redistributed into the different existing factors. No runs were unmapped. Finally, around 91 % of species with the base run profile value were identified within the InterQuartile Range (IQR, e.g. 25th - 75th percentile of bootstrap runs) for all factors considered.

Finally, the rotational ambiguity of this 6-factor PMF configuration was also investigated using the Fpeak parameter. Dif-

ferent Fpeak values from -5 to 5 were used to generate a more realistic PMF solution. The results from the nonzero Fpeak values were generally consistent with the runs associated with the zero Fpeak value (e.g. Base Model run), thus illustrating a low rotational ambiguity of the final PMF solution.



**Table A.1.** Method Detection Limits (MDLs) and Analytical Uncertainties ($u$) for each species used in PMF modeling simulations.

| Species | MDL–MEGAPOLI ($\mu g\ m^{-3}$) | MDL–FRANCIPOL ($\mu g\ m^{-3}$) | $u$ (%) |
|---|---|---|---|
| Ethane* | 0.025 | 0.024 | 15 |
| Ethylene* | 0.023 | 0.024 | 15 |
| Propane* | 0.037 | 0.024 | 15 |
| Propene* | 0.035 | 0.024 | 15 |
| *Iso*-butane* | 0.048 | 0.024 | 15 |
| *N*-butane* | 0.048 | 0.024 | 15 |
| Acetylene* | 0.022 | 0.024 | 15 |
| *Iso*-pentane* | 0.060 | 0.024 | 15 |
| *N*-pentane* | 0.060 | 0.024 | 15 |
| *N*-hexane* | 0.013 | 0.013 | 15 |
| Isoprene* | 0.024 | 0.024 | 20 |
| Benzene** | 0.071 | 0.071 | 20 |
| Toluene** | 0.259 | 0.240 | 20 |
| Xylenes + $C_8$** | 0.259 | 0.259 | 20 |
| Methanol** | 0.317 | 0.330 | 20 |
| Acetonitrile** | 0.068 | 0.084 | 20 |
| Acetaldehyde** | 0.167 | 0.167 | 20 |
| Acetone** | 0.092 | 0.018 | 20 |
| MVK[1] + MACR[2] + ISOPOOHs[3]** | 0.020 | 0.020 | 20 |
| $\Sigma$ VOC | 1.629 | 1.542 | 20 |

[1] MVK = Methylvinylketone

[2] MACR = Methacrolein

[3] ISOPOOHs = Isoprene hydroxy hydroperoxides (Rivera-Rios et al., 2014)

* Hydrocarbons measured using a GC-FID by LSCE (MEGAPOLI, LHVP) and AIRPARIF (FRANCIPOL, "Les Halles" subway station).

** Masses measured using a PTR-MS by LCP (MEGAPOLI, LHVP) and LSCE (FRANCIPOL, LHVP).



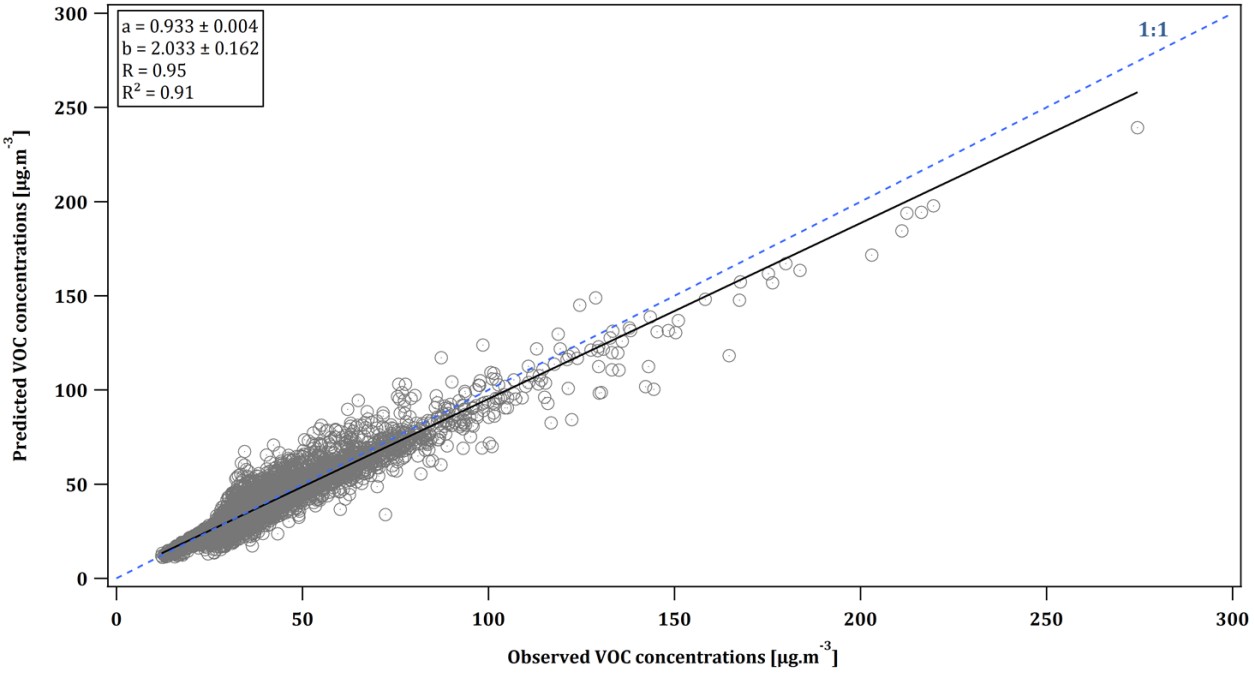

**Figure A.1.** Agreement between total predicted and observed VOC concentrations based on the 6-factor PMF solution.