# Peer review of "Seasonal variability and source apportionment of volatile organic compounds (VOCs) in the Paris megacity (France)"

_Atmospheric Chemistry and Physics, 2016_

## Referee Comment (RC1) · Anonymous Referee #1 · 6 May 2016

General comments

The presented manuscript presents data on NMHCs and OVOCs from two sampling sites in Paris covering about 10 months of quasi-continuous measurements as well as on-site measurements at three potential major source categories. Since long-term datasets covering more than several weeks (usually obtained from short campaigns) are quite rare, in particular for combined NMHCs and OVOCs, this dataset is very valuable. However, the presented paper needs some major improvement before it can be considered for publication in ACP.

The analysis of data was kept at a very descriptive level. Diurnal and annual variations are shown for single compounds and for the resolved factors from the PMF analysis. A
detailed source apportionment has been performed and the conducted PMF seems to be solid. All applied methods are quite standard and widely used in the scientific community. But apart from the new data, none of the methods are original. The manuscript could profit a lot from deeper analysis of the very nice dataset. The authors mention that the originality of the work derives from using near-field speciation profiles to refine the identification of the profiles. It is a good approach to compare modeled profiles with real measurements and it definitely helps to apportion source profiles. Despite the fact that this is also not a new approach, since there is a lot of literature, where it has been done before, it is new for Paris, and thus it is very useful to have this additional information for the SA. The authors only mention the measured profiles in one half-sentence for motor vehicle sources and for the background factor and not at all in the apportionment section of the wood burning factor. For the latter factor, the agreement is only partially fulfilled, since a lot of ethane and propane is assigned to the factor by PMF but not detected in the fireplace experiment. The authors should discuss the discrepancy and make more use of the added value by the experiments.

The performed analysis of the wind fields and the usage of the simple wind roses for interpretation of air-mass origin are not sufficient. The authors should consider a more elaborate analysis of the air-mass history, e.g. by trajectory analysis/clusters or to take a look into general weather patterns to really draw conclusions.

The structure of the paper could be improved. Many parts of the results section, where interpretation and already discussion is performed, would better fit into the discussion section. Also some parts of the results could be a bit and more stringent and focused. The English throughout the manuscript is okay, however, some rephrasing could sharpen the expressions and bring the content a bit more to the point. Past tense should be used more consequently when methods and results etc. are described.

Specific and technical comments:

2-26: 12 million inhabitants, omit "of"

3-16: some citations could be helpful

3-28: should be elaborated further in the source apportionment section.

4-3ff: some redundant information from the introduction

4-8: please state which background site

The authors should discuss and prove that it is valid to merge the two datasets from two locations. Emissions in urban areas can be quite heterogeneous upon receptor location, wind patterns etc. and some further explanation is needed.

6-8f: calibration once a month seems little; how was the stability of the systems within the months checked? How was the zeroing performed?

7-5: since it is only a very small selection of papers out of the many studies from urban areas, I would put in an "e.g.," before the citations.

7-6: not too many studies were performed in rural areas, but the ones could be mentioned, e.g. Lanz et al. 2009 ACP (doi:10.5194/acp-9-3445-2009) and Leuchner et al. 2015 ACP (doi:10.5194/acp-15-1221-2015)

7-20: "acetonitrile"

7-22: "except"

7-25: why was the proportion of missing values estimated? It should be know exactly.

7-27: "constraints"

It is quite problematic that isoprene had such a high amount of missing values thus it is nor too surprising that the biogenic factor could not be resolved very clearly from some substances.

7-32: since most of PMF studies describe their results in ppb units, it might be nice to have some information how much ppb/ppt the units equal for comparison (throughout the manuscript)

8-1: how were the uncertainties determined? Was there a standard procedure for each substance following guidelines from GAW or ACTRIS?

8-7: what other tests were performed? Might be interesting for other readers.

8-10: it is not entirely clear what exactly was done and what modeling parameters are meant here. Please specify.

8-14: For what reason was the total VOC mass included in the model? What is the benefit of it? In the results it basically looks like this parameter is just apportioned equally to all factors more or less.

9-5: omit "specially".

9-20: "omit" instead of "get out from".

9-22: what is meant by "w/w"?

10-23: Please re-phrase "weather indicators"

11-3ff: the analysis of air-mass origin need to be more profound; only wind direction does not explain the origin of the air masses. To determine if air masses were originating from the clean marine BL or from LRT or deriving from industrial areas from the PBL of central Europe a trajectory analysis (e.g. trajectory clusters) or at least an analysis of general weather patterns could bring further information and support to the assumptions in the manuscript. The wind rose plots could go to the supplement or replaced by a more profound analysis.

11-12: Omit "From this table, it was observed that"

11-23f: what is the reason for the difference? Meteorological conditions? Should be discussed in the discussion section.

11-29: "global studies"

11-34: Last sentence is quite generic.

12-3: omit "time"

12-15: better to use a 24h time instead of am/pm, avoids confusion

12-17ff: the increase occurs until 21h. And after that the levels stay quite high. Is there another (meteorological) reason for this? Interpretation of the graphs should be done in the discussion section.

12-26: Format of citations should be "Gros et al. (2011)" etc.

12-32ff: the conclusions cannot be drawn from wind roses. As stated before a trajectory analysis or similar is needed for that.

12-4ff, entire section: parts of the whole section better fit into the discussion.

13-6: "peaks" rather than increases; values high for longer time (see CO)

13-8: "to" instead of "with"

13-10: it is a bit hard to read because the description jumps between the normal manuscript and the supplement. It might be worth to think about including important graphs that are discussed here into the main part and leave the supplement to really supplemental information.

13-10:"..., however,..."

13-18: in the plots no clear difference can be see between the two species

13-24: these substances are not "emissions" but secondary products of isoprene oxidation

13-27f: into methods section

13-30: "such as" instead of "like"

14-7: "fair" instead of "good"

14-9ff: link to respective figure should be inserted; a table or small figure of the absolute

contributions could be useful

Chapter 3.4: In general, reactivity of the different factors could be investigated in addition to stating only mass contributions; the biogenic factor and the motor vehicle factor would potentially gain importance compared to the factors with more stable gases such as background/natural gas and gas evaporation

14-13: why are the values so high in the middle of the night? Meteorology?

14-19: No reference to the tunnel experiment is given here. Since it is the value added to have these kind of data it should be described here (and discussed later)

14-26: if part of the isoprene is from vehicles, it should show up in F1, since it would be a combustion product instead of an evaporative source.

14-31ff: link to figure 9 should be given.

15-1: correct citation: "Frachon (2009; pers. communication)"

15-30: a lot of acetonitrile was apportioned also to F4, why is that?

16-6: "on" instead of "with" Chapter 3.4.3: reference to experiment needs to be included; a lot of ethane and propane is allocated to this factor that should very likely not be in there

16-16: "includes isoprene's"

16-24: r-value of the PAR correlation missing

16-27: very high values at night, why? Stablity of the atmosphere?

17-14: what does "panel 5" mean?

17-14: the PBL is higher in the afternoon, that contradicts the statement of the diurnal pattern. Emissions might be higher and also a temperature dependency.

17-18: not shown

17-22: "except"

17-31: BLH not defined. Better to use only one abbreviation, e.g. PBL Chapter 3.4.6: the annual course of the background does not fit with similar observations; there should be a maximum in February and minimum in late summer due to the OH reactivity (delayed by 2 months to the annual solar cycle because of the long-lifetime of ethane)

18-7ff: cannot be concluded only from wind roses. Statements need to be supported that it really results from different air masses, since annual cycle does not fit.

Chapter 3.5: pure discussion, also better fits into discussion section.

18-21: background not really fresh emission, but aged air mass

18-21ff: additional information on the contribution of reactivity of each factor would be very useful

19-4: most of the conclusions are quite speculative and need to be supported by more profound analysis

20-9: this discussion would fit to the discussion of factors

20-13: "combination"

20-33: there are some other longer time series, but the dataset indeed is very useful

21-23: absolute values could also be interesting in addition for comparison to other studies

22-12: why were exactly these studies chosen? There are many other studies described in the literature from e.g., Houston, Santiago de Chile, etc. Graphics: some of the axis titles and labels (e.g., Figs. 8, 9 but also all others) are quite small and might be hard to read when formatted to the actual page size.

Fig. 5 could go to supplement or omitted if replaced with a more profound air mass investigation

[Figure]

Fig. 8: logarithmic scale makes it hard to see the "real" profile for absolute contributions"

47-12: the factors could be shown in supplement, would be interesting to see

47-6ff: quite important information. Could be moved to methods section of main text.

Appendix A: here the calculation of uncertainties could be shown.

---

## Referee Comment (RC2) · Anonymous Referee #2 · 7 Jun 2016

**acp-2016-185: Seasonal variability and source apportionment of volatile organic compounds (VOCs) in the Paris megacity (France)**
**Baudic et al.**

General/scientific comments

This paper reports the results of a source apportionment by positive matrix factorisation (PMF) of the concentrations of a suite of ambient VOCs measured in urban background air in Paris over a period of several months from January to November 2010. VOCs were measured both by on-line GC and by PTR-MS. In order to help in the assignment of some of the factors emanating from the application of PMF, the authors compared the speciated VOC profiles of the factors with speciated VOC profiles the authors separately measured at three locations where they assume that a single emission source will dominate the ambient VOC, specifically: (1) measurements during busy (and traffic jam) periods in a highway tunnel, to represent vehicle related VOC emissions; (2) measurements close to a domestic gas flue, to represent natural gas source; and (3) measurements at a fireplace facility, to represent residential wood-burning emissions (the authors acknowledge that their measured VOC profile from this source may be less quantitative than for other source profiles). To further assist in the assignment of PMF factors to particular VOC emission sources the authors also make use of additional co-located atmospheric compositional data available to them, such as NO, CO and black carbon (BC).

The authors present results in which the ambient urban background VOC has been apportioned into six sources, each of which has been assigned an identification, albeit that one factor is assigned to be a mixed natural gas/background source. The analysis presented includes speciated VOC profiles for each factor and monthly and average-hourly variation in the absolute contribution (ug/m3) of each of the six identified factors/sources. The largest contributions in total are from traffic-related activities (through two identified factors: motor-vehicle exhaust, and gasoline evaporation), although all six identified sources have not that dissimilar relative contributions, on average. A noteworthy observation is significant contributions to ambient VOC from wood burning, 18% on average but up to ~50% at times in winter. Biogenic emissions were also reported to be significant, 15% on average but more in summer.

The authors have a large dataset of time-resolved speciated VOC over an extended time period, almost a year. They have used standard, but appropriate, statistical methods to endeavour to decompose the ambient measurements into individual source contributions. These are statistical, rather than dispersion-/chemistry-based source apportionments. These methods have been widely employed to apportion ambient PM, but less so for VOC.

The presentation of results and their discussion are largely descriptive, in the sense that the authors present the details for their factor/source contributions and their monthly and hourly variations, which are rationalised with general text about anticipated behaviour of particular sets of pollutant mixtures in the urban background atmosphere. The authors also present a qualitative and quantitative comparison of their VOC source apportionment with previous literature.

Although in one sense the presentation of this work could be described as 'formulaic' – following previous data presentation and analysis styles – nonetheless the large dataset presented here for a large European city presents a valuable addition to the VOC apportionment literature. A particular feature is the presentation of VOC speciated profiles for three potential VOC sources, although the authors acknowledge some shortcomings in these. The paper is already lengthy and contains much new data, supported by detailed descriptions of the data collection and processing protocols and results. As an overall summary, the content of the paper is suitable for publication in ACP.

Technical comments

The tables and figures are generally clearly presented. The written text is largely unambiguous in conveying its meaning, but it is overly long in places. There are instances where introductory sentences to a section could be substantially abbreviated, or even deleted as repeating what the reader will have picked up from the methods section. The authors should be encouraged to edit text further for conciseness of expression.

The following are more specific comments.

P6, L18: should read "Raw data were corrected using…."?

P6, L24: delete "is" before "analyzer"

P6, L32: delete "at"

P7, L22: should read "except for"

P8, L9: I don't understand what is being described in the sentence beginning "Finally, these different processings…." Please rephrase this sentence to make clear to what "it" is referring in this sentence and to clarify what the procedure undertaken was.

P10, L26: the phrase "fairly comparable" is not scientifically precise.

P10, L29: "a few flurries" of what? Please write specific statements about the nature of the weather.

P11, L16: the phrase "contribute to the tune of" is too colloquial; please use more direct wording.

P11, L30-L35: there are several instances in these lines of text of negative values for VOC concentrations. These are surely some aberration (albeit repeated aberration) of typing error. Please correct.

P13, L3: I do not understand the scientific sense of the sentence starting "With an atmospheric residence time…" How does the statement at the end of this sentence (about methanol emissions contributing to background levels) derive from, or otherwise relate to, the text at the start of the sentence about methanol residence time? Please reword to clarify.

P14, L24: I do not follow the scientific logic here. The text appears to state that iso-pentane is known to be a key tracer for gasoline evaporation, but also to say that iso-pentane was not present in the speciated profile the authors have assigned in their work to gasoline evaporation.

P14, L32: please explain more clearly what about the monthly change "remains ambiguous".

P20, L19: The phrasing that the mean temperature was "in the range +/- 20 degC" does not make sense. Either quote the range, or quote the mean and some recognised statistic of the variation about the mean. Likewise for later in this sentence in connection with "+/- 16 degC".

Caption of Table 1: It would be helpful for the caption to remind the reader with a statement of the time resolution of the raw data from which these statistical summaries are derived, and of the time duration/dates of the total dataset.

---

## Author Comment (AC1) · 29 Jul 2016

Dear Referee#1,

We would like to thank you for your general feedback and your useful comments/questions for improving the quality of this manuscript. All your comments were taken into account in the revised version of the manuscript.

Authors' answers were uploaded as a *.pdf file and were displayed as a supplement to this comment.

Sincerely, Mrs. Alexia Baudic
Please also note the supplement to this comment:
http://www.atmos-chem-phys-discuss.net/acp-2016-185/acp-2016-185-AC1-supplement.pdf

Interactive
comment
* * *
[Figure]

**Supplement:**

* * *
**"Seasonal variability and source apportionment of volatile organic compounds (VOC) in the Paris megacity (France)" by A. Baudic et al.**

**Authors' Responses to Referee #1**

We would like to thank the Referee #1 for her/his general feedback and each of her/his useful comments/questions for improving the quality of this manuscript. All of them have been taken into account when preparing the revised version of the manuscript. In the present document, authors' answers to the specific comments addressed by Referee #1 are mentioned in **blue**, while changes made to the revised manuscript are shown in *italic.*

General comments

The presented manuscript presents data on NMHCs and OVOCs from two sampling sites in Paris covering about 10 months of quasi-continuous measurements as well as on-site measurements at three potential major source categories. Since long-term datasets covering more than several weeks (usually obtained from short campaigns) are quite rare, in particular for combined NMHCs and OVOCs, this dataset is very valuable. However, the presented paper needs some major improvement before it can be considered for publication in ACP.

The analysis of data was kept at a very descriptive level. Diurnal and annual variations are shown for single compounds and for the resolved factors from the PMF analysis. A detailed source apportionment has been performed and the conducted PMF seems to be solid. All applied methods are quite standard and widely used in the scientific community. But apart from the new data, none of the methods are original. The manuscript could profit a lot from deeper analysis of the very nice dataset. The authors mention that the originality of the work derives from using near-field speciation profiles to refine the identification of the profiles. It is a good approach to compare modeled profiles with real measurements and it definitely helps to apportion source profiles. Despite the fact that this is also not a new approach, since there is a lot of literature, where it has been done before, it is new for Paris, and thus it is very useful to have this additional information for the SA. The authors only mention the measured profiles in one half-sentence for motor vehicle sources and for the background factor and not at all in the apportionment section of the wood burning factor. For the latter factor, the agreement is only partially fulfilled, since a lot of ethane and propane is assigned to the factor by PMF but not detected in the fireplace experiment. The authors should discuss the discrepancy and make more use of the added value by the experiments.

The performed analysis of the wind fields and the usage of the simple wind roses for interpretation of air-mass origin are not sufficient. The authors should consider a more elaborate analysis of the air-mass history, e.g. by trajectory analysis/clusters or to take a look into general weather patterns to really draw conclusions.

The structure of the paper could be improved. Many parts of the results section, where interpretation and already discussion is performed, would better fit into the discussion section. Also some parts of the results could be a bit and more stringent and focused. The English throughout the manuscript is okay; however, some rephrasing could sharpen the expressions and bring the content a bit more to the point. Past tense should be used more consequently when methods and results etc. are described.

[1]
* * *
The large dataset presented here for Paris constitutes a valuable addition to the VOC apportionment literature as it combines non-methane hydrocarbons (NMHC) and oxygenated VOC (OVOC) measurements performed over an extended time period, almost a year – which is new for Paris. The objectives of this paper are to (i) assess VOC concentration levels measured in Paris for further comparisons in other urban areas, (ii) give some information about seasonal and diurnal variabilities of a selection of VOC and more particularly (iii) present a Source Apportionment (SA) analysis performed using the Positive Matrix Factorization (PMF) standard statistical method. In order to help strengthen the identification of factors/sources issued from PMF simulations, we used speciation profiles determined from near-field measurements (inside a highway tunnel, at a fireplace and from a domestic gas flue). PMF results therefore appear more robust when combining with source profiles studies.

We agree with the reviewer that near-field observations have not been enough presented in the manuscript. We only compared PMF factor profiles with speciated source profiles by exploring major compounds. In the revised version of the manuscript, we now discuss similarities and discrepancies between the two approaches (see comments n° 45 - 50 and sub-sections 3.4.1, 3.4.3, 3.4.6).

To explain the seasonal variability of some emission sources (especially, the PMF factor mixing natural gas and background influences), we had performed monthly wind roses, what was not objectively considered as sufficient. As advised, an elaborate analysis of the air-mass history using the HYSPLIT model was done. To determine main air-mass origins affecting the "Île-de-France" region, a clustering analysis was also undertaken (see comments n° 22, 31, 59). We note that Paris is mainly affected by clean air masses coming from the west and usually associated with local and regional pollution conditions. To a lesser extent, air masses originated from the north and northeast were especially observed during springtime and are typically associated with continental influences from the Eastern Europe, the Benelux area and the north of Germany.

In this new version of the manuscript, results and discussion sections were grouped with the purpose of avoiding redundancies and any overlapping. We chose to present, interpret and discuss VOC data and the modeled PMF factors variabilities and compare them with other SA studies performed in Europe and in the world within one main section. Following reviewer's suggestions, the authors made efforts to write a revised version of the manuscript with conciseness. Some rephrasing also aimed at bringing the content a bit more to the point.

Note that a number of changes have been made to address the issues that emerged from the reviewer. Several figures and tables were added and modified in the revised manuscript (e.g. air masses trajectory analysis – Fig. 5; comparison between "highway tunnel" and "motor vehicle exhaust" profiles – Fig. 9; relative and absolute contributions of reactivity of each PMF factor – Fig. 11) and in the supplementary (e.g. PMF profiles issued from simulations using the FRANCIPOL dataset & comparison of speciated PMF profiles issued from our two different datasets – Section S1; representativeness of meteorological parameters in 2010 – S4; mean absolute contributions of factors per month in 2010). Consequently, figures and tables numeration is now different in this new version.
* * *
Specific and technical comments:

**1/** P2, L-26: 12 million inhabitants, omit "of".
**Correction applied in the revised manuscript:** *"Paris and its surroundings (also called the Île-de-France region) constitute the second largest European megacity with about 12 million inhabitants, representing 20% of the French national population distributed over only 2% of its territory (Eurostat, 2014)."*

**2/** P3, L-16: Some citations could be helpful.
**As suggested, a small selection of citations** *(e.g. Crippa et al., 2013; Skyllakou et al., 2014; Ait-Helal et al., 2014; Beekmann et al., 2015 and references therein)* **was added to the revised manuscript. It comprises different scientific studies documenting gaseous and particulate compounds in ambient air within the MEGAPOLI project: variabilities, source apportionment, origins and modeling.**

**3/** P3, L-28: Should be elaborated further in the source apportionment section.
**The following sentence "The originality of this study stands in [...] to refine the identification of apportioned sources" was kept in the introduction in order to inform readers about the added value of near-field speciation profiles. This approach thus allows reinforcing the source apportionment analysis and the corresponding discussion in Section 3.**

**4/** P4, L-3ff: Some redundant information from the introduction.
**The following sentence "Ambient air measurements of VOCs [...] within the framework of two different research projects" was removed entirely from the text to avoid creating duplication and confusion as main information were given in the introduction.**

**5/** P4, L-8: Please state which background site.
The authors should discuss and prove that it is valid to merge the two datasets from two locations. Emissions in urban areas can be quite heterogeneous upon receptor location, wind patterns etc. and some further explanation is needed.
**As reported in Table 1, VOC measurements were undertaken at two different locations between March and November 2010. The first sampling site is the "Laboratoire d'Hygiène de la Ville de Paris" (LHVP) located in the southern part of Paris (13th district – 48°51'N, 02°20'E). LHVP is juxtaposed with a large public garden (called "Parc de Choisy") at approximately 400 meters from Place d'Italie (grouping a shopping center and main boulevards). The second monitoring site is the "Les Halles" subway station (1st district – 48°51'N, 02°20'E) located about 2 kilometers away from LHVP.**

**Table 1.** Summary table restating specific features of sampling locations considered in this study.

| Site | Coordinates | Experiments | |
|---|---|---|---|
| LHVP | 48°82'N – 02°35'E | MEGAPOLI (J-F 2010) | GC-FID + PTR-MS |
| | | FRANCIPOL (M-N 2010) | PTR-MS |
| Les 'Halles' subway station | 48°514N – 02°20'E | FRANCIPOL (M-N 2010) | GC-FID |

LHVP = Laboratoire d'Hygiène de la Ville de Paris.
J-F 2010 = January-February 2010 (MEGAPOLI experiment)      M-N 2010 = March-November 2010 (FRANCIPOL experiment)
GC-FID = Gas Chromatograph with a Flame Ionization Detector      PTR-MS = Proton Transfer Reaction – Mass Spectrometer

**Due to the low intensity of the surrounding activities, these two locations correspond to actual and previous urban background stations of the Airparif regional monitoring air quality network and have been considered as such within a previous study**

(Gros et al., 2011). This station typology is based on two main criteria: (1) the population density is at least 4 000 inhabitants per square kilometre within a 1-km radius of the station and (2) no major traffic road is located within 300 metres (criteria from 2008/50/CE European Directive).

The characterization of site typologies can be conducted by studying the nitrogen monoxide (NO): nitrogen dioxide ($NO_2$) ratio. NO is known to be a vehicle pollution indicator whereas $NO_2$ has also an important secondary fraction. To consider a station as an urban background site, the ratio $R$ between annual average NO and $NO_2$ concentrations ($NO/NO_2$) should be less than 1.5 ppb/ppb as indicated in a report (Mathé, 2010) written at the national level for regulatory purposes. It compiles the criteria for air quality measurement site implementation.

We observe here a very close NO: $NO_2$ ratio (expressed as ppb: ppb) between both sites in 2010: 0.40 (LHVP) *versus* 0.38 (Les Halles). The same conclusion can be made for the years 2008 and 2009 (0.45 ± 0.01 for LHVP and 0.48 ± 0.01 ppb: ppb for "Les Halles"). With NO: $NO_2$ ratios very similar and less than 1.5, this confirms that these two locations have the same site typology and can be considered as urban background stations.

The concept of the spatial representativeness is also important in that context. Mathé (2010) considers that the stations should preferably have a radius of the area of representativeness from 100 meters to 2 kilometers or an area of several square kilometers for urban background sites. However, this representativeness area is dependent on many factors: the topographic location, the local variability of emissions, wind patterns and pollutants concerned.

To prove that it is valid to merge these two specific datasets from different locations, we opted for a comparison of PMF results derived from MEGAPOLI-FRANCIPOL (as performed in the present paper) and FRANCIPOL data files respectively. PMF modeling simulations were performed using only the FRANCIPOL dataset (it corresponds to 88% of the total data matrix) and operating with similar processing conditions: *(i)* data quality control, *(ii)* concentration and uncertainty files checking, *(iii)* same species categories assignment and *(iv)* the optimum factors number by exploring several statistical parameters: IM, IS, $Q_{true}/Q_{expected}$, $R^2$ for VOC. Just like final PMF results, a 6-factor solution was then chosen in terms of interpretability and fitting scores (see Figure 1). A good agreement was found for the "Motor Vehicle Exhaust", "Evaporative sources", "Wood Burning", "Biogenic" sources (0.93 < R < 0.99 -- 0.87 < $R^2$ < 0.98). More significant differences were identified for "Solvents use" and "Natural Gas & Background" sources (0.71< R < 0.78 – 0.51 < $R^2$ < 0.60), especially for aromatic compounds (e.g. benzene, toluene, xylenes + $C_8$). A ~ 50 % loss of aromatics contributions is observed when considering the "MEGAPOLI-FRANCIPOL" dataset compared to the 'FRANCIPOL' one. It is compensated by a contribution of these compounds in the "Natural Gas and Background" source.

This observation can be explained by:

   i.    The challenge of splitting "Solvents use" and "Natural Gas & Background" sources when considering a 5-factor PMF solution (see Section A2 in the initial manuscript). PMF artifacts may occur.

[4]

    **ii.** **The summer variability of the "solvents use" source is more meaningful when considering the "FRANCIPOL" dataset. Including the "MEGAPOLI-FRANCIPOL" dataset (or adding a winter influence) might down the weight of aromatics in the industrial source profile.**

**Based on all of these different reasons, we believe that the merging of the two datasets is quite valid.**

**The sub-section 2.1 (e.g. Sampling sites description) was revised for a better understanding of the involved sampling sites. To make the manuscript more readable, another sub-section was created to discuss the representativeness of sampling stations. The graphs showing the comparison of PMF results between MEGAPOLI-FRANCIPOL and FRANCIPOL (Figure 2) have been added in the supplementary section S1.**

**In the revised manuscript, it now reads:**

*"Due to the low intensity of the surrounding activities, the LHVP and Les Halles sites are considered as urban background stations by AIRPARIF and by previous scientific studies (Favez et al., 2007; Sciare et al., 2010; Gros et al., 2011). In accordance with the 2008/50/EC European Directive, this station typology is based on two main criteria: (1) the population density is at least 4 000 inhabitants per square kilometre within a 1-km radius of the station and (2) no major traffic road is located within 300 metres.*

*This characterization of site typologies can be conducted by studying the nitrogen monoxide (NO): nitrogen dioxide ($NO_2$) ratio. NO is known to be a vehicle pollution indicator whereas $NO_2$ has also an important secondary fraction. To consider a station as an urban background site, the ratio R between annual average NO and $NO_2$ concentrations ($NO/NO_2$) should be less than 1.5 ppb/ppb as indicated in a report (Mathé, 2010) written at the national level for regulatory purposes.*

*We observe here a very close NO: $NO_2$ ratio (expressed as ppb: ppb) between both sites in 2010: 0.40 (LHVP) versus 0.38 (Les Halles). The same conclusion can be made for the years 2008 and 2009 (0.45 ± 0.01 for LHVP and 0.48 ± 0.01 ppb: ppb for "Les Halles"). With NO: $NO_2$ ratios very similar and less than 1.5, this confirms that these two locations have the same site typology and can be considered as urban background stations.*

*To prove that it was valid to merge these two specific datasets from different locations, we opted for a comparison of PMF results derived from MEGAPOLI-FRANCIPOL (as performed in the present paper) and FRANCIPOL data files respectively. PMF modeling simulations were performed using only the FRANCIPOL dataset. A good agreement was found for the majority of the emission sources. The graphs showing the comparison of PMF results between MEGAPOLI-FRANCIPOL and FRANCIPOL are shown in the supplementary section S1".*

Directive 2008/50/EC of the European Parliament and of the Council of 21 May 2008 on ambient air quality and cleaner air for Europe. Official Journal of the European Communities. 11 June 2008.

Mathé, F. (2010). Evolution de la classification et des critères d'implantation des stations de mesure de la qualité de l'air – Participation à la réactualisation du guide de classification des stations, LCSQA-Mines Douai report.
* * *
**Figure 1** - **Source composition profiles of the 6-factor PMF solution derived from the FRANCIPOL dataset (March-November 2010).** The concentrations (in μg.m$^{-3}$) and the percent of each species apportioned to the factor are displayed as a pale blue bar and a color box, respectively). **(a)** F1 – Motor Vehicle Exhaust; **(b)** F2 - Evaporative sources; **(c)** F3 - Wood Burning; **(d)** F4 - Biogenic; **(e)** F5 – Solvents use source; **(f)** F6 - Natural Gas and Background.

[Figure]

[6]
* * *
**Figure 2 - Comparison of speciated PMF profiles issued from both FRANCIPOL AND MEGAPOLI/FRANCIPOL datasets.** The contributions of species in the PMF profile are expressed in %. **(a)** F1 – Motor Vehicle Exhaust; **(b)** F2 - Evaporative sources; **(c)** F3 - Wood Burning; **(d)** F4 - Biogenic; **(e)** F5 – Solvents use source; **(f)** F6 - Natural Gas and Background**.**

[Figure]

[Figure]
* * *
[Figure]

[Figure]
* * *
[Figure]

**6/** P6, L-8f: Calibration once a month seems little; how was the stability of the systems within the months checked? How the zeroing was performed?
**Full calibrations were performed once a month by the regional air quality network monitoring AIRPARIF using a standard gas mixture containing only propane (25.5 ppb, Air Liquide). This certified gas is sampled in the same conditions than ambient air (e.g. 30 min. sampling and 30 min. analysis) during 6 hours. Response coefficients were calculated for each compound using the Effective Carbon Number (ECN) method, described in Badol et al. (2004).**

**In addition, the stability of GC-FID systems was checked over time by controlling chromatographic profiles, the baseline and the stability of intrinsic parameters (pressure, temperature, leak tests). The zeroing was performed every 6 months using a zero air bottle in order to detect any instability or problem with the system.**

**Correction applied in the revised manuscript:** *"A full calibration was performed once a month with a standard gas mixture containing only propane during 6 hours. As the FID response is proportional to the Effective Carbon Number (ECN) in the molecule, calibration coefficients were calculated for each compound and regularly checked so that they drifted no more than ± 5% (tolerance threshold). In addition, a zeroing was carried out every 6 months using a zero air bottle in order to detect any instability or problem with the GC-FID system. LoD were assessed at 0.024 µg.m$^{-3}$ for all the selected compounds, except for n-hexane (0.013 µg.m$^{-3}$) (0.023 – 0.004 ppb)."*

Badol, C., Borbon, A., Locoge, N., Léonardis, T. and Galloo, J.-C.: An automated monitoring system for VOC ozone precursors in ambient air: development, implementation and data analysis, Anal. Bioanal. Chem., 378, 1815-1824, 2004.

**7/** P7, L-5: Since it is only a very small selection of papers out of the many studies from urban areas, I would put in an "e.g.," before the citations.
**As suggested, an "e.g." was added before the citations, both for studies from urban and rural areas.**

**8/** P7, L-6: Not too many studies were performed in rural areas, but the ones could be mentioned, e.g. Lanz et al. 2009 ACP (doi: 10.5194/acp-9-3445-2009) and Leuchner et al.2015 ACP (doi: 10.5194/acp-15-1221-2015).
**The two suggested references were added to the selection of papers of the studies from rural areas. Subsequently, bibliographic references were also updated.**

**Correction applied in the revised manuscript:** *" Concerning VOC, PMF studies have been conducted in urban (e.g Brown et al., 2007 – LA, USA; Lanz et al., 2008 – Zürich, Switzerland; Morino et al., 2011 – Tokyo, Japan; Yurdakul et al., 2013 – Ankara, Turkey) and rural areas (e.g. Sauvage et al., 2009 – France; Lanz et al., 2009 – Jungfraujoch, Switzerland; Leuchner et al., 2015 – Hohenpeissenberg, Germany)."*

**9/** P7, L-20: "acetonitrile".
**The word "acetonitrile" was correctly modified in the revised manuscript.**

**10/** P7, L-22: "except".
**The word "expected" was substituted by "expect".**
* * *
**11/** P7, L-25: Why was the proportion of missing values estimated? It should be know exactly.
**The virtual number "-999" was defined as the missing value indicator in the concentration input file. The PMF model was programmed to find all instances of "-999" in the data file. It is also able to provide both the proportions of raw and modeled samples for each species. The percentage of missing values is thus exactly known. It is ranging from 19 % (especially for compounds measured by PTR-MS) to 41 % (only for isoprene). To avoid any misunderstanding, the corresponding sentence was rephrased.**

**Correction applied in the revised manuscript:** *"The proportion of missing values is ranging from 19 % (especially for compounds measured by PTR-MS) to 41 % (only for isoprene)".*

**12/** P7, L-27: "constraints"
It is quite problematic that isoprene had such a high amount of missing values thus it is nor too surprising that the biogenic factor could not be resolved very clearly from some substances.
**On one hand, the word "constrains" was substituted by "constraints".**

**On the other hand, the high proportion of missing values for isoprene is partly due to analytical issues on the operating GC-FID in July 2010. Each missing data point was substituted with the median concentration of this species over all the measurements and associated with an uncertainty of four times the species-specific median (as suggested in Norris et al., 2014). Missing values are usually downweighted by large uncertainty values. In this case, concentrations are typically below uncertainties. Subsequently, they were determined to have a low Signal-to-Noise (1.7 for isoprene here – the lowest ratio of our species selection). This implies significant difficulties in modeling this compound, as demonstrated by statistical parameters ($R^2$ = 0.29 – Slope = 0.53). The limitations of the PMF model to simulate isoprene have been kept in mind within the reconstructed results description and discussions.**

**13/** P7, L-32: Since most of PMF studies describe their results in ppb units, it might be nice to have some information how much ppb/ppt the units equal for comparison (throughout the manuscript).
**For further qualitative and quantitative comparisons, VOC concentrations are expressed both in micrograms per cubic meter (µg.m$^{-3}$) and in parts per billion (ppb). Conversion factors (µg.m$^{-3}$ to ppb) were calculated considering the molecular mass of a gaseous compound (g.mol$^{-1}$) over the air molar volume (24 L at 20°C).**

**Correction applied in the revised manuscript: Throughout the manuscript, each concentration/value/result initially in µg.m$^{-3}$ is also mentioned in ppb (except for PMF results). The conversion factor for each species is reported in Table 1.**

**14/** P8, L-1: How were the uncertainties determined? Was there a standard procedure for each substance following guidelines from GAW or ACTRIS?
**The determination of uncertainties on data measured both by PTR-MS and GC-FID did not specically follow a standard procedure from Global Atmosphere Watch (GAW) or Aerosols, Clouds, and Trace gases Research InfraStructure (ACTRIS) network.**

**(O) VOC data measured by PTR-MS were provided both by LCE (Megapoli period) and LSCE (Francipol period). From the available data, uncertainty calculations were estimated by taking into account errors on standard gas, calibrations, blanks,**

**reproducibility/repeatability, linearity and Relative Humidity (RH) parameters. The measurement uncertainty was calculated from the theory of propagation of the errors and was found to be ~ 20% (Gros et al., 2011; Dolgorouky et al., 2012). Finally, PMF empirical tests were performed and aimed at changing input analytical uncertainty from 20% to 25% for species measured by PTR-MS. As a consequence of these tests, no significant change was observed in PMF modeling results.**

**VOC data measured by GC-FID during the Francipol campaign were provided by the regional air quality monitoring network AIRPARIF. Nevertheless, no uncertainty calculations were supplied to us. As these calibration data were no longer available (processing software changes, old releases not adequate, compatibility issues), we could not make an exhaustive uncertainty calculation. In this present study, GC-FID data uncertainties were estimated at 15% in agreement with previous studies (Gros et al., 2011; Gaimoz et al., 2011; Dolgorouky et al., 2012).**

**Correction applied into the methods section (Subsection 2.3.2 – VOC measurements using a Proton Transfer Reaction-Mass Spectrometer (PTR-MS)):** *"The analytical uncertainty on all data was estimated by taking into account errors on standard gas, calibrations, blanks, reproducibility/repeatability, linearity and Relative Humidity (RH) parameters. The measurement uncertainty was estimated at ± 20% in agreement with previous studies (Gros et al., 2011; Dolgorouky et al., 2012)."*

**Correction applied into the methods section (Subsection 2.4.2 – NMHC on-line measurements by Gas Chromatography (GC)):** *"The analytical uncertainty on all data was estimated at ± 15% in agreement with previous studies (Gros et al., 2011; Gaimoz et al., 2011, Dolgorouky et al., 2012)."*
* * *
**15/** P8, L-4: What other tests were performed? Might be interesting for other readers?

**16/** P8, L-10: It is not entirely clear what exactly was done and what modeling parameters are meant here.

**As previously reported, isoprene exhibits a high proportion of missing values (41 %). This actually means that 59% of isoprene data are modeled by the PMF technique, what is not objectively considered as sufficient. To address this problem, input values of isoprene were modified when detecting missing data. We have investigated the possibility of replacing missing values with data more appropriate than the median. Indeed, a "virtual" averaged pattern was calculated from 1-h real samples observed in June and August (to keep the summer variability of isoprene). Another option was to preserve raw data of isoprene (no subsequent changes) and to increase the analytical uncertainty initially estimated at 15% (gradually from 15 % to 30%) – instead of categorizing isoprene as *weak* (as uncertainties are tripled > lower Signal to Noise). This was intended to better display this biogenic compound. PMF simulations were performed by considering these different options. As a consequence of these tests, no significant improvement on the quality of modeling isoprene was observed. Regular statistical parameters such as $R^2$, slope, slope/intercept SE (Standard Error) were used to draw such conclusions. As empirical tests have not helped, isoprene is still categorizing as *strong*.**

**Correction applied in the revised manuscript:** *"To address this lack of isoprene data, several empirical tests (e.g. simulating an averaged seasonal/diurnal cycle of isoprene or increasing the analytical uncertainty of raw data from 15 % up to 30 %) were conducted within PMF simulations with the aim of better modeling the variability of this compound. As a consequence of these tests, no significant improvement on the quality of modeling isoprene was observed. Finally, isoprene is still categorizing as strong here".*

**17/** P8, L-14: For what reason was the total VOC mass included in the model? What is the benefit of it? In the results it basically looks like this parameter is just apportioned equally to all factors more or less.

**The total VOC mass was included in the PMF model in order to directly estimate VOC source contributions (as a whole) and not just to quantify the proportion of each species in a factor profile (when considering individually species). ∑VOC was defined as *Total Variable* and automatically categorized as *weak* to lower its influence in the final PMF results. Adding the ∑VOC variable allowed to ensure a good diagnosis of final PMF results (for instance, VOC predicted *vs.* VOC measured – Fig. A.1 in the initial version of the manuscript).**

**Considering ∑VOC in the model is usually made in PMF studies because it is worthwhile when comparing the contribution of VOC sources with modeling results from previous studies and emission inventories, in which the sum of all individual VOC species emitted from a specific source is taken into account.**

**18/** P9, L-5: Omit "specially".

**Correction applied in the revised manuscript:** *"The speciated profiles of these different anthropogenic sources and their representativity are given here."*

**19/** P9, L-20: "Omit" instead of "get out from".

**Correction applied in the revised manuscript:** *"In order to omit any local background, nighttime values (as suggested in Ammoura et al., 2014) were subtracted from the peak VOC concentrations."*
* * *
**20/** P9, L-22: What is meant by "w/w"?

**The term "w/w" means "weight of each compound" out of "weight of the total VOC mass".**

**21/** P10, L-23: Please re-phrase "weather indicators".

**Correction applied in the revised manuscript:** *"Meteorological parameters (e.g. temperature, relative humidity, rainfall, sun exposure, boundary-layer height, wind speed and direction) are known to be key factors governing seasonal and diurnal variations of air pollutant levels."*

**22/** P11, L-3ff: The analysis of air-mass origin need to be more profound; only wind direction does not explain the origin of the air masses. To determine if air masses were originating from the clean marine BL or from LRT or deriving from industrial areas from the PBL of central Europe a trajectory analysis (e.g. trajectory clusters) or at least an analysis of general weather patterns could bring further information and support to the assumptions in the manuscript. The wind rose plots could go to the supplement or replaced by a more profound analysis.

**As recommended in general comments, an elaborate analysis of the air-mass history was performed and presented in the revised manuscript as described below.**

**Correction applied into the methods section (Subsection 2.3.3 – Additional data available):** *"In order to determine the air-masses origin, 5-day backtrajectories were calculated every 3 hours from the PC based version of the Hybrid Single Particle Lagrangian Integrated Trajectory (Hysplit) model (Stein et al., 2015) with Global Data Assimilation System (GDAS) meteorological field data. Backtrajectories were set to end at Paris coordinates (49°02'N, 02°53'E) at 500 m a.g.l."*

**Within this study, 2 164 backtrajectories were calculated every 3 hours using the HYSPLIT model. A clustering algorithm was used to group these trajectories depending on their direction, their speed and their altitude (Sirois and Bottenheim, 1995; Sauvage et al., 2009). For each backtrajectory, *xyz* (latitude, longitude, altitude) coordinates of the trajectory endpoint were used as input variables for the clustering technique. To determine the optimum number of clusters (N), plotting the percentage of change in Total Spatial Variance (TSV) against the number of cluster was used. A slope break is observed for N=4 where the change in TSV (%) is at its maximum (~70%). The clustering analysis of backtrajectories was performed throughout the study period (J-N 2010) and is presented in Figure 3.**

**Paris is mostly influenced over the year by air masses coming from the west (ca. 62 % - CL#1 + CL#2) and usually associated with clean marine air influences from the Atlantic Ocean. They are typically representative of local and regional pollution conditions, as already observed in Gros et al., 2011; Gaimoz et al., 2011; Dolgorouky et al., 2012 and Petit et al., 2015. To a lesser extent, Paris can be affected by northeast air masses (26%) originating from Eastern France, the Benelux area, Northern Germany and are indicative of continental imports of long-lived pollutants. North air masses (passing through the United Kingdom and the north of France) can also affect the receptor site (11%). Air masses coming from the west (LRT and SRT) are generally observed in summer and autumn (ca. 32 – 41%) whereas northeast air masses (CL#3) are found to be significant in winter (34%) - specifically in February - and most frequently in spring (ca. 40%).**

**This trajectory analysis was then used in Section 3 when discussing the air-mass origin of the background factor.**
* * *
**Correction applied in the revised manuscript:** *"For the year 2010, Paris was mainly influenced by air masses coming from the west (62%) and usually associated with clean marine air influences from the Atlantic Ocean (see Fig.5).They are typically representative of local and regional pollution conditions, as already observed in Gros et al., 2011; Gaimoz et al., 2011; Dolgorouky et al., 2012 and Petit et al., 2015. To a lesser extent, Paris can be affected by northeast air masses (26%) originating from Eastern France, the Benelux area, Northern Germany and are indicative of continental imports of long-lived pollutants (Gaimoz et al., 2011). Air masses coming from the west are generally observed in summer and autumn (32-41%) whereas northeast air masses are found to be significant in winter (34%) and most frequently in spring (ca. 40%) due to the stagnation of an anticyclone surrounding the British Isles (Monthly weather report for Paris and its surroundings during April 2010, Météo-France) during that period."*

*Sauvage, S., Plaisance, H., Locoge, N., Wroblewski, A., Coddeville, P and Galloo, J.C.: Long term measurement and source apportionment of non-methane hydrocarbons in three French rural areas, Atm. Env., 43, 2430-2441, doi: 10.1016/j.atmosenv.2009.01.001, 2009.*

*Sirois, A. and Bottenheim, J.W.: Use of backward trajectories to interpret a 5-Year Record of PAN and O3 ambient air concentrations at Kejimkujik national Park, Nova-Scotia, Journal of Geographical Research-Atmospheres, 100 (D2), 2867-2881, 1995.*

*Stein, A.F., Draxler, R.R., Rolph, G.D., Stunder, B.J.B., Cohen, M.D. and Ngan, F.: NOAA's HYSPLIT Atmospheric Transport and Dispersion Modeling System, Bull. Am. Meteorol. Soc., 96(12), 2059-2077, doi:10.1175/BAMS-D-14-00110.1, 2015.*
*.*

[Figure]

| Cluster CL# | Air mass origins | Year (J-N) | Winter | Spring | Summer | Autumn |
|---|---|---|---|---|---|---|
| CL#1 | LRT (West) | 29 % | 24 % | 25 % | 32 % | 32 % |
| CL#2 | SRT (West) | 33 % | 29 % | 25 % | 41 % | 33 % |
| CL#3 | Continental influences (NE) | 26 % | 34 % | 39 % | 19 % | 22 % |
| CL#4 | North | 11 % | 13 % | 11 % | 8 % | 12 % |

*LRT: Long Range Transport*

*SRT: Short Range Transport*

**Figure 3 -** Average trajectories obtained after clustering analysis and the relative proportion of clusters (%) over the year and per season.

[16]
* * *
**23/** P11, L-12: Omit "From this table, it was observed that".

**Correction applied in the revised manuscript:** *"The average composition in VOC was mainly characterized by oxygenated species (0.7 – 5.9 µg.m⁻³ (0.4 – 4.5 ppb); 36.5 % of the TVOC mass), alkanes (0.5 – 4.6 µg.m⁻³ (0.1 – 3.7 ppb); 39.1 %) followed by aromatics (1.1 – 3.3 µg.m⁻³ (0.3 – 0.9 ppb); 16.9 %) and to a lesser extent by alkenes, alkynes and dienes (0.3 – 1.6 µg.m⁻³ (0.1 – 1.3 ppb); 7.5 %)."*

**24/** P11, L-23f: What is the reason for the difference? Meteorological conditions? Should be discussed in the discussion section.

**Mean concentration levels of aromatics and OVOCs measured by PTR-MS in 2007 (Gaimoz et al., 2011) were reported in Table 2 (of the initial manuscript) and compared with our study (2010). For a relevant comparison, VOC measurements performed during the same sampling time (25 May – 14 June) were considered. A significant decrease in VOC concentrations was observed between spring 2007 and spring 2010, especially for aromatics (xylenes + C₈, benzene). This finding is in agreement with decreasing downward trends (2003-2013) of aromatics concentrations observed during springtime in Paris (Waked et al., 2016) and more globally with the gradual decline of NMHC emissions over the last few years in France.**
**Note that the study of Waked et al. (2016) did not examine the trends of oxygenated compounds.**

**Correction applied in the revised manuscript:** *"These differences in aromatics concentration levels are consistent with decreasing downward trends of NMHC recorded during springtime in Paris (Waked et al., 2016). We note that the study of Waked et al. (2016) about VOC trends in Paris only concerns NMHC and not oxygenated compounds. As these ones are significantly impacted by biogenic and secondary sources, it is not surprising to observe a different variation between 2007 and 2010."*

Waked, A., Sauvage, S., Borbon, A., Gauduin, J., Pallares, C., Vagnot, M.-P., Léonardis, T. and Locoge, N.: Multi-year levels and trends of non-methane hydrocarbon concentrations observed in ambient air in France, Atmos. Environ., 141, 263-275, 2016.

**25/** P11, L-34: Last sentence is quite generic.
**The corresponding sentence was removed from the revised manuscript.**

**26/** P11, L-29: "global studies".
**Correction applied in the revised manuscript:** *"Average VOC concentrations were also calculated in line with sampling periods of the other European and global studies over different years (see Table 2)."*

**27/** P12, L-3: Omit "time".
**The title of the subsection 3.3 was modified in the revised manuscript.**
**Correction applied:** *"Seasonal and diurnal variations".*

**28/** P12, L-15: Better to use a 24h time instead of am/pm, avoids confusion.
**When time is mentioned throughout the manuscript, the 24-h format is used now.**
* * *
**29/** P12, L-17ff: The increase occurs until 21h. And after that the levels stay quite high. Is there another (meteorological) reason for this? Interpretation of the graphs should be done in the discussion section.

**CO emissions are mainly related to road-traffic and/or wood burning activities, for which they would be more intense in autumn and in winter, respectively. The diurnal variability of CO is characterized by a "double wave" profile. A first peak is observed between 08-10 h and a second one from 18 to 21 h. After 21 h, CO levels stay quite high (240-260 ppb). This finding can be explained by on-going emissions but also by atmospheric dynamics. Indeed, the PBL is lower at night. This implies an accumulation of CO (and other pollutants) into the atmosphere. In addition, a few photochemical reactions occur due to lower OH concentrations and solar radiations. All these features explain quite high levels of CO at night.**

**Correction applied in the revised manuscript:** *"After 21h, CO levels stay quite high (240-260 ppb) due to several factors: on-going emissions (traffic and wood-burning activities), lower photochemical reactions and atmospheric dynamics (the shallower boundary layer leads to more accumulation of CO (and other co-emitted species))."*

**30/** P12, L-26: Format of citations should be "Gros et al. (2011)" etc.
**The format of citations was corrected in the revised manuscript.**
**Correction applied in the revised manuscript:** *"These observations are in agreement with the conclusions from Gros et al. (2011) and Gaimoz et al. (2011)."*

**31/** P12, L-32ff: The conclusions cannot be drawn from wind roses. As stated before a trajectory analysis or similar is needed for that.
**Please refer to the comment n°22.**

**32/** P12, L-4ff: Entire section: part of the whole section better fit into the discussion.
**As reported ahead, we have chosen to group results and discussions section together. Overwise, it would have led to too many repetitions. The whole section 3.3 presents and discusses seasonal and diurnal variabilities of some selected VOC with a focus on trace gases (NO, CO), aromatics and oxygenated species.**

**33/** P13, L-6: "Peaks" rather than increases; values high for longer time (see CO).
**Correction applied in the revised manuscript:** *"In winter and autumn, methanol shows a "double wave" profile with two peaks at 10-11 h and 19-20 h, suggesting the influence of anthropogenic activities (e.g. road-traffic, wood burning sources)."*

**34/** P13, L-10: It is a bit hard to read because the description jumps between the normal manuscript and the supplement. It might be worth to think about including important graphs that are discussed here into the main part and leave the supplement to really supplemental information.
**The diurnal cycle of methanol observed in winter and summer has been moved to Figure 7(c) in the revised manuscript.**
* * *
**35/** P13, L-8: "to" instead of "with".
**Correction applied in the revised manuscript:** *"A slight delay (1-2 hours) is observed for methanol in comparison to other primary species (for instance, aromatics), that highlights the secondary origin of this species."*

**36/** P13, L-10: "…, however, …"
**Correction applied in the revised manuscript:** *"This night-time maximum of methanol has already been observed in urban environments, however, with no clear explanation (Solomon et al., 2015)."*

**37/** P13, L-18: In the plots no clear difference can be seen between the two species.
**Acetaldehyde and acetone exhibit fairly comparable seasonal and diurnal cycles, suggesting that they share the same common source(s). Mean species concentrations remain stable throughout the year, with however, higher levels in summer. They increase from sunrise to a maximum at noon and slightly decrease during the afternoon. We found that this decreasing did not occur in the same way between the two species. From 12 – 18 h, average acetaldehyde concentrations are linearly decreasing (~ 1 µg.m$^{-3}$ or 0.5 ppb) while mean acetone levels show a slower decline rate (~0.5 µg.m$^{-3}$ or 0.25 ppb) with a tiny raise at 17h. This difference may be explained by emission strengths, but also by the reactivity of species (acetone is less reactive than acetaldehyde).**

**Correction applied in the revised manuscript:** *"For these two OVOC species, the reduction of concentrations does not occur in the same way. From 12 – 18 h, average acetaldehyde concentrations are linearly decreasing (~ 1 µg.m$^{-3}$ or 0.5 ppb) while mean acetone levels show a slower decline rate (~ 0.5 µg.m$^{-3}$ or 0.25 ppb) with a tiny raise at 17h. This finding depends on their emission sources and strengths (e.g. biogenic, solvents use), but also on their respective photochemical reaction rates (1.5 x 10$^{-11}$ cm$^3$ molecule$^{-1}$ s$^{-1}$ for acetaldehyde and 1.8 x 10$^{-13}$ cm$^3$ molecule$^{-1}$ s$^{-1}$ for acetone)."*

**38/** P13, L-24: These substances are not "emissions" but secondary products of isoprene oxidation.
**We agree methylvinylketone (MVK), methacrolein (MACR) and isoprene hydroxy hydroperoxides (ISOPOOH) are known to be the first-order isoprene oxidation products (Spaulding et al., 2003). Sorry for the confusion.**
Spaulding, R. S. (2003). Characterization of secondary atmospheric photooxidation products: Evidence for biogenic and anthropogenic sources. Journal of Geophysical Research, 108.
**Correction applied in the revised manuscript:** *"The formation of these secondary compounds mostly occurs in summer, but also in winter (Fig.7f)."*

**39/** P13, L-27f: Into methods section.
**Introductory sentences to this section were removed in the revised manuscript as corresponding information was already introduced in the methods section.**

**40/** P13, L-30: "such as" instead of "like".
**Correction applied in the revised manuscript:** *"The temporal source variations were assessed with independent parameters used as tracers of specific sources such as inorganic gases (NO, NO$_2$, CO), black carbon (BC) and meteorological data (temperature)."*
* * *
**41/** P14, L-7: "fair" instead of "good"
**Correction applied in the revised manuscript:** *"This factor 1 displays fair correlations with nitrogen monoxide/dioxide (NO/NO₂), carbon monoxide (CO) and black carbon (from its fossil fuel fraction), which are known to be relevant vehicle exhaust markers (0.53 < r < 0.64)."*

**42/** P14, L-9ff: Link to respective figure should be inserted; a table or small figure of the absolute contributions could be useful.
**The link to the respective figure has been inserted in the revised manuscript.**
**A table of the mean absolute contributions per factor and per month is reported in Section S7 in the Supplement of the new manuscript.**

**43/** Chapter 3.4: In general, reactivity of the different factors could be investigated in addition to stating only mass contributions; the biogenic factor and the motor vehicle factor would potentially gain importance compared to the factors with more stable gases such as background/natural gas and gas evaporation.
**As advised, the reactivity of different resolved PMF sources has been investigated taking into account the factor concentration of each species with their OH rate constant ($k_{OH}$).**

**Using the definition of the OH reactivity, the total reactivity of a source $R_{total(k)}$ can be estimated as follows:**

$$R_{total(k)} = \sum_{j=1}^{m} f_{kj} * k_{compound+OH}$$

**Where $f_{kj}$ is the $j$th species mass concentration from the $k$th source and $k_{compound+OH}$ represents the constant rate of a compound ($j$) with the OH radical ($k_{compound+OH}$ expressed in $cm^3.molecule^{-1}.s^{-1}$) (Atkinson et Arey, 2003).**

**The absolute total reactivity of the $k$th source ($s^{-1}$) is reported in Table 2. The reactivity of Motor Vehicle Exhaust (F1), Solvents use (F5) and Biogenic (F4) sources is among the highest ranging from 29 to 40 $s^{-1}$. These factors appear as the most reactive emission sources (as they are constituting of reactive species such as isoprene, aromatics, propene and ethylene) compared to factors with more stable gases (ethane, propane, butanes). As an example, the absolute reactivity of evaporative sources or natural gas and background factor is ranging from 13-18 $s^{-1}$.**

| | F1(MVE) | F2 (EVAP) | F3 (WB) | F4 (BIO) | F5 (SOLV.) | F6 (NGB) |
|---|---|---|---|---|---|---|
| $R_{total(k)}$ ($s^{-1}$) | 40 | 13 | 19 | 29 | 33 | 18 |

**Table 2 –** Absolute total reactivity of PMF factors expressed in $s^{-1}$.

**Then, the relative contribution of reactivity $R_{ik}$ of each factor ($k$) to the $i$th sample was calculated from the contribution $g_{ik}$ from the $k$th source to $i$th sample and the $R_{total(k)}$ from the source $k$:**

$$R_{ik} = \frac{g_{ik} \times R_{total(k)}}{\sum_{k=1}^{p} g_{ik} \times R_{total(k)}} \times 100$$

[20]
* * *
**The relative contribution of reactivity of each factor is illustrated in Figure. 4 (and Fig. 12 in the revised manuscript). Solvents use and motor vehicle exhaust factors are considered here as the two main sources exhibiting a high reactivity contribution (26 % and 23 %, respectively). This can be explained by high constant rates of aromatics and alkenes which are the major components of profiles associated to these two emission sources. The contribution of the biogenic source is surprisingly weak (17 %). Although isoprene is an extremely reactive species, this factor exhibits a dominating weight of OVOCs for which constant rates can be low. Instead, the relative contribution of the mixed source "natural gas and background" is surprisingly high (16%) due to the presence of aromatics (toluene and xylenes) in the factor profile. The lower contribution of reactivity is represented by the evaporative sources factor (5%) which contains more stable gases (propane, butanes).**

Atkinson, A. and Arey, J.: Atmospheric Degradation of Volatile Organic Compounds, Chem. Rev., 103(12), 4605-4638, doi:10.1021/cr0206420, 2003.

**This approach is not being given in a specific section, but is included in the few words in the Sub-section 3.5 (VOC source contribution).**

**In the revised manuscript, it now reads:** *"The reactivity of each modeled factor has also been investigated by considering the factor concentration of each species with their OH rate constant (kOH) (Atkinson and Arey, 2003) and is reported in relative (and absolute) contributions in Fig. 12. Among all the emission sources identified by PMF, solvents use and motor vehicle exhaust factors appear as the main reactive sources [26 % (33 $s^{-1}$) and 23 % (40 $s^{-1}$), respectively]. This can be explained by high constant rates of aromatics and alkenes mainly associated to these two emission sources. The contribution of the biogenic source is surprisingly weak (17 %). Although isoprene is an extremely reactive species, this factor exhibits a high weight of OVOCs for which constant rates can be low. Instead, the relative contribution of the mixed source "natural gas and background" is surprisingly high (16%) due to the presence of aromatics (toluene and xylenes) in the factor profile. The lower contribution of reactivity is represented by the evaporative sources factor [5 % (13 $s^{-1}$)] which contains more stable gases (propane, butanes)."*

[21]

[Figure]

**Figure 4 –** Relative and absolute contributions of reactivity of each PMF factors (% and s⁻¹, respectively).
* * *
**44/** P14, L-13: Why are the values so high in the middle of the night? Meteorology?

**The diurnal variability of the "Motor Vehicle Exhaust" source is mainly driven by the daily cycle of species constituting the factor profile (e.g. ethylene, propene, *iso-/n*-pentane and toluene). It is characterized by a "double wave" profile: a first peak is observed between 08-10h and a second one from 18 to 21 h, usually corresponding to morning and evening rush hours during weekdays. After 21h, this factor exhibits quite high contributions (7-8 μg.m$^{-3}$). This observation can be explained by several factors: on-going emissions (until midnight), lower photochemical reactions and atmospheric dynamics (the lower PBL leads to more accumulation of pollutants at night). Only after 1h and until 6h, the reduction of the source contributions is significant as additional vehicular emissions are now limited.**

**Correction applied in the revised manuscript:** *"After 21h, the absolute contributions of this factor stay quite high (7-8 μg.m$^{-3}$) due to several factors: on-going emissions (until midnight), lower photochemical reactions and atmospheric dynamics (the shallower boundary layer leads to more accumulation of pollutants at night)."*

**45/** P14, L-19: No reference to the tunnel experiment is given here. Since it is the value added to have these kinds of data it should be described here (and discussed later).
**Near-field measurements performed in a highway tunnel allowed us to characterize the fingerprint of the traffic source in Paris. This intensive experiment helps to highlight an average speciated profile representative of local primary vehicular emissions (exhausts). Photochemical reactions were considered of minor importance.**

**The originality of this study stands in using this near-field profile to refine the identification of a traffic source derived from modeling simulations. PMF results therefore appear more robust when comparing with independent speciated profiles.**

**To evaluate the relevance of the modeled traffic source, a comparison between speciated profiles from the tunnel experiment and the PMF analysis was performed and is reported in Figure 5 (in the Figure 9 in the revised manuscript). Traffic profiles are in general coherent and consistent amongst themselves (R= 0.83; R$^2$ = 0.69). Indeed, a good agreement is observed between these two profiles for the major species such as *iso-/n*-pentane, toluene, ethylene, propene. Instead, significant differences in mass contributions of ethane (almost a factor of 10), acetylene (considered as a key combustion compound emitted from traffic not identified in the PMF profile), isoprene (represented by evaporative sources) and oxygenated species were found.**

**These differences can potentially be explained by several reasons. Firstly, the proportion of VOC emitted from traffic may be different depending upon the types of vehicles/engines/fuels (Montero et al., 2010). VOC emissions can also be dependent on the use of vehicles (age, maintenance…), driving situations and thermal conditions (hot soak…). Secondly, the vehicle fleet composition can be another reason that explains these differences. In Paris intra-muros, the share of passenger cars was estimated between 60 and 80 % of the total composition of the fleet in circulation. In addition, the proportions of two-wheelers (10-20%) and Light Duty Vehicles (5-10%) have significantly increased in the recent years (Airparif, 2013). Traffic limitations prohibit heavy goods vehicles entering in the centre of Paris. In the tunnel, heavy vehicles are allowed (5% of the fleet composition). Light Duty Vehicles account for nearly 90% of the total fleet composition. Motorized scooters represent less than 2% of the total vehicles (Ammoura et al., 2014). Finally, PMF artefacts cannot be excluded. The apportionment of ethane**
* * *
**concentrations/contributions into the different PMF factors can also play an important role in the differences observed (factor of 10 for this species between near-field and PMF profiles). From the interpretability and fitting scores, we suppose that the contribution of ethane in the "Motor Vehicle Exhaust" factor is overestimated (same as the "Wood Burning" factor) at the expense of the mixed "Natural Gas and Background" source (for which ethane contributions seem to be underestimated – see the comment n°58).**

**Correction applied in the revised manuscript:** *"To evaluate the relevance of this factor, a comparison between speciated profiles from tunnel measurements (Fig. 1) and PMF simulations was done and is reported in Fig. 9. Traffic profiles are in general coherent and consistent amongst themselves, thus allowing to label this factor as a "Motor Vehicle Exhaust" source. Indeed, a good agreement is observed between these two profiles for the major species such as iso-/n-pentane, toluene, ethylene, propene. Instead, significant differences in mass contributions of ethane (almost a factor of 10), acetylene (considered as a key combustion compound emitted from traffic not identified in the PMF profile), isoprene (represented by evaporative sources) and oxygenated species were found.*

*These differences can potentially be explained by several reasons. Firstly, the proportion of VOC emitted from traffic may be different depending upon the types of vehicles/engines/fuels (Montero et al., 2010). VOC emissions can also be dependent on the use of vehicles (age, maintenance...), driving situations and thermal conditions (hot soak...). Secondly, the vehicle fleet composition is different in the centre of Paris and in a highway tunnel. Although the proportion of passengers' cars and Light Duty Vehicles (LDV) accounts for 60-90% of the total composition of the fleet in circulation in both cases, the share of two-wheelers and heavy goods vehicles can be different. Indeed, heavy vehicles are subject to traffic limitations prohibiting their entry in Paris whereas as they are allowed in the highway tunnel (5%). The proportion of two-wheelers is significant in Paris (10-20%) (Airparif, 2013) while they represent less than 2% of the total vehicles in the tunnel. Finally, PMF artefacts cannot be excluded. We suppose that the contribution of ethane in the "Motor Vehicle Exhaust" factor is overestimated (same as the "Wood Burning" factor) at the expense of the mixed "Natural Gas and Background" source (for which ethane contributions seem to be underestimated)."*

Airparif: Evolution de la qualité de l'air à Paris entre 2002 et 2012, Juillet 2013.

Montero, L., Duane, M., Manfredi, U., Astorga, C., Martini, G., Carriero, M., Krasenbrink, A. and Larsen, B.-R.: Hydrocarbon emission fingerprints from contemporary vehicle/engine technologies with conventional and new fuels, Atmos. Environ., 44, 2167-2175, 2010.

[24]

[Figure]

**Figure 5 –**Comparison of speciated profiles issued from the highway tunnel experiment and PMF simulations (F1 – Motor Vehicle Exhaust).
The species contributions are expressed in %. NF = Near-Field
* * *
**46/** P14, L-26: If part of the isoprene is from vehicles, it should be show up in F1, since it would be a combustion product instead of an evaporative source.

**Borbon et al. (2001) have shown that isoprene could be emitted by traffic emissions. In the present study, it also shows up in the tunnel profile (Fig. 2 in the initial manuscript) in which vehicle exhausts and gasoline evaporation are represented. Isoprene emissions usually depend on temperature and could then be associated with evaporative sources.**

**Note that the label of this source (previously named "Gasoline Evaporation") has been changed by a more generic term "Evaporative sources" as it includes gasoline evaporation from vehicles, storage, extraction and distribution of gasoline or Liquid Petroleum Gas (LPG).**

**47/** P14, L-31ff: Link to figure 9 should be given.
**The link to the respective figure has been inserted in the revised manuscript.**

**48/** P15, L-1: Correct citation: "Frachon (2009: pers. Communication)".
**As suggested above, the format of this citation was corrected in the revised manuscript.**

**49/** P15, L-30: A lot of acetonitrile was apportioned also to F4, what is that?
**The biogenic source is essentially associated with a large contribution of isoprene and oxygenated species (such as methanol or acetone). Among OVOC, acetonitrile also appears in this source. It could be attributed to a mixing with other temperature-related sources or an artefact from the PMF model (Leuchner et al., 2015).**

**Correction applied in the revised manuscript:** *"Amounts of light alkanes (butanes, iso-pentane, n-hexane) and acetonitrile were found in this profile and could be attributed to a mixing with other temperature-related sources or artefacts from the PMF model (Leuchner et al., 2015)".*

**50/** P16, L-6: "on" instead of "with". Chapter 3.4.3: reference to experiment needs to be included; a lot of ethane and propane is allocated to this factor that should very likely not be in there.
**Correction applied in the revised manuscript:** *"Wood burning emissions linked to home/building heating are obviously highly dependent on meteorological conditions and particularly on cold temperatures".*

**Near-field measurements performed at a fireplace allowed us to have an idea of the wood-burning fingerprint in Paris. As reported in the main text (see Section 2.4.2), this speciated profile is based on a limited number of data. This confers it a more qualitative (predominant species identification) than quantitative added-value. It can only be compared to see if main compounds are the same (as conducted here), but this limitation should be kept in mind.**

**Usually, a low contribution of ethane and propane in the "Residential heating/wood-burning" profile is observed from near-field measurements (Barrefors and Petersson, 1995; Wang et al., 2014) and in source apportionment (SA) analyses based on urban ambient air measurements (Lanz et al., 2007). However, it may be that higher contributions of these two species can be found (Badol et al., 2008, Waked et al., in preparation).**
* * *
**As reported for the "Motor Vehicle Exhaust" factor (see the comment n°45), we suspect there are some artefacts within modeled PMF results. As an example, the contributions of ethane and propane in the "Wood Burning" factor seem to be significantly overestimated (compared to the fireplace profile) at the expense of the mixed "Natural Gas and Background" factor (for which ethane contributions seem to be underestimated – see the comment n°58).**

Badol, C, Locoge, N., Léonardis, T. and Galloo, J.-C.: Using a source-receptor approach to characterize NMHC behaviour in a French urban area influenced by industrial emissions. Part I: Study area description, data set acquisition and qualitative data analysis of the data set, Sci. Tot. Environ., 389(2-3), 441-452, doi: 10.1016/j.scitotenv.2007.09.003, 2008.

Waked, A., Sauvage, S., Gros, V., Baudic, A., Sanchez, O., Rio, C. and Locoge, N. : Source apportionment of NMHCs in French urban sites (Paris and Strasbourg) for a period of 8 years (2005-2013), Atmos. Environ., in preparation.

Wang, H., Lou, S., Huang, C., Qiao, L., Tang, X., Chen, C., Zheng, L., Wang, Q., Zhou, M., Lu, S. and Yu. S.: Source Profiles of Volatile Organic Compounds from Biomass Burning in Yangtze River Delta, China, Aerosol Air Qual. Res., 14, 818-828, doi: 10.4209/aaqr.2013.05.0174, 2014.

**Correction applied in the revised manuscript:** *"It also includes ethylene (57.4%), benzene (22.7%) and oxygenated compounds, such as acetonitrile, acetaldehyde and methanol (with 18.3%, 12.6% and 6.6%, respectively). Acetonitrile is a hydrocarbon commonly used as a marker of biomass burning (Holzinger et al., 1999). All these chemical species typically reflect an anthropogenic source related to wood combustion processes (Lanz et al., 2008; Leuchner et al., 2015) in agreement with the fireplace emission profile (see Sub-section 2.4.2, Fig. 3). No full comparison between both speciation profiles was possible as the fireplace profile was based on a limited number of data. With this in mind, only a qualitative approach allowed to identify predominant species emitted from this source and confirm the term "wood burning" assigned to this factor."*
* * *
**51/** P16, L-16: "includes isoprene's"
**Correction applied in the revised manuscript:** *"In addition, this factor profile includes isoprene's oxidation products [...] with more than 48 %, methanol and acetone – a selection of compounds having a large contribution from biogenic emissions (Kesselmeier and Staudt, 1998; Guenther, 2002)."*

**52/** P16, L-24: r-value of the PAR correlation missing.
**During the MEGAPOLI-FRANCIPOL field campaigns, Photosynthetically Active Radiation (PAR) was not measured. That's the reason why r-value of the PAR correlation is missing. Usually, solar radiations and temperature are key factors governing biogenic emissions (Steiner and Goldstein, 2007). To avoid confusion, the corresponding sentence has been rephrased.**

**Correction applied in the revised manuscript:** *"Biogenic emissions are directly related to solar radiations (Steiner and Goldstein, 2007) and ambient temperature (r > 0.7)."*

**53/** P16, L-27: very high values at night, why? Stability of the atmosphere?
**The biogenic factor is mainly driven by isoprene and oxygenated species (MVK+MCR+ISOPOOH; methanol, acetone). Biogenic emissions are usually dependent on solar radiations and temperatures (Steiner and Goldstein, 2007) – gradually increasing during the day. High nighttime contributions of the biogenic source can be explained by the presence of oxygenated species (long-lived compounds already present in the atmosphere and/or secondarily formed from the oxidation of isoprene) in the profile combined with lower photochemical reactions and atmospheric dynamics (a low PBL height) at night.**

**Correction applied in the revised manuscript:** *"Biogenic emissions are directly related to solar radiations (Steiner and Goldstein, 2007) and ambient temperature (r > 0.7). For that reason, the highest factor contributions occur in summer (10 µg.m⁻³ on average) with a contribution up to 14.3 µg.m⁻³ in July. Daily mean contributions gradually increase from 09h. A slight delay (03h) is observed in comparison with diurnal temperature/solar radiations variations (for which values increase from sunrise at 06 h). We assume that chemistry affects this source factor as it takes part in the formation of secondary species (MAK+MVK+ISOPOOH, for instance) from the oxidation of primarily emitted compounds (isoprene, OVOC). Diurnal contributions reach their maximum at the end of the day (~19h). Highest nighttime contributions of this source can be explained by the presence of oxygenated species (long-lived compounds already present in the atmosphere and/or secondarily formed from the oxidation of isoprene) in the profile combined with lower photochemical reactions and atmospheric dynamics (a low PBL height) at night."*

**54/** P17, L-14: what does "panel 5" mean?
**Panel 5 represents "Monthly box and whiskers plots of the solvents use source. To avoid confusion, numbers from 1 to 5 were graphically attributed to each source. Consequently, Panel 1 = Motor Vehicle Exhaust; Panel 2 = Evaporative sources; Panel 3 = Wood Burning, Panel 4 = Biogenic; Panel 5 = Solvents use; Panel 6 = Natural Gas & Background.**

**55/** P17, L-14: the PBL is higher in the afternoon, that contradicts the statement of the diurnal pattern. Emissions might be higher and also a temperature dependency.
**The seasonal and diurnal variabilities of the "Solvents use" source are presented in Figure 10, Panel 5 (in the initial version of the manuscript). We assume that the daily variation of**

[28]
* * *
**this factor is dependent on the season considered. Diel variations of the factor is compared here between winter (top right) and spring/summer/autumn (bottom right).**

**In winter, factor contributions increase after 6 h and reach their maximum between 11 h and 19 h (15-20 µg.m$^{-3}$) before a long and gradual decline in the evening. Higher contributions in winter can be explained by lower photochemical reactions and atmospheric dynamics. A shallower PBL (and consequently, less intense vertical dynamics) leads to more accumulation of pollutants.**

**In spring/summer/autumn, factor contributions also increase at sunrise, but reach their maximum between 08 and 10 h (typical of anthropogenic activities). They progressively decrease during the afternoon. This gradual decline (not earlier observed in winter) is influenced by greater photochemical reactions and more intense vertical dynamics during these three seasons, leading to dispersion and dilution processes (and consequently, lower source contributions during the afternoon).**

**Correction applied in the revised manuscript:** *"The highest contribution of this factor is observed during winter (14.2 µg.m$^{-3}$) with a contribution of up to 20.9 µg.m$^{-3}$ in January. In winter, factor contributions increase at 6h and reach their maximum between 11h and 19h (15-20 µg.m$^{-3}$) before a long and gradual decline in the evening (see Fig.10, panel 5 – top right). Higher contributions in winter can be explained by lower photochemical reactions (combined with weaker OH concentrations/UV radiations) and atmospheric dynamics. Indeed, a shawoller PBL (and consequently, less intense vertical dynamics) leads to more accumulation of pollutants and thus to higher source contributions. The daily wintertime variability of this source is in agreement with the diel cycle of independent tracers (ethanol, butan-2-one).*

*Reconstructed contributions associated to this factor are also significant in summer (12.6 µg.m$^{-3}$ in July), which could be mainly explained by the evaporation of solvent inks, paints and other applications during that month due to higher temperatures. In spring/summer/autumn, factor contributions also increase at sunrise, but reach their maximum between 08 and 10 h (typical of anthropogenic activities). They progressively decrease during the afternoon (see Fig.10, panel 5 – bottom right). This gradual decline (not earlier observed in winter) is influenced by greater photochemical reactions and more intense vertical dynamics during these three seasons, leading to dispersion and dilution processes (and consequently, lower source contributions during the afternoon)."*

**56/** P17, L-18: not shown.
**The diurnal cycles of the "Solvents use" factor observed in winter and spring/summer/autumn are presented in Figure 10, panel 5 in the revised manuscript.**

**57/** P17, L-22: "except"
**The word "expected" was substituted by "expect".**

**58/** P17, L-31: BLH not defined. Better to use only one abbreviation, e.g. PBL Chapter 3.4.6: the annual course of the background does not fit with similar observations; there should be a maximum in February and minimum in late summer due to the OH reactivity (delayed by 2 months to the annual solar cycle because of the long-lifetime of ethane).
**Only the abbreviation PBL (Planetary Boundary Layer) has been used throughout the revised manuscript.**
* * *
**The factor mixing "Natural Gas & Background" (NGB) sources is mainly driven by ethane (45% of species sum). The annual course of this PMF factor does not fit with that usually observed in the literature whereas the diurnal cycle does (see Fig.10 – panel 6). As reported in previous comments n°45 and 50, PMF artefacts cannot be ruled out. Indeed, a problem with the distribution of ethane within PMF factors was raised. We assume that higher ethane contributions were assigned to the "Motor Vehicle Exhaust" and "Wood Burning" factors. Consequently, this allocation issue has effects on the NGB source (as ethane is a key component in the factor profile). For that reason, we suspect that the NGB factor contributions were underestimated (especially in winter) for the benefits of the "Wood Burning" factor (another source significantly contributing during this season).**

**Correction applied in the revised manuscript:** *"The average contribution of this mixed source (combining both natural gas and background emissions) is in the range of 9.2 μg.m$^{-3}$ during the whole studied period. Lowest source contributions were observed in winter, which does not fit with that reported in the literature. As mentioned in the "Motor Vehicle Exhaust" and "Wood Burning" sub-sections (3.4.1 and 3.4.3, respectively), PMF artefacts cannot be ruled out. Indeed, a problem with the distribution of ethane (considered as the key species of the mixed source) within PMF factors was raised. We assumed that higher ethane contributions were partly assigned to the "Motor Vehicle Exhaust" and "Wood burning" factors. Consequently, we assumed that the "Natural Gas and Background" factor contributions were underestimated (especially in winter) for the benefits of the "Wood Burning" factor (another source significantly contributing during this season).*

**59/** P18, L-7ff: cannot be concluded only from wind roses. Statements need to be supported that it really results from different air masses, since annual cycle does not fit.
**Please refer to previous comments n°22 and 31.**

**60/** Chapter 3.5: pure discussion, also better fits into discussion section.
**Please refer to the previous comment n°32.**

**61/** P18, L-21: background not really fresh emission, but aged air mass
**To avoid confusion, the corresponding sentence has been rephrased.**
**Correction applied in the revised manuscript:** *"This Source Apportionment (SA) analysis concluded that the predominant sources at the receptor site were road-traffic-related activities (including motor vehicle exhaust, 15 % of the Total VOC (TVOC) mass on the annual average, and evaporative sources, 10 %), with the remaining sources from natural gas and background (23 %), solvents use (20 %), wood burning (17 %) and biogenic activities."*

**62/** P18, L-21ff: additional information on the contribution of reactivity of each factor would be very useful.
**Please refer to the comment n°43.**

**63/** P19, L-4: most of the conclusions are quite speculative and need to be supported by more profound analysis.
**Please refer to previous comments n°22, 31 and 59.**

**64/** P20, L-9: this discussion would fit to the discussion of factors.
**Please refer to the previous comment n°32.**
* * *
**65/** P20, L-13: "combination"
**Correction applied in the revised manuscript:** *"To accurately compare VOC sources proportions between 2007 and 2010 (for a similar combination of hydrocarbons and masses), the contribution of each main factor was recalculated for the specific time period May-June 2010 (Fig.11, right pie chart)."*

**66/** P20, L-33: there are some other longer time series, but the dataset indeed is very useful.

**67/** P21, L-23: absolute values could also be interesting in addition for comparison to other studies
**Please refer to the previous comment n°42.**

**68/** P22, L-12: why were exactly these studies chosen? There are many other studies described in the literature from e.g., Houston, Santiago de Chile, etc.
**This Source Apportionment (SA) study was initially compared with some SA studies performed in Europe and in the World in order to evaluate the consistency and the representativeness of PMF results at a large scale. The selection of SA studies is nowhere near complete. SA studies presented in the manuscript were chosen on the basis of one or more criteria: (i) VOC (NMHC and/or OVOC) measurements should be undertaken in urban background sites (to omit any local influences of emission sources), (ii) SA should be performed from long-term VOC time series, (iii) similar source categories (factor identification) should be reported.**

**Following reviewer' suggestions, we decided to keep the comparison between our results with SA studies conducted in Europe, but to shorten the comparison with other cities in the world. Nevertheless, we now mention more references of global SA studies without commenting in detail their results.**

**Correction applied in the revised manuscript:** *"The consistency in VOC contributions in European urban areas raises the question of their representativity at a larger scale. There are currently many other urban SA studies described in the literature (e.g. Jorquera and Rappenglück, 2004 – Santiago/Chile; Buzcu et al., 2006 – Houston/TX; Brown et al., 2007 – LA; Cai et al., 2010 - Shanghai/China; Morino et al., 2011 – Tokyo/Japan; Yurdakul et al., 2013 – Ankara/Turkey; Zheng et al., 2013 – Mexico). Results of these studies are not detailed here but one common feature for European and global scales is the importance of the road-traffic source (between 30 % and 50%). One difference concerns the industrial sector which plays (in the investigated European cities) a lower role than in studied urban areas from other continents.*

*Governmental regulations and standards to control pollutants emissions and economic developments may differ between European countries and the world. The location of sampling points (or distances from main sources) and meteorological conditions can strongly affect VOC concentrations and their respective emission sources in the considered urban environments."*

Jorquera, H. and Rappenglück, B.: Receptor modeling of ambient VOC at Santiago, Chile, Atmos. Environ., 38, 4243-4263, doi:10.1016/j.atmosenv.2004.04.03
* * *
**69/** Graphics: Some of the axis titles and labels (e.g., Figs. 8, 9 but also all others) are quite small and might be hard to read when formatted to the actual page size.
**The font size of the axis titles and labels for all figures was increased in the revised manuscript for an easier reading.**
* * *
**70/** Fig. 5 could go to supplement or omitted if replaced with a more profound air mass investigation.
**The performed analysis of the wind fields was replaced with a more elaborate air mass investigation presented in Figure 5. Wind roses plots were moved to the supplement (Section S5).**

**71/** Fig. 8: Logarithmic scale makes it hard to see the "real" profile for absolute contributions.
**Usually, absolute contributions of each species apportioned in the factor (resolved by PMF) are plotted on a logarithmic scale (see Norris et al., 2014). For a better appreciation of PMF outputs, we opted for a linear scale. We also omitted the Total Variable called "VOC".**

Norris, G., Duvall, R., Brown, S. and Bai, S. (2014). EPA Positive Matrix Factorization (PMF) 5.0: Fundamentals & User Guide. Prepared for the US. Environmental Protection Agency (EPA), Washington, DC, by the National Exposure Research Laboratory, Research Triangle Park; Sonoma Technology, Inc., Petaluma.

**72/** P47, L-12: The factors could be shown in supplement, would be interesting to see.
**PMF modeling simulations derived from the FRANCIPOL dataset (March – November 2010) were performed in order to support to the assumptions related to the accuracy and robustness of final PMF results. These PMF outputs were incorporated into the Section S1 in the Supplement.**

**73/** P47, L-6ff: quite important information. Could be moved to methods section of main text.
**The dataset composition was already mentioned in the methods section of the main text. In the A1 section, only a recap is now provided.**

**74/** Appendix A: here the calculation of uncertainties could be shown.
**As reported in the previous comment n°14, a comprehensive calculation of uncertainties could not be performed on all data and therefore, we chose to not dedicate an additional appendix to this issue.**

---

## Author Comment (AC2) · 29 Jul 2016

Dear Referee#2,

We would like to thank you for your general feedback and your useful comments/questions for improving the quality of this manuscript. All your comments were taken into account in the revised version.

Authors' answers were uploaded as a *.pdf file and were displayed as a supplement to this comment.

Sincerely, Mrs. Alexia Baudic

[Figure]

Please also note the supplement to this comment:
http://www.atmos-chem-phys-discuss.net/acp-2016-185/acp-2016-185-AC2-supplement.pdf
* * *
[Figure]

**Supplement:**

* * *
**Acp-2016-185: "Seasonal variability and source apportionment of volatile organic compounds (VOC) in the Paris megacity (France)" -- Baudic et al.**

**Authors' Responses to Referee #2**

We would like to thank the Referee #2 for her/his general feedback and each of her/his useful comments/questions for improving the quality of this manuscript. All of them have been taken into account when preparing the revised version of the manuscript. In the present document, authors' answers to the specific comments addressed by Referee #2 are mentioned in **blue**, while changes made to the revised manuscript are shown in *italic*.

General/scientific comments

This paper reports the results of a source apportionment by positive matrix factorization (PMF) of the concentrations of a suite of ambient VOCs measured in urban background air in Paris over a period of several months from January to November 2010. VOCs were measured both by on-line GC and by PTR-MS. In order to help in the assignment of some of the factors emanating from the application of PMF, the authors compared the speciated VOC profiles of the factors with speciated VOC profiles the authors separately measured at three locations where they assume that a single emission source will dominate the ambient VOC, specifically: (1) measurements during busy (and traffic jam) periods in a highway tunnel, to represent vehicle related VOC emissions; (2) measurements close to a domestic gas flue, to represent natural gas source; and (3) measurements at a fireplace facility, to represent residential wood-burning emissions (the authors acknowledge that their measured VOC profile from this source may be less quantitative than for other source profiles). To further assist in the assignment of PMF factors to particular VOC emission sources the authors also make use of additional co-located atmospheric compositional data available to them, such as NO, CO and black carbon (BC).

The authors present results in which the ambient urban background VOC has been apportioned into six sources, each of which has been assigned an identification, albeit that one factor is assigned to be a mixed natural gas/background source. The analysis presented includes speciated VOC profiles for each factor and monthly and average-hourly variation in the absolute contribution (ug/m3) of each of the six identified factors/sources. The largest contributions in total are from traffic-related activities (through two identified factors: motor-vehicle exhaust, and gasoline evaporation), although all six identified sources have not that dissimilar relative contributions, on average. A noteworthy observation is significant contributions to ambient VOC from wood burning, 18% on average but up to ~50% at times in winter. Biogenic emissions were also reported to be significant, 15% on average but more in summer.

The authors have a large dataset of time-resolved speciated VOC over an extended time period, almost a year. They have used standard, but appropriate, statistical methods to endeavour to decompose the ambient measurements into individual source contributions. These are statistical, rather than dispersion-/chemistry-based source apportionments. These methods have been widely employed to apportion ambient PM, but less so for VOC.

The presentation of results and their discussion are largely descriptive, in the sense that the authors present the details for their factor/source contributions and their monthly and hourly variations, which are rationalised with general text about anticipated behaviour of particular sets of pollutant mixtures in the urban background atmosphere. The authors also present a qualitative and quantitative comparison of their VOC source apportionment with previous literature.
* * *
Although in one sense the presentation of this work could be described as 'formulaic' – following previous data presentation and analysis styles – nonetheless the large dataset presented here for a large European city presents a valuable addition to the VOC apportionment literature. A particular feature is the presentation of VOC speciated profiles for three potential VOC sources, although the authors acknowledge some shortcomings in these. The paper is already lengthy and contains much new data, supported by detailed descriptions of the data collection and processing protocols and results. As an overall summary, the content of the paper is suitable for publication in ACP.

**The authors thank Reviewer#2 very much for his/her attention to our manuscript. All comments addressed by both reviewers have been taken into account in the preparation of the revised version of the manuscript. In this respect, several figures and tables were notably added and modified (e.g. air masses trajectory analysis – Fig. 5; comparison between "highway tunnel" and "motor vehicle exhaust" profiles – Fig. 9; relative and absolute contributions of reactivity of each PMF factor – Fig. 11) and in the supplementary (e.g. PMF profiles issued from simulations using the FRANCIPOL dataset & comparison of speciated PMF profiles issued from our two different datasets – Section S1; representativeness of meteorological parameters in 2010 – S4; mean absolute contributions of factors per month in 2010). Please note that figures and tables numeration is now different in this new version.**

Technical comments

**1/** The tables and figures are generally clearly presented. The written text is largely unambiguous in conveying its meaning, but it is overly long in places. There are instances where introductory sentences to a section could be substantially abbreviated, or even deleted as repeating what the reader will have picked up from the methods section. The authors should be encouraged to edit text further for conciseness of expression.
**Following reviewer's suggestions, the authors made efforts to write a revised version of the manuscript with conciseness. Some rephrasing also aimed at bringing the content a bit more to the point.**

The following are more specific comments.

**2/** P6, L-18: should read "Raw data were corrected using…."?
**The initial sentence "Raw data were compensated using…" was substituted by the following one:** *"Raw data were corrected using the algorithm described in Weingartner et al. (2003) and Sciare et al. (2011)".*

**3/** P6, L-25: delete "is" before "analyzer"
**Correction applied in the revised manuscript:** *"Nitrogen monoxide and dioxide (NO, NO$_2$) were measured by chemiluminescence using an AC31M analyzer (Environment SA, Poissy, France) and ozone (O$_3$) was monitored with an automatic UltraViolet absorption analyzer (41M, Environment SA, Poissy, France)."*

**4/** P6, L-32: delete "at"
**Correction applied in the revised manuscript:** *"Standard meteorological parameters […] were provided by the French national meteorological service "Météo-France" from continuous measurements recorded at the Paris-Montsouris monitoring station (14th district, 48°49'N, 02°20'E), located about 2 km away from the LHVP site."*

[2]
* * *
**5/** P7, L-22: should read "except for"
**Correction applied in the revised manuscript:** *"This combination of hydrocarbon species and masses is similar to that from Gaimoz et al. (2011), except for iso-butene."*

**6/** P8, L-9: I don't understand what is being described in the sentence beginning "Finally, these different processings…." Please rephrase this sentence to make clear to what "it" is referring in this sentence and to clarify what the procedure undertaken was.
**In this study, only 59% of isoprene data are modeled by the PMF technique, what is not objectively considered as sufficient. To address this problem, input values of isoprene were modified when detecting missing data. We have investigated the possibility of replacing missing values with data more appropriate than the median. Indeed, a "virtual" averaged pattern was calculated from 1-h real samples observed in June and August (to keep the summer variability of isoprene). Another option was to preserve raw data of isoprene (no subsequent changes) and to increase the analytical uncertainty initially estimated at 15% (gradually from 15 % to 30%) – instead of categorizing isoprene as *weak* (as uncertainties are tripled implying a lower Signal to Noise). This was intended to better display this biogenic compound. PMF simulations were performed by considering these different options. As a consequence of these tests, no significant improvement on the quality of modeling isoprene was observed. Regular statistical parameters such as $R^2$, slope, slope/intercept SE (Standard Error) were used to draw such conclusions. As empirical tests have not helped, isoprene is still categorizing as *strong*.**

**In the previous version of the manuscript, "processings" referred to "empirical tests" and "it" to isoprene. To avoid any ambiguity in the meaning of this sentence, corresponding lines were rephrased.**

**Correction applied in the revised manuscript:** *"To address this lack of isoprene data, several empirical tests (e.g. simulating an averaged seasonal/diurnal cycle of isoprene or increasing the analytical uncertainty of raw data from 15 % up to 30 %) were conducted within PMF simulations with the aim of better modeling the variability of this compound. As a consequence of these tests, no significant improvement on the quality of modeling isoprene was observed. Finally, isoprene is still categorizing as strong here".*

**7/** P10, L-26: the phrase "fairly comparable" is not scientifically precise.
**The corresponding sentence was rephrased.**
**More specific statements about weather conditions are given here.**
**Correction applied in the revised manuscript:** *"Air temperatures observed during the campaign were comparable to standard values determined by the French national meteorological service "Météo-France" (available at: http://meteofrance.com), with however an uncommon cold wintertime (Bressi et al., 2013- Fig. S1a). Temperatures recorded in January and February 2010 were respectively between -2°C and -3.5°C below normal values (see. Section S2 in the Supplement). Extreme unusual cold-air outbreaks and a few snow flurries affected the Paris region, thus explaining higher temperature anomalies during that period. Levels of hours of sunshine and rainfall were globally consistent with standard values, with however some discrepancies in winter/autumn and spring, respectively (Fig. S3)."*

**8/** P10, L-29: "a few flurries" of what? Please write specific statements about the nature of the weather.
**Please refer to the previous comment n°7.**
* * *
**9/** P11, L-15: the phrase "contribute to the tune of" is too colloquial; please use more direct wording.

**Correction applied in the revised manuscript:** *"Both alkanes and OVOCs significantly contribute up to 75 % of the TVOC concentrations."*

**10/** P11, L30-L35: there are several instances in these lines of text of negative values for VOC concentrations. These are surely some aberration (albeit repeated aberration) of typing error. Please correct.

**A comparison between mean concentrations of aromatics and OVOCs measured in this study and ambient air levels reported in the literature for different urban atmospheres (see Table 2) was made. It aims at highlighting existing differences in VOC levels monitored for a given time period between our work (Paris) and another European or global studies (the selection is not exhaustive here). We note that VOC concentrations measured in European cities (Paris, Barcelona, and London) are in the same order of magnitude depending on the compounds. Instead, more significant differences in VOC levels were found between Paris and Houston, Beijing, Mohali or Mexico City.**

**The negative values were due to differences between VOC concentrations from our study compared to others. But to avoid confusion, differences are now only given with absolute values.**

**Correction applied in the revised manuscript:** *"Average VOC concentrations were also calculated in line with sampling periods of the other European and global studies over different years (see Table 2). In this study, measured VOC levels were in the range of those found with some European cities (Barcelona, London – from 0.1 to 2.1 ppb concentration differences). However, average VOC levels observed in Paris were significantly lower than those measured in Houston (USA – from 0.1 to 6.9 ppb concentration differences) and more particularly in Beijing (China - from 2.5 to 8.9 ppb), in Mexico City (Mexico – from 0.1 to 27.4 ppb) and in Mohali (India – from 0.9 to 32.7 ppb)".*

**It is important to note that Table 2 was modified and all reported values are given in ppb (see the comment n° 13 – Referee 1).**

**11/** P13, L-3: I do not understand the scientific sense of the sentence starting "With an atmospheric residence time…" How does the statement at the end of this sentence (about methanol emissions contributing to background levels) derive from, or otherwise relate to, the text at the start of the sentence about methanol residence time? Please reword to clarify.

**We agree with this comment. There is no direct link between the beginning and the end of this sentence. This additional information did not bring an added value to the scientific approach. To avoid confusion, we opted to omit the start of this sentence.**

**Correction applied in the revised manuscript:** *"Methanol is usually released into the atmosphere by vegetation and man-made activities contributing to a relatively high background levels during most of the year."*

**12/** P14, L-24: I do not follow the scientific logic here. The text appears to state that iso-pentane is known to be a key tracer for gasoline evaporation, but also to say that iso-pentane was not present in the speciated profile the authors have assigned in their work to gasoline evaporation.

**In this study, the "evaporative sources" factor is mainly characterized by the presence of propane and butanes (*iso-/n-*). The species composition of the F2 profile is consistent**

[4]
* * *
with those obtained from other SA studies (Brown et al., 2007; Gaimoz et al., 2011; Waked et al., 2016). In addition to $C_3$-$C_4$ species, pentanes (*iso-/n-*) and toluene are also considered as key tracers for gasoline evaporation (Salameh et al., 2016 – in preparation).  A small contribution of these species was identified in F2. However, their highest contributions were assigned to the "Motor Vehicle Exhaust" factor" (F1) in agreement with that observed during the tunnel experiment.
To avoid any misunderstanding, the corresponding sentence was removed.

**13/** P14, L-32: please explain more clearly what about the monthly change "remains ambiguous".
The seasonal cycle of "Evaporative sources" profile is expected to be significant in spring and summer (when higher temperatures are observed). The minimum mean contribution of this source is observed in July when road-traffic emissions are usually significant (See Figure 10 – Panel 1).
This explanation is given with more details in the main text (P14, L-31 – P15, L-5).

**14/** P20, L19: The phrasing that the mean temperature was "in the range +/- 20 degC" does not make sense. Either quote the range, or quote the mean and some recognized statistic of the variation about the mean. Likewise for later in this sentence in connection with "+/- 16 degC".
**Correction applied in the revised manuscript:** *"The mean temperature recorded in May-June 2007 and 2010 was 20°C and 16°C, respectively".*

**15/** Caption of Table 1: It would be helpful for the caption to remind the reader with a statement of the time resolution of the raw data from which these statistical summaries are derived, and of the time duration/dates of the total dataset.
Caption of Table 1:
Statistical summaries ($\mu g.m^{-3}$) of selected VOC concentrations measured at urban background sites. Statistics were calculated based on hourly mean data, initially obtained every 30 min. (ethane > isoprene) and every 5 to 10 min. (for aromatics and OVOCs). These measurements were undertaken from 15 January to 22 November 2010 (~ 10 months. A conversion factor is provided here to convert VOC concentrations ($\mu g.m^{-3}$) into (ppb) mixing ratios.